# *Bifidobacterium longum* and prebiotic interventions restore early-life high-fat/high-sugar diet-induced alterations in feeding behavior in adult mice

Cristina Cuesta-Marti [1,2] ✉, Eduardo Ponce-España [1,3,4,10], Friederike Uhlig [1,2,5,10], Iris Stoltenborg [6], Luiza A. Wasiewska [2,7], Lamiah Kareem [2], Dara Hedayatpour [2,8], Loreto Olavarría-Ramírez [1,2], Cristina Rosell-Cardona [2], Thomaz. F. S. Bastiaanssen [2], Gabriel. S. S. Tofani [1,2], Benjamin Valderrama [1,2], Klara Vlckova [2], Suzanne L. Dickson [6], Aonghus Lavelle [1,2], Catherine Stanton [2,8], R. Paul Ross [2], John F. Cryan [1,2], Timothy G. Dinan [2,9], Gerard Clarke [2,9], Siobhain M. O'Mahony [1,2] & Harriët Schellekens [1,2] ✉

An unhealthy diet disrupts feeding behavior and the gut microbiota, but whether early-life dietary effects persist, or can be restored later in life, remains unclear. We investigated whether microbiota-targeted interventions (FOS + GOS or *Bifidobacterium longum* APC1472) could restore early-life high-fat/high-sugar (HFHS) diet-induced feeding alterations in adult female and male mice. HFHS exposure exclusively in early-life induced persistent, sex-specific feeding alterations in adult mice, despite normalized body weight. Early-life HFHS diet reduced hypothalamic cells expressing feeding-related markers (POMC, GHSR, PNOC, NOD2) in adult mice. Females were more vulnerable, with reduced LEPR+ cells and disrupted arginine/tryptophan metabolism, while males showed impaired peptidoglycan sensing and steroid metabolism. We show that microbiota interventions restore these effects via distinct mechanisms. FOS + GOS induced extensive microbiome compositional shifts and sex-specific restoration of gut-brain pathways, while *B. longum* APC1472 induced greater behavioral restoration with minimal microbiome compositional changes. These findings highlight sex-specific vulnerabilities and mechanism-dependent therapeutic potential of microbiota-based interventions after exposure to early-life unhealthy diets.

Early life, including the perinatal and postnatal time window, is a critical period for offspring development[1,2]. Environmental factors such as diet, metabolic status or stress have been shown to impact neurodevelopment and can lead to long-lasting adverse metabolic effects, such as glucose intolerance, hyperleptinemia and weight gain, with enduring metabolic risk across multiple generations of offspring[3,4]. Exposure to unhealthy diets, such as high-fat diet (HFD) or western diets, during early-life has been shown to have effects on offspring's

appetite regulation[5–8], food intake[9–11], preference for palatable foods[10–12] as well as dopaminergic signaling and circuitry[11,13,14] involved in hedonic feeding[15,16]. Understanding these early-life dietary influences on feeding behavior is crucial given the global rise in obesity and eating disorders, as well as the increasing consumption of Western-style diets, including during pregnancy and early-life periods[17,18]. Feeding behavior involves highly interconnected homeostatic mechanisms that maintain energy balance based on metabolic needs, and non-homeostatic mechanisms driven by reward and motivation[19–21]. It follows that neurons that sense metabolic needs to maintain energy balance[19–21] and neuronal populations involved in reward-based feeding[22–25], along with non-neuronal cells[26–29] respond to hormones regulating food intake and energy balance, for instance via nutrient sensing[30]. These peripheral signals reach the brain directly through the blood-brain barrier (BBB) or indirectly via vagal afferent signaling into the nucleus tractus solitarius (NTS) in the brainstem. From there, projections reach the hypothalamus, which integrates them to regulate homeostatic and reward-based feeding behaviors[19,21,31]. Whilst the NTS itself has been implicated as a critical regulator of meal size and satiation[32–34], the hypothalamus acts in concert with brainstem and limbic pathways to integrate peripheral and central signals controlling appetite. Within the hypothalamus, the arcuate nucleus (ARC) is especially important for feeding regulation[19,21,31]. Its strategic location near the median eminence and adjacent to the third ventricle allows it to be one of the first points of access for peripheral signals, such as gut hormones and metabolites[19,21]. The anorexigenic propiomelanocortin (POMC) neurons and the orexigenic agouti-related protein (AgRP)/neuropeptide Y (NPY) neurons are two opposing neuronal populations located in the ARC and crucial for the homeostatic regulation of appetite and food intake[35–37]. These neurons express multiple receptors, integrating several locally produced and circulating neurotransmitters and neuropeptides, including leptin, ghrelin, NPY, and melanocortins[38]. Maternal dietary interventions during the perinatal period have been shown to alter the expression of these hypothalamic neuropeptides and induce metabolic programming in offspring[7,39].

The gut microbiota has emerged as a modulator of hypothalamic processes, including neuroendocrine signaling and metabolic regulation[40,41]. This modulation may occur through multiple pathways, including vagal-brainstem-hypothalamic circuits[34,35] and the blood-brain interface at the median eminence[21,42,43]. Recently, the gut microbiota has been postulated as a key regulator of both homeostatic[37,40,44] and reward-based feeding[45–48]. Mice born and raised without microbiota (germ-free) have higher food intake and altered expression of hypothalamic markers of food intake regulation, including the orexigenic NPY and AgRP, and the anorexigenic POMC[40]. Depletion of the microbiota in mice using an antibiotic cocktail has also been shown to induce a higher consumption of and preference for palatable foods, as well as alterations in reward signaling and neuronal activation in reward-related regions including the nucleus accumbens and the ventral tegmental area[45–47]. Microbiota-targeted interventions, including pre-, pro-, or synbiotics or microbiota-derived metabolites such as short-chain fatty acids (SCFAs) and caseinolytic peptidase B (ClpB), have also been shown to impact the regulation of food intake[49–55] and non-homeostatic feeding[47,49,56].

Early life is a critical period for the establishment of the gut microbiota[57], which is significantly impacted by diet[58–65]. *Bifidobacterium* species, particularly *B. longum* subsp. *infantis*, and *B. breve*, are key early colonizers of the infant gut microbiota due to their unique ability to metabolize human milk oligosaccharides found in breast milk[66–68]. Through the fermentation of complex carbohydrates, *Bifidobacterium* can produce metabolites such as acetate and lactate, which have been shown to modulate central nervous system (CNS) function[69] and other physiological processes[66,67]. A reduction in the abundance of *Bifidobacterium* spp. has been correlated with chronic diseases, such as

obesity[70]. Interventions with specific *Bifidobacterium* strains have beneficial effects for the host, including their metabolic status[71,72]. While enduring colonization of probiotics is not required to exert biological effects on the host[73,74], *B. longum* strains have been shown to colonize the entire gastrointestinal tract (including the small intestine, cecum, and colon), with the highest abundances typically detected in the colon[75–77]. For instance, the putative probiotic *B. longum* APC1472 has previously been shown to improve metabolic health in HFD-fed mice as well as in overweight and obese humans[72], and to attenuate signaling by the orexigenic hormone, ghrelin[78]. The prebiotics fructo-oligosaccharides (FOS) and galacto-oligosaccharides (GOS) have previously been shown to increase the abundance of *Bifidobacterium*[79]. Individual administration of FOS and GOS has been shown to prevent HFD-induced obesity and obesity-associated metabolic disorders[80,81].

While the gut microbiota has been shown to modulate homeostatic and hedonic feeding pathways in adult studies[40,45–47,49], and, separately, to influence the host's developing neuro-immune-endocrine pathways[57,59,82], the mechanisms through which the early-life microbiota affects the homeostatic and hedonic feeding systems remain to be fully elucidated. Overall, whether early-life exposure to an unhealthy diet can prime enduring alterations in offspring homeostatic and non-homeostatic behaviors, and the underlying mechanisms, are not yet fully understood. Here, we investigate the potential of microbiota-targeted approaches, via prebiotic combination FOS + GOS or the putative probiotic *B. longum* APC1472, to attenuate detrimental enduring effects of an early-life high-fat/high-sugar (HFHS) diet, as a model of the western diet, on offspring's feeding behavior in adulthood. Our findings indicate that microbiota-targeted interventions can restore the enduring, sex-specific alterations in feeding behavior and associated molecular alterations induced by an early-life HFHS diet exposure in adult female and male mice via distinct mechanisms. Importantly, early-life HFHS diet reduces the abundance of *Bifidobacterium* genus, which is restored by FOS + GOS administration, and *B. longum* APC1472 administration increases *B. longum* abundance in the offspring.

## Results

### Microbiota-targeted interventions ameliorate the effects of an early-life high-fat/high-sugar diet on adult feeding behavior in a sex-specific manner and are associated with increased *Bifidobacterium* or *B. longum* abundance

Maternal exposure to HFHS diet significantly increased weight gain (Supplementary Fig. S1C, D and Supplementary Table S1) and gonadal and mesenteric white adipose tissue weights (Supplementary Fig. S1E, F) in dams, which were restored by supplementation with the prebiotic FOS + GOS and the putative probiotic *B. longum* APC1472. HFHS diet did not alter brown adipose tissue, cecum, or liver weight, nor colon or small intestine length in dams when normalized to body weight (Supplementary Fig. S1E, F). The offspring of these HFHS-exposed obese dams (Fig. 1A, B; Supplementary Tables S1 and S2) showed increased body weight gain during the milk suckling and post-weaning period, which was attenuated by *B. longum* APC1472 in female offspring at postnatal day (PND) 28 and PND35, and in male offspring at PND35 (Fig. 1B and Supplementary Data 1). This increase in body weight, with greater gain in male than female offspring, was particularly apparent after weaning, when offspring consume HFHS themselves rather than through the mother's milk, suggesting that this is related to food intake (Fig. 1A, B and Supplementary Fig. S1G). Female and male offspring receiving the HFHS diet removed significantly more food from the food hopper (Fig. 1C and Supplementary Fig. S1G). Interestingly, we found that early-life HFHS-exposed offspring did not consume all the food removed from the food hopper but instead, crumbled part of it (Fig. 1C and Supplementary Fig. S1G), a phenomenon previously described as food crumbling or grinding[83]. Supplementation with *B. longum* APC1472 attenuated both increased

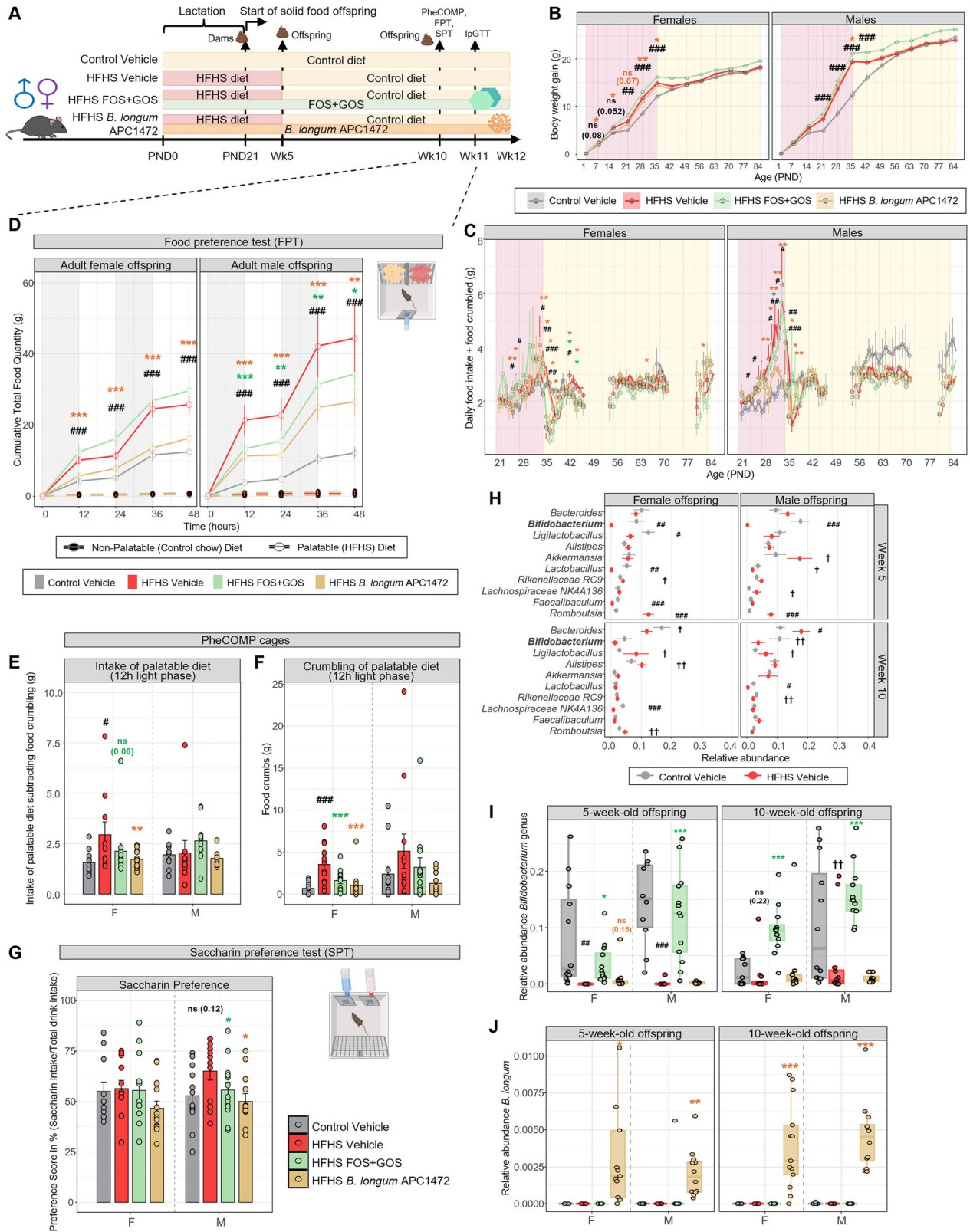

food intake and food crumbling observed in female and male offspring during the early-life exposure to the HFHS diet, to levels similar to those of control offspring (Fig. 1C and Supplementary Fig. S1G). Body weight normalized following the dietary switch from HFHS diet to control chow diet at week 5 (Fig. 1B and Supplementary Data 1). While in female mice, there were no significant differences after 1 week on the control diet, the normalization of body weight in male offspring

was delayed for another week (Fig. 1B and Supplementary Data 1), in line with previous studies showing sexual dimorphism in body weight and metabolic health depending on the diet exposure and age[84–86].

The food crumbling behavior toward the palatable diet persisted into adulthood (Fig. 1D) when mice, after 6 weeks on control diet, were given a choice between control chow diet and palatable HFHS diet (food preference test). Adult male and female mice exposed to the

**Fig. 1 | Exposure to a high-fat/high-sugar diet during early-life has long-term effects on *Bifidobacterium* and *B. longum* abundance and on feeding behavior in a sex-specific manner in adult female and male mice. A** Summary of experimental design. **B** Weekly body weight gain of female and male offspring ($n = 12–13$/group/sex) throughout the study. **C** Daily food intake of female and male offspring throughout the study ($n = 5–6$ cages/group/sex with 12–13/group/sex). Data represent food removed from the hopper, including both food consumed and food crumbled, during early-life high-fat/high-sugar (HFHS) diet exposure. Missing values of daily food intake represent the days that feeding behavior tests were performed. **B, C** Pink background highlights the weeks where the corresponding groups received the HFHS diet during early-life, and yellow highlights the weeks where all the groups received the control chow diet. **D** Food preference test (FPT) in adult female and male mice ($n = 11–13$/group/sex; 10–11 weeks of age) given a choice between non-palatable diet (control chow; black dots and discontinuous lines) and palatable diet (HFHS diet; white dots and continuous lines) during dark (gray) and light phase (white). **E** Quantification of palatable HFHS diet intake in adult female and male mice ($n = 9–13$/group/sex; 10–11 weeks of age) during the light phase (12 h) in the PheCOMP system. **F** Quantification of food manipulation as food crumbling of the palatable HFHS diet in adult female and male mice (10–11 weeks of age) during the light phase (12 h) in the PheCOMP system (Female: control vehicle $n = 12$, HFHS vehicle $n = 12$, HFHS FOS + GOS $n = 12$, HFHS *B. longum* APC1472 $n = 13$; Male: control vehicle $n = 12$, HFHS vehicle $n = 12$, HFHS FOS + GOS $n = 13$, HFHS *B. longum* APC1472 $n = 12$). **G** Saccharin preference test (SPT) in adult female and male mice ($n = 11–13$/group/sex; 10–11 weeks of age) given a choice between saccharin and water over 48 h. Preference score calculated as % of saccharin intake/total intake. Relative abundance of the **H** top genera altered by the early-life HFHS diet, **I** *Bifidobacterium* genus, and **J** *Bifidobacterium longum* species in female and male offspring at postnatal day (PND) 35 (week (wk) 5) and PND70 (wk 10) ($n = 12–13$/group/sex). Values represented as group average showing the biological replicates and error bar represented as mean ± SEM. **B–G** Data analyzed for diet effects by 3-way ANOVA followed by Tukey's two-sided pairwise post-hoc test, between Control Vehicle and HFHS Vehicle (#$p < 0.05$, ##$p < 0.01$, ###$p < 0.001$), and restoration effects by each individual microbiota-targeted intervention (Fructo-oligosaccharides and Galacto-oligosaccharides (FOS + GOS) or *B. longum* APC1472) analyzed by planned orthogonal contrast (pairwise comparison of Control Vehicle–HFHS Vehicle, followed by comparison of HFHS Vehicle–HFHS FOS + GOS or HFHS Vehicle–HFHS *B. longum* APC1472) (*$p < 0.05$, **$p < 0.01$, ***$p < 0.001$, in green for restoration effects of FOS + GOS, and orange of *B. longum* APC1472). **H–J** Pairwise comparisons were performed using two-sided t-tests between the two dietary interventions within Vehicle-treated animals, and between HFHS-exposed animals, including HFHS Vehicle-treated animals and each microbiota intervention (prebiotic FOS + GOS and putative probiotic *B. longum* APC1472) for *P*-adjusted (*q*) values using Benjamini-Hochberg correction for multiple comparison ($n = 11–13$/group/sex). Restoration effects of microbiota-targeted interventions were evaluated when both diet and intervention effects were significant and the effect sizes showed opposing directions (†$q < 0.2$, ††$q < 0.1$, #$q < 0.05$, ##$q < 0.01$, ###$q < 0.001$ for diet effect; †$q < 0.2$, ††$q < 0.1$, *$q < 0.05$, **$q < 0.01$, ***$q < 0.001$ in green for restoration effects of FOS + GOS, and orange of *B. longum* APC1472). Data presented as boxplots display the range from the first to the third quartile (box), and the median (center line) boxplots display the minimum, first quartile, median, third quartile, with whiskers extending to 1.5× the interquartile range. For sex effect and interactions effects and additional detailed statistical analysis, see https://github.com/CristinaCuesta-Marti/earlylifeHFHS_feeding_microbiota_2025/tree/outputs/overall_stats_lm. Precise sample sizes (*n* numbers) per condition and outcome are detailed in Supplementary Data 4. Schematic diagrams in **A, D, G** were created with BioRender: Schellekens, H. (2025) https://BioRender.com/w4ypyhz. Mouse illustration in panel A was obtained from SciDraw (https://scidraw.io/drawing/549, by Heath Robinson, CC BY 4.0, DOI: 10.5281/zenodo.7058520). g grams; h hour, IpGTT intraperitoneal glucose test, kcal kilocalories, ns not significant.

HFHS diet in early life demonstrated an increased preference for the palatable diet and removed significantly more of it from the food hoppers compared to control mice (Fig. 1D). We observed that *B. longum* APC1472 normalized the heightened food preference induced by the early-life HFHS exposure in adult male and female adult mice, to levels seen in control animals in females (Fig. 1D). Supplementation with the prebiotic FOS + GOS also normalized food preference and/or crumbling behavior only in male mice (Fig. 1D).

To separate the food consumption from food manipulation behavior, we measured the food crumbling, and food intake in a separate experiment using PheCOMP cages, where adult female and male mice were given access to the palatable HFHS diet during the light phase. Female, but not male mice, exposed to the early-life HFHS diet increased both food consumption (Fig. 1E) and food crumbling behavior (Fig. 1F) of the palatable diet compared to control mice. A non-significant increase in food crumbling was also observed in early-life HFHS exposed male mice compared to the control mice (ANOVA: $F_{2, 33} = 2.19$, $p = 0.128$, $\eta^2 = 0.12$; Post-hoc Tukey (diet effect): mean difference = 2.73, $p = 0.33$) and *B. longum* APC1472 supplementation reduced this food manipulation or crumbling ($F_{1, 34} = 4.12$, $p = 0.050$, $\eta^2 = 0.11$) (Fig. 1F). Both microbiota-targeted interventions attenuated the food intake (Fig. 1E) and the crumbling behavior (Fig. 1F) in females exposed to early-life HFHS, with *B. longum* APC1472 supplementation completely restoring those effects (Figs. 1E and 1F). The microbiota-targeted interventions also restored the early-life HFHS diet-induced increases in eating rate, first meal size and average meal size in the adult female mice (Supplementary Fig. S1H).

To assess differences in non-caloric preference for sweetness, we performed the saccharin preference test[87–89]. Interestingly, early-life HFHS diet induced a trend for increased saccharin preference in adult males (ANOVA: $F_{2, 34} = 2.23$, $p = 0.123$, $\eta^2 = 0.12$; Post-hoc Tukey (diet effect): mean difference = 12.2%, $p = 0.123$), which was significantly reduced by the prebiotic FOS + GOS and *B. longum* APC1472 (Fig. 1G). Intriguingly, no differences in saccharin preference were observed in females (Fig. 1G). Next, we assessed striatal content and turnover of dopamine and serotonin (5-HT) for their involvement in the regulation of food intake and reward-based feeding[15,16,90,91]. While the exposure to an early-life HFHS diet did not have significant lasting effects on the content of dopamine, its metabolites DOPAC and HVA, or its precursor L-DOPA in the striatum of adult female or male mice, adult female mice showed some effects on its metabolism, including an increased L-DOPA/dopamine ratio (ANOVA: $F_{2, 14} = 9.31$, $p = 0.003$, $\eta^2 = 0.57$; Post-hoc Tukey (diet effect): mean difference = 0.004, $p = 0.052$) and a reduced DOPAC/HVA ratio (ANOVA: $F_{2, 14} = 2.96$, $p = 0.085$, $\eta^2 = 0.30$; Post-hoc Tukey (diet effect): mean difference = −0.234, $p = 0.306$). The latter of which was significantly increased by FOS + GOS intervention (Supplementary Fig. S2A). Early-life HFHS did not have enduring effects on DOPAC/dopamine and HVA/dopamine ratios in the striatum of adult females or males (Supplementary Fig. S2A). Interestingly, early-life HFHS diet induced a trend toward reducing 5-HT levels in the striatum of adult male mice (ANOVA: $F_{2, 14} = 9.06$, $p = 0.003$, $\eta^2 = 0.56$; Post-hoc Tukey (diet effect): mean difference = −52.45, $p = 0.080$) and both microbiota-targeted interventions significantly increased 5-HT levels. Additionally, *B. longum* APC1472 normalized the non-significant reduction in serotonin turnover (5-HIAA/5-HT ratio) induced by the early-life HFHS diet in adult female mice (ANOVA: $F_{2, 14} = 9.24$, $p = 0.003$, $\eta^2 = 0.57$; Post-hoc Tukey (diet effect): mean difference = −0.064, $p = 0.090$) (Supplementary Fig. S2A), while diet intervention did not alter 5-HIAA content in females or males. Both microbiota-targeted interventions normalized the trend toward reducing striatal noradrenaline levels in adult males exposed to an early-life HFHS (ANOVA: $F_{2, 15} = 3.13$, $p = 0.073$, $\eta^2 = 0.29$; Post-hoc Tukey (diet effect): mean difference = −59.41, $p = 0.068$) (Supplementary Fig. S2A). Notably, early-life HFHS exposure did not have enduring effects on locomotor activity in adult female or male mice (Supplementary Fig. S2B).

To further elucidate the long-lasting impact of an early-life HFHS diet on offspring's glucose metabolism in adulthood, we performed an intraperitoneal glucose test (IpGTT) (Supplementary Fig. S2C, D). Surprisingly, adult male offspring exposed to HFHS diet in early-life showed an improved glucose tolerance compared to control mice, as

demonstrated by a higher clearing rate (Supplementary Fig. S2C, D). These effects were normalized by the administration of FOS + GOS after 60–120 min of glucose injection and of *B. longum* APC1472 after 90 and 120 min post-glucose injection in male mice (Supplementary Fig. S2C). FOS + GOS administration also normalized the diet-induced reduction in AUC in adult males (Supplementary Fig S2D). In females, there were no diet-induced changes in glucose clearance (Supplementary Fig. S2C, D). Interestingly, early-life HFHS diet induced a non-significant increase in basal glucose levels in adult females and males (Supplementary Fig. S2E) and supplementation with *B. longum* APC1472 significantly reduced these levels in female mice (Supplementary Fig. S2E).

Furthermore, adult male mice exposed to early-life HFHS diet had significantly higher circulating HDL and LDL levels, the former of which was restored by supplementation with the prebiotic FOS + GOS (Supplementary Table S3). Supplementation with *B. longum* APC1472 reduced the non-significant increase in gonadal and mesenteric white adipose tissue weight (normalized to body weight) induced by an early-life HFHS diet exposure in adult male mice (Supplementary Fig. S3A). Early-life HFHS diet did not have significant enduring effects on brown adipose tissue, liver, or cecum weights (normalized to body weight) in male mice. In females, whilst there were no significant diet effects in any organ measurements, *B. longum* APC1472 decreased colon length (normalized to body weight) (Supplementary Fig. S3A). Importantly, FOS + GOS intervention increased caecum weight compared to all other groups in both sexes (Supplementary Fig. S3), consistent with established prebiotic effects of oligosaccharides through enhanced microbial fermentation[92,93]. This physiological adaptation likely contributes to the observed body weight increase in FOS + GOS-administered mice (Fig. 1B). No changes were observed in the levels of glucose, triglycerides, protein, alanine transaminase, or aspartate aminotransferase or total cholesterol in plasma of adult female and male mice exposed to the early-life HFHS diet compared to controls (Supplementary Table S3). We also measured circulating levels of active ghrelin in 12-week-old fasted female and male mice, but observed no differences (Supplementary Fig. S3B).

HFHS-related changes in microbiota have been associated with alterations in feeding behavior[45,46]. In our study, alpha diversity analysis revealed that early-life HFHS diet induced a trend towards increased microbial richness (Chao1) in 5-week-old female mice ($t_{6.49} = 1.47$, $q = 0.189$, effect size = −0.25, $R^2$ marginal = 0.036) and no significant alterations in richness were observed in 5-week-old males or 10-week-old female and male mice ($q > 0.2$) (Supplementary Fig. S4A–C). Early-life HFHS diet did not significantly alter evenness as measured by Shannon and Simpson indexes at week 5 or 10 of age in either sex ($q > 0.2$) (Supplementary Fig. S4A–C). We also observed that early-life HFHS diet exposure altered beta diversity in 5-week-old male offspring (PERMANOVA: ($F_{3, 45} = 5.99$, $p = 0.001$, $R^2 = 0.285$); pairwise (diet effect): $F_{1, 23} = 7.65$, $q = 0.0012$, $R^2 = 0.258$ and female (PERMANOVA: $F_{3, 45} = 5.83$, $p = 0.001$, $R^2 = 0.280$; pairwise (diet effect): $F_{1, 23} = 7.31$, $q = 0.002$, $R^2 = 0.249$). These alterations were normalized in 10-week-old males after 5 weeks on control diet during adolescence and adulthood (Males PERMANOVA: ($F_{3, 45} = 6.19$, $p = 0.001$, $R^2 = 0.292$); Males pairwise (diet effect): $F_{1, 23} = 1.96$, $q = 0.09$, $R^2 = 0.082$; Females PERMANOVA: ($F_{3, 45} = 7.08$, $p = 0.001$, $R^2 = 0.320$); Females pairwise (diet effect): $F_{1, 23} = 3.02$, $q = 0.002$, $R^2 = 0.121$) (Supplementary Fig. S4D). The prebiotic FOS + GOS altered beta diversity compared to both control (Females pairwise: $F_{1, 23} = 6.11$, $q = 0.002$, $R^2 = 0.217$; Males pairwise: $F_{1, 24} = 5.35$, $q = 0.0012$, $R^2 = 0.189$) and early-life HFHS-exposed mice (Females pairwise: $F_{1, 23} = 10.15$, $q = 0.002$, $R^2 = 0.316$; Males pairwise: $F_{1, 24} = 6.21$, $q = 0.0012$, $R^2 = 0.213$) at 5 weeks of age in females and males. At 10 weeks, FOS + GOS continued to alter beta diversity in females (Pairwise Control Vehicle vs HFHS FOS + GOS: $F_{1, 23} = 10.88$, $q = 0.002$, $R^2 = 0.331$; Pairwise HFHS Vehicle

vs HFHS FOS + GOS: $F_{1, 23} = 13.93$, $q = 0.002$, $R^2 = 0.388$) and males (Pairwise Control Vehicle vs. HFHS FOS + GOS: $F_{1, 24} = 6.82$, $q = 0.002$, $R^2 = 0.229$; Pairwise HFHS Vehicle vs. HFHS FOS + GOS: $F_{1, 24} = 8.68$, $q = 0.002$, $R^2 = 0.274$) (Supplementary Fig. S4D). *B. longum* APC1472 administration altered beta diversity compared to HFHS Vehicle in 5-week-old (Pairwise: $F_{1, 24} = 2.53$, $q = 0.021$, $R^2 = 0.099$) and 10-week-old females (Pairwise: $F_{1, 24} = 2.07$, $q = 0.044$, $R^2 = 0.083$), but not in males at either timepoint (5 weeks: $F_{1, 23} = 0.87$, $q = 0.432$, $R^2 = 0.038$; 10 weeks: $F_{1, 23} = 1.24$, $q = 0.288$, $R^2 = 0.053$). HFHS diet reduced the relative abundance of *Bifidobacterium* genus in dams (Supplementary Fig. S4E, F) and 5-week-old female and male and 10-week-old male offspring (Fig. 1H, I). Importantly, FOS + GOS administration was able to restore the alteration in *Bifidobacterium* relative abundance induced by the HFHS diet exposure in dams (Supplementary Fig. S4F) and 5- and 10-week-old female and male offspring (Fig. 1I). This suggests that the restoration by the prebiotic administration may occur via compensation for the lack of fermentable fiber and other carbohydrates in the HFHS diet. The supplementation with *B. longum* APC1472 increased the relative abundance of the *B. longum* species in dams (Supplementary Fig. S4G) and 5-week- and 10-week-old female and male offspring (Fig. 1J). We observed an increase in the Bacillota/Bacteroidota ratio following early-life HFHS exposure compared to control mice in both sexes at 5 weeks of age with persisting effects at 10 weeks only in male mice, which was restored by FOS + GOS administration in 5-week and 10-week-old female mice (Supplementary Fig. S4H).

Altogether, these results demonstrate that microbiota-targeted interventions ameliorate enduring sex-specific effects of an early-life HFHS diet on feeding behavior and glucose metabolism in adult female and male mice, through mechanisms involving increased *Bifidobacterium* or *B. longum* abundance.

## Enduring sex-specific effects of an early-life exposure to an unhealthy diet on host-derived and microbiota-derived metabolites

Here, we studied the impact of an early-life HFHS diet exposure and the microbiota-targeted interventions on circulating host- and microbial-derived metabolites in 12-week-old female and male mice. We observed that early-life HFHS diet had enduring effects by differentially altering the blood metabolome of 12-week-old female and male mice in a sex-specific manner (Fig. 2A). Adult female mice exposed to the early-life HFHS diet displayed alterations in circulating metabolites belonging to different metabolite classes, including the microbiota-derived metabolites kynurenine, indole-3-carboxaldehyde, hippuric acid, or pantothenic acid, and the host-derived metabolites creatinine, creatine, tryptophan, 15(S)-HETE, lactic acid or the L-amino acids alanine and arginine (Fig. 2A). Early-life HFHS diet-exposed adult males displayed alterations in microbiota-derived metabolites including the secondary bile acid deoxycholic acid, and the tryptophan metabolite formylkynurenine, as well as in the host-derived metabolites tryptophan, serotonin, thiamine, succinic acid, and the primary bile acids cholic acid and β-muricholic acid (Fig. 2A). Supplementation with the prebiotic FOS + GOS restored some of the altered circulating metabolites in adult mice exposed to early-life HFHS diet (Fig. 2A). In females, this included the amino acid arginine, the fatty acid metabolism intermediate decanoylcarnitine, the nucleotide metabolite deoxyuridine, and the polyamine spermidine. In males, this included the primary bile acids cholic acid and β-muricholic acid, the secondary bile acid deoxycholic acid, the nucleoside 5-methylcytidine, and corticosterone (Fig. 2A). Noteworthy, we also observed a reduction in corticosterone circulating levels in adult male mice that were exposed to an early-life HFHS diet, which was also normalized by both microbiota-targeted interventions (Supplementary Fig. S4I). Supplementation with the putative probiotic *B. longum* APC1472 was able to significantly restore

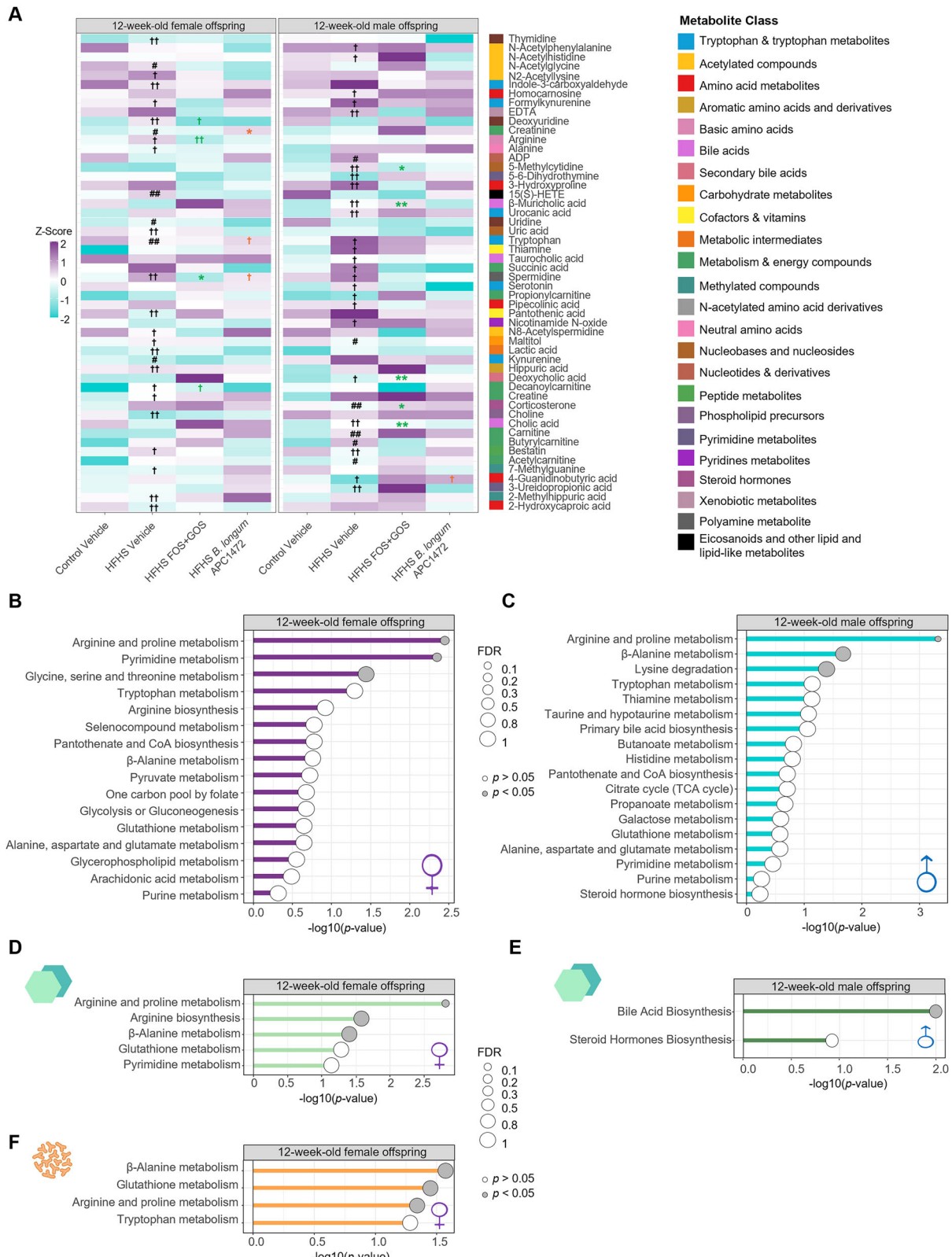

metabolites altered by the early-life HFHS diet, including tryptophan, creatinine and spermidine in adult females, and 4-guanidinobutyric Acid (4-GBA) in adult males (Fig. 2A).

Pathway enrichment analyses of the blood metabolome identified arginine and proline, pyrimidine, and glycine, serine and threonine metabolism as significantly altered by early-life HFHS diet exposure in adult female mice (Fig. 2B), while β-alanine, arginine and proline

metabolism, and lysine degradation were significantly altered in adult male mice (Fig. 2C). Tryptophan metabolism was also an enriched pathway in adult male and female mice exposed to the early-life HFHS diet, although it did not reach statistical significance (Fig. 2B, C). Supplementation with FOS + GOS significantly restored metabolites involved in arginine biosynthesis, and β-alanine, arginine and proline metabolism in early-life HFHS exposed adult female mice (Fig. 2D), and

**Fig. 2 | Exposure to a high-fat/high-sugar diet during early-life has long-term sex-specific effects on blood metabolome of adult female and male mice.**
**A** Heatmap of top altered circulating metabolites by the exposure to a high-fat/high-sugar (HFHS) diet during early-life compared to control diet (P-adjusted ($q$) < 0.2), and the effect of the supplementation of the microbiota-targeted interventions, either the prebiotic FOS + GOS or the bacterial strain *B. longum* APC1472 in 12-week-old female and male mice ($n$ = 6/group/sex). Enrichment analysis of blood metabolites altered by the early-life HFHS exposure in 12-week-old (**B**) female and (**C**) male mice ($q$ < 0.2) ($n$ = 6/group). Enrichment analysis of blood metabolites altered by the exposure to a HFHS diet during early-life and restored by the supplementation with the prebiotic FOS + GOS in 12-week-old (**D**) female (light green) and (**E**) male (dark green) mice ($q$ < 0.2) ($n$ = 6/group). **F** Enrichment analysis of blood metabolites altered by the exposure to a HFHS diet during early-life and restored by the supplementation with the bacterial strain *B. longum* APC1472 in 12-week-old female mice ($q$ <0.2) ($n$ = 6/group). **A**–**F** Heatmap and enrichment analysis were performed on centered log-ratio (CLR)-transformed values of altered and/or restored blood metabolites of 12-week-old female and male ($n$ = 6/group/sex).
**A** Pairwise comparisons were performed using two-sided t-tests between the two dietary interventions within Vehicle-treated animals, and between HFHS-exposed animals, including HFHS Vehicle-treated animals and each microbiota intervention (prebiotic FOS + GOS and putative probiotic *B. longum* APC1472) for P-adjusted ($q$) values using Benjamini–Hochberg correction for multiple comparison ($n$ = 11–13/group/sex). Restoration effects of microbiota-targeted interventions were evaluated when both diet and intervention effects were significant and the effect sizes showed opposing directions (†$q$ < 0.2, ††$q$ < 0.1, #$q$ < 0.05, ##$q$ < 0.01, ###$q$ < 0.001 for diet effect; †$q$ < 0.2, ††$q$ ˂ 0.1, *$q$ < 0.05, **$q$ ˂ 0.01, ***$q$ < 0.001 in green for restoration effects of FOS + GOS, and orange of *B. longum* APC1472). **B**–**F** Pathway enrichment analysis was performed using MetaboAnalyst's hypergeometric test with input metabolites selected based on the previous statistical analysis for metabolites altered by the diet (**B**, **C**) and restored by the microbiota-targeted interventions (**D**–**F**) ($q$ < 0.2). Pathway significance was assessed with False Discovery Rate (FDR) correction. Data as transformed $p$ values (−log10($p$-value)), where circle size represents the enrichment ratio (FDR), and gray circles highlight the significant enrichment ($p$ < 0.05). For sex effect and interactions effects and additional detailed statistical analysis, see https://github.com/CristinaCuesta-Marti/earlylifeHFHS_feeding_microbiota_2025/tree/outputs/overall_stats_lm. Schematic diagrams in **B**–**G** were created with Affinity Designer 2. CoA coenzyme A, TCA tricarboxylic acid cycle.

restored metabolites involved in bile acid and steroid hormone biosynthesis in adult male mice (Fig. 2E). Glutathione metabolism as well as β-alanine, arginine and proline metabolism were the pathways enriched based on the metabolites restored by *B. longum* APC1472 supplementation in adult female mice exposed to an early-life HFHS diet (Fig. 2F).

Next, we assessed the content of SCFAs, including acetate, butyrate, propionate and valerate, and the BCFAs isobutyrate and isovalerate in the cecum of 5- and 12-week-old female and male mice (Supplementary Fig. S5A, B), to investigate whether these microbiota-derived metabolites may contribute to the early-life HFHS diet-induced alterations in adult feeding behavior. At week 5, females exposed to the early-life HFHS showed a non-significant increase in acetate ($t_{11.34}$ = −1.58, $p$ = 0.141, $d$ = 0.85) and butyrate ($t_{7.38}$ = −2.06, $p$ = 0.076, $d$ = 1.20) levels, with no restoration by FOS + GOS or *B. longum* APC1472 interventions, even though FOS + GOS induced a non-significant reduction in butyrate levels ($t_{13.30}$ = 2.08, $p$ = 0.116, $d$ = −1.00) (Supplementary Fig. S5A). Early-life HFHS exposed males showed significant reductions in all SCFAs and BCFAs analyzed compared to control at week 5, with FOS + GOS and *B. longum* APC1472 administration restoring acetate levels (Supplementary Fig. S5A). No significant changes were observed in the SCFAs levels in 12-week-old female or male mice exposed to the early-life HFHS diet compared to control (Supplementary Fig. S5B).

To explore the microbial origins potentially responsible for SCFA production, we examined the SCFA-producing genera altered by the early-life HFHS diet exposure (Supplementary Data 2 and Supplementary Fig. S6D, together with other taxonomic levels (phyla, order, family) in Supplementary Fig. S6A–C; comprehensive data available at https://github.com/CristinaCuesta-Marti/earlylifeHFHS_feeding_microbiota_2025/tree/outputs/overall_stats_lm). At week 5, early-life HFHS altered 42 bacterial genera in females and 37 in males ($q$ < 0.2). Among the altered genera, key SCFA producers included *Intestinimonas* and *Bifidobacterium* (altered in both sexes), while *Roseburia, Faecalibaculum, Anaerotruncus*, and *Lachnoclostridium* were female-specific, and *Lachnospiraceae* subgroups (FCS020, NK4A136) were male-specific alterations. At week 10, early-life HFHS induced persistent alterations in 20 genera in females and 29 in males ($q$ < 0.2). Several SCFA-producing genera showed persistent alterations across both timepoints: *Intestinimonas* in both sexes, *Clostridium sensu stricto* 1 in females, and *Dorea, Lachnospiraceae* subgroups (including FCS020 and UCG-006) in males. Notably, FOS + GOS administration restored 15 genera in 10-week-old females and 20 genera in 10-week-old males (Supplementary Data 2). FOS + GOS consistently restored *Intestinimonas* at both timepoints in both sexes and *Romboutsia* at both timepoints in females. *B. longum* APC1472 restored 5 genera in 10-week-old females and 9 genera in 10-week-old males (Supplementary Data 2). Interestingly, *B. longum* APC1472 restored *Lachnospiraceae* family members across timepoints in both sexes. Overall, these findings highlight sex-specific alterations of the microbiome induced by the early-life HFHS diet and sex- and intervention-specific restoration effects.

To explore persistent functional alterations relevant to gut-brain communication underlying the microbiota compositional changes observed at week 10, we performed functional profiling of KEGG orthologs using the PICRUSt2[94] pipeline. At week 10, females exposed to the early-life HFHS diet had long-term alterations in 13 gut-brain modules (GBMs) (Supplementary Fig. S7A). FOS + GOS restored 10 of these modules, including degradation of dopamine, tryptophan, 17-beta-estradiol, glutamate, and inositol, while *B. longum* APC1472 restored only acetate degradation and butyrate synthesis in 10-week-old females (Supplementary Fig. S7A). Early-life HFHS altered 5 GBMs, comprising synthesis of propionate, DOPAC, kynurenine, and nitric oxide in 10-week-old male mice, with FOS + GOS restoring 4 modules (all except nitric oxide synthesis), while *B. longum* APC1472 showed a trend toward restoring DOPAC synthesis ($t_{18.3}$ = −1.30, $q$ = 0.208, $d$ = 0.53) (Supplementary Fig. S7A).

To identify functional pathways linking persistent microbiota changes at week 10 to systemic metabolic outcomes, we filtered PICRUSt2-derived pathways to retain only those involved in the metabolism of blood metabolites significantly altered by early-life HFHS diet or restored by interventions in adult mice (12 weeks of age) ($q$ < 0.2). At 10 weeks, females exposed to the early-life HFHS diet had alterations in 13 metabolic pathways predominantly involved in arginine metabolism, fatty acid biosynthesis, and tryptophan metabolism (Supplementary Fig. S7B). FOS + GOS administration restored 5 pathways, mainly involved in arginine and fatty acid metabolism, and *B. longum* APC1472 restored only fatty acid β-oxidation. In males, early-life HFHS altered 3 pathways related to fatty acid metabolism and arginine biosynthesis, with both interventions showing restorative effects on fatty acid β-oxidation. These findings highlight that most circulating metabolites altered by early-life HFHS in adult mice corresponded to the predicted functional capacities of the microbiota related to respective metabolic pathways.

Taken together, these results indicate that an early-life HFHS exposure has sex-specific lasting alterations in circulating metabolites, including microbiota-derived ones, as well as microbiota-SCFA profiles and microbiota functionality in adult mice, which can be ameliorated via microbiota-targeted interventions.

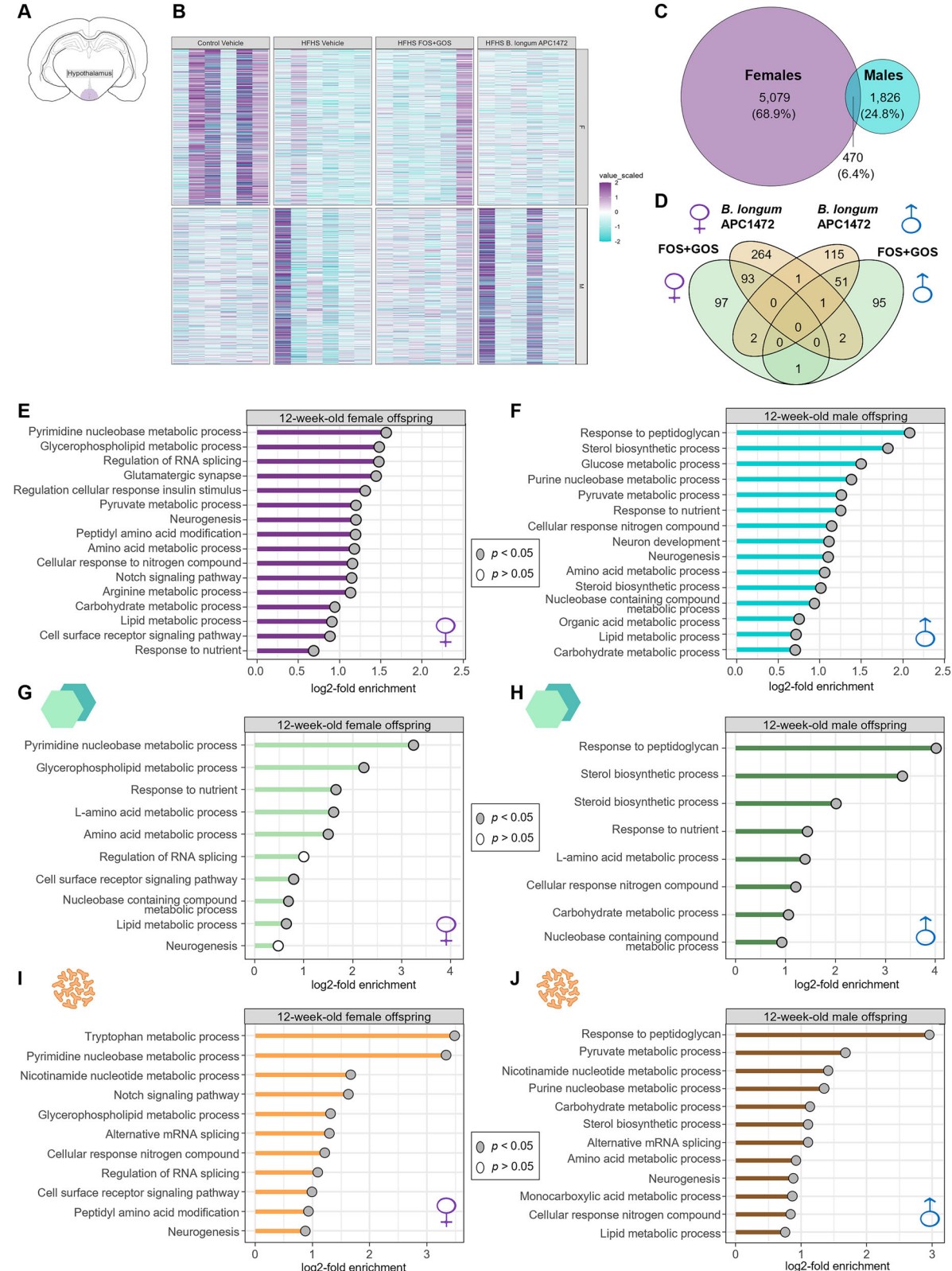

## Early-life HFHS has an enduring and sex-specific impact on hypothalamic transcriptomic profile in adult mice

To identify potential molecular signatures contributing to the observed behavioral phenotypes, we assessed the long-lasting effect of an early-life HFHS diet on the transcriptomic profile of the hypothalamus of 12-week-old female and male mice, given the hypothalamus's central role in feeding regulation and energy homeostasis[19,21]. We

showed that an early-life HFHS diet had long-lasting sex-specific effects on the transcription of hypothalamic genes with a higher number of altered genes in adult female than male mice (5070 vs. 1826) (Fig. 3A–C). We observed that supplementation with either the pre-biotic FOS + GOS or the *B. longum* APC1472 impacted the hypotha-lamic transcriptomic profile, restoring some of the molecular alterations induced by early-life HFHS in a sex-specific manner, with

**Fig. 3 | Exposure to a high-fat/high-sugar diet during early life has long-term sex-specific effects on transcriptomic profile of the hypothalamus of adult female and male mice. A** Schematic representation of the hypothalamus, as the brain region studied via transcriptomic analyses. **B** Heatmap of top altered hypothalamic genes by the exposure to a high-fat/high-sugar (HFHS) diet during early-life compared to control diet ($q < 0.2$), and the effect of the supplementation of the microbiota-targeted interventions, either the prebiotic FOS + GOS or the bacterial strain *B. longum* APC1472 in 12-week-old female and male mice ($n = 6$/group/sex). Venn diagram of **C** top altered hypothalamic genes by the exposure to a HFHS diet during early-life and **D** restored by the prebiotic FOS + GOS and the putative probiotic *B. longum* APC1472 in 12-week-old female and male mice compared to control diet ($q < 0.2$), ($n = 6$/group/sex). Targeted enrichment for hypothalamic genes altered ($q < 0.2$) by the exposure to a HFHS diet on early-life in 12-week-old **E** female and **F** male mice ($n = 6$/group/sex). Targeted enrichment for hypothalamic genes altered by the exposure to a HFHS diet on early-life and restored by the supplementation with the prebiotic FOS + GOS in 12-week-old **G** female and **H** male mice ($n = 6$/group/sex, $q < 0.2$). Targeted enrichment for hypothalamic genes altered by the exposure to a HFHS diet on early-life and restored by the supplementation with the bacterial strain *B. longum* APC1472 in 12-week-old **I** female and **J** male mice ($n = 6$/group/sex, $q < 0.2$). **A–J** Heatmap, venn diagrams and enrichment analysis were performed on centered log-ratio (CLR)-transformed values of altered and/or restored hypothalamic genes of 12-week-old female and male ($n = 6$/group/sex).

**B–D** Pairwise comparisons were performed using two-sided *t*-tests between the two dietary interventions within Vehicle-treated animals, and between HFHS-exposed animals, including HFHS Vehicle-treated animals and each microbiota intervention (prebiotic FOS + GOS and putative probiotic *B. longum* APC1472) for *P*-adjusted ($q$) values using Benjamini-Hochberg correction for multiple comparison ($n = 6$/group/sex). Restoration effects of microbiota-targeted interventions were evaluated when both diet and intervention effects were significant and the effect sizes showed opposing directions (†$q < 0.2$, ††$q < 0.1$, #$q < 0.05$, ##$q < 0.01$, ###$q < 0.001$ for diet effect; †$q < 0.2$, ††$q < 0.1$, *$q < 0.05$, **$q < 0.01$, *\*$q < 0.001$ in green for restoration effects of FOS + GOS, and, orange of *B. longum* APC1472). **E–J** Pathway enrichment was performed using hypergeometric testing on a priori selected Gene Ontology relevant terms. Input genes were selected based on the statistical analysis described above for genes altered by the diet (**E**, **F**) and restored by the microbiota-targeted interventions (**G–J**) ($q < 0.2$). Data are presented as log2-fold enrichment where gray circles highlight significantly enriched pathways ($p < 0.05$). For sex effect and interactions effects and additional detailed statistical analysis, see https://github.com/CristinaCuesta-Marti/earlylifeHFHS_feeding_microbiota_2025/tree/outputs/overall_stats_lm. Mouse brain slice illustration in **A** was obtained from SciDraw (https://scidraw.io/drawing/646, by Roberta Schellino, CC BY 4.0, DOI: 10.5281/zenodo.10390020) and was adapted using Affinity Designer 2. Schematic diagrams in **D–J** were created with Affinity Designer 2. F females, M males.

greater restorative effects observed in adult females supplemented with *B. longum* APC1472 (Fig. 3D). Analysis of commonly restored genes revealed that FOS + GOS supplementation specifically restored *Hcar1* expression in both male and female mice, encoding a receptor involved in metabolic regulation and anti-inflammatory responses. *B. longum* APC1472 commonly restored *Fen1* and *Piezo2* across both sexes, genes associated with DNA repair mechanisms and mechanosensation, respectively. These findings indicate few shared molecular pathways through which these microbiota-targeted interventions ameliorate HFHS-induced hypothalamic transcriptomic alterations independent of sex (Supplementary Data 3).

We then performed targeted gene enrichment analysis, using Gene Ontology (GO), to determine the functional pathways of genes altered by early-life HFHS exposure[95,96]. Interestingly, pathways critical for energy and metabolic homeostasis as well as neuroendocrine function were significantly altered in adult females exposed to the early-life HFHS (Fig. 3E). These included carbohydrate, lipid, amino acid, glycerophospholipid and pyruvate metabolic processes, along with regulation of cellular response to insulin stimulus, cell surface receptor signaling pathway, and Notch signaling pathway (Fig. 3E). Adult females exposed to early-life HFHS also had altered pathways involved in the response to nutrients and environmental cues, and cellular response to nitrogen compounds (Fig. 3E). Neurogenesis and glutamatergic synapse pathways, which are key for neurodevelopment and synaptic regulation, were also enriched in adult female mice exposed to early-life HFHS (Fig. 3E). Early-life HFHS exposed adult female mice also had altered pathways involved in genetic and molecular regulation, including regulation of RNA splicing, peptidyl amino acid modification, and pyrimidine metabolism (Fig. 3E). Adult males exposed to early-life HFHS showed altered pathways key for energy, metabolic, and neuroendocrine function, including glucose, carbohydrate, amino acid, lipid, organic acid, and pyruvate metabolic processes, as well as steroid and sterol biosynthetic processes (Fig. 3F). Pathways key for the response to nutrients and environmental cues, including response to nutrients, response to peptidoglycan and cellular response to nitrogen compounds were also enriched in adult males exposed to the early-life HFHS (Fig. 3F). Adult males also had altered pathways crucial for neurodevelopment as well as genetic and molecular regulation, such as neurogenesis, neuron development, and nucleobase-containing compound and purine metabolic processes (Fig. 3F).

Next, we assessed functional gene pathways restored by supplementation with each of the microbiota-targeted interventions. We

identified common gene pathways restored by the interventions in both male and female mice, while others were specific to each intervention and/or biological sex. Supplementation with the prebiotic FOS + GOS was able to restore some of the key pathways for energy, metabolic, and neuroendocrine function, including glycerophospholipid, lipid, amino acid, and L-amino acid metabolic processes in adult females exposed to early-life HFHS diet (Fig. 3G). FOS + GOS also restored other pathways in the adult females, including the response to nutrient, nucleobase-containing compound and pyrimidine metabolic processes and cell surface receptor signaling (Fig. 3G). In males exposed to early-life HFHS, FOS + GOS administration restored genes involved in pathways key for energy, metabolic and neuroendocrine function, including carbohydrate and L-amino acid metabolic processes and steroid and sterol biosynthetic processes (Fig. 3H). In addition, pathways related to environmental sensing, including response to nutrients, response to peptidoglycan, and cellular response to nitrogen compounds as well as nucleobase-containing compound metabolic processes, were enriched in adult male mice supplemented with the prebiotic FOS + GOS (Fig. 3H).

Supplementation with the putative probiotic *B. longum* APC1472 in adult females exposed to early-life HFHS significantly restored genes involved in key pathways for energy, metabolic and neuroendocrine function, including glycerophospholipid, tryptophan and nicotinamide nucleotide metabolic processes, as well as cellular response to nitrogen compounds, cell surface receptor signaling, and Notch signaling (Fig. 3I). An enrichment in pathways involved in neurodevelopment as well as genetic and molecular regulation, including neurogenesis, regulation of RNA splicing and alternative mRNA splicing, peptidyl amino acid modification and pyrimidine metabolism was observed in early-life HFHS exposed adult females supplemented with *B. longum* APC1472 (Fig. 3I). In adult males exposed to early-life HFHS, *B. longum* APC1472 administration restored genes involved in energy, metabolic, and neuroendocrine function, including the pathways carbohydrate, amino acid, lipid, monocarboxylic acid, pyruvate, tryptophan, and nicotinamide nucleotide metabolic processes (Fig. 3J). An enrichment in the response to peptidoglycan, alternative mRNA splicing, neurogenesis and purine metabolic processes was also identified in early-life HFHS-exposed adult males supplemented with the *B. longum* APC1472 (Fig. 3J).

To identify pathways of interest linking circulating metabolites and hypothalamic gene expression in gut-brain communication, we performed integrated pathway analyses using the previously reported circulating metabolites and hypothalamic genes that were significantly

altered by early-life HFHS diet exposure and those restored by each microbiota-targeted intervention ($q < 0.2$) in 12-week-old female and male mice (Supplementary Fig. S8). Integrative analysis revealed that early-life HFHS diet altered developmental programming and signaling pathways in adult females, with mTOR signaling, axon guidance, and glutamatergic synapse, as well as FoxO, ErbB, PI3K-Akt, and EGFR tyrosine kinase inhibitor resistance, showing the highest statistical significance (FDR < 0.05). Additional pathways showing nominal enrichment (*p-value < 0.05*) included insulin signaling, as well as pyruvate, glycerophospholipid, and tryptophan metabolism in adult females (Supplementary Fig. S8A). In adult males, integrative analysis revealed that early-life HFHS diet altered energy homeostasis and signaling pathways, including the AMPK signaling pathway, insulin resistance and cAMP signaling (FDR < 0.1), as well as focal adhesion, oxytocin signaling, insulin signaling, and beta-alanine metabolism (*p-value < 0.05*). Additional pathways showed trends toward enrichment, including mTOR signaling, and PI3K-Akt signaling, along with metabolic pathways such as cholesterol and glycerophospholipid metabolism (Supplementary Fig. S8B).

While integrative analysis of pathways restored by the microbiota-targeted interventions did not yield any FDR-significant results in either sex, several nominally significant pathways (*p-value < 0.05*) were observed, highlighting potential effects on developmental, signaling, and metabolic processes. Integrative analysis revealed that FOS + GOS administration restored glycerophospholipid metabolism and pyrimidine metabolism pathways in adult females, with arginine and proline metabolism also showing restoration, consistent with the findings in the blood metabolome (Supplementary Fig. S8C). In adult males receiving FOS + GOS, the pentose phosphate pathway demonstrated nominal enrichment, while additional pathways, including fatty acid metabolism, tryptophan metabolism, and bile acid biosynthesis, showed nominal trends (Supplementary Fig. S8D). Integrative analysis in adult females revealed that *B. longum* APC1472 restored (nominal *p*-value enrichment) developmental programming pathways, with DNA replication showing the highest statistical significance, followed by ErbB signaling and Hedgehog signaling, as well as metabolic pathways including pantothenate and CoA biosynthesis, consistent with the observed HFHS-induced alterations in the blood metabolome. The energy homeostasis pathways, mTOR signaling and AMPK signaling, showed trends toward restoration (Supplementary Fig. S8E). In adult males, *B. longum* APC1472 administration mainly restored pathways related to neuroendocrine function, such as growth hormone synthesis, secretion, and action pathway, alongside trends in metabolic homeostasis pathways including glucagon signaling and AMPK signaling, as well as neurotransmitter pathways, particularly dopaminergic synapse (Supplementary Fig. S8F).

These results demonstrate the enduring effect of exposure of an early-life HFHS diet on the hypothalamic transcriptomic profile of genes involved in metabolic, cellular, and immune processes in adulthood in a sex-specific manner, which can be ameliorated by a microbiota-targeted interventions. Multi-omics integrative analysis further revealed sex-specific alterations and restoration of pathways linking circulating metabolites to hypothalamic transcriptomics.

### Enduring molecular impact of an early-life HFHS diet on POMC neurons and GHSR and LEPR peptidergic receptors in the arcuate nucleus of the hypothalamus of adult male and female mice

Using hypothalamic transcriptomic analysis via RNA sequencing, we identified non-significant downregulation of *Lepr* ($t_{9.25} = 2.10$, $q = 0.064$, $d = 1.21$), of *Pomc* ($t_{9.82} = -0.99$, $q = 0.348$, $d = -0.57$), and *Ghsr* ($t_{6.67} = 1.07$, $q = 0.321$, $d = 0.62$) expression in adult female mice exposed to early-life HFHS diet compared to control mice (Fig. 4A). Using RNAscope in situ hybridization, we identified significant alterations in *Pomc*, *Ghsr*, and *Lepr* expression in the ARC nucleus of adult female and male mice exposed to the early-life HFHS diet (Fig. 4B–E

and Supplementary Figs. S9, S10). Specifically, we observed a reduction in the number of cells expressing *Ghsr*, *Lepr*, and *Pomc* individually (or GHSR+, LEPR+ and POMC+, respectively) in the ARC of adult females exposed to early-life HFHS, while GHSR+ and POMC+ cells were only reduced in adult males exposed to early-life HFHS compared to control (Fig. 4C), which may highlight a specific role for ARC *Lepr* in the feeding behavior observed in females. We also observed that both microbiota-targeted interventions restored the number of cells expressing *Ghsr* and *Pomc* in adult females exposed to early-life HFHS, while supplementation with *B. longum* APC1472 also restored the alteration in *Lepr*-expressing cells (Fig. 4C). Both microbiota-targeted interventions also restored the alteration in the number of cells expressing *Pomc* in adult males exposed to early-life HFHS (Fig. 4C). Moreover, the average *Ghsr* expression in GHSR+ cells in the ARC was down-regulated in adult female and male mice exposed to the early-life HFHS diet, which reversed by the microbiota-targeted interventions only in females (Fig. 4D). Adult females exposed to early-life HFHS also had lower average *Lepr* expression in *Lepr*-expressing cells and lower *Pomc* expression in POMC+ cells in the ARC, the latter of which was restored by the microbiota-targeted interventions in adult females (Fig. 4D).

We also observed a reduction in the number of POMC+ cells co-expressing *Lepr* in the ARC of both female and male mice exposed to the early-life HFHS diet compared to control (Fig. 4E). Both microbiota-targeted interventions restored the number of POMC+ cells co-expressing *Lepr* in the ARC in adult females (Fig. 4E). Early-life HFHS exposure downregulated the average expression of *Lepr* in double LEPR+ and POMC+ cells only in adult females exposed to early-life HFHS and induced a non-significant reduction in *Pomc* expression in these double positive cells in adult females ($p = 0.053$) (Supplementary Fig. S10B). Both microbiota-targeted interventions were able to restore the alterations in average expression of *Lepr* expression in cells co-expressing *Pomc* and *Lepr* in adult female mice (Supplementary Fig. S10B). We also found that early-life HFHS diet reduced the number of double GHSR+ and LEPR+ cells in the ARC of adult female and male mice, and only *B. longum* APC1472 was able to restore the alteration induced in females (Fig. 4E). Adult female and male mice exposed to the early-life HFHS diet had lower average *Ghsr* expression in double GHSR+ and LEPR+ cells, and females also showed a reduced *Lepr* expression in these double positive cells (Supplementary Fig. S10B). Both microbiota-targeted interventions restored the alterations in *Ghsr* expression in double GHSR+ and LEPR+ cells only in adult females (Supplementary Fig. S10B).

These findings indicate that early-life exposure to a HFHS diet has long-lasting effects on neuropeptide-expressing ARC neuronal populations, which are crucial for feeding regulation. Supplementation with the microbiota-targeted interventions showed greater restorative effects in females exposed to early-life HFHS compared to males. Together, these data demonstrate a key sex-specific modulating role of the gut microbiota on early-life diet effects and neuronal priming of feeding behavior.

### Enduring molecular impact of an early-life HFHS diet on PNOC GABAergic neurons and *Nod2*-expressing cells in the arcuate nucleus of the hypothalamus of adult male and female mice

Using hypothalamic transcriptomic analysis via RNA sequencing, we identified that exposure to early-life HFHS diet induced non-significant changes in *Pnoc* expression (downregulation: $t_{5.86} = 2.21$, $q = 0.07$, $d = 1.28$) and *Nod2* expression (upregulation: $t_{8.52} = -1.63$, $q = 0.139$, $d = -0.94$) in adult male mice, the latter of which was restored by both microbiota-targeted interventions (Fig. 5A). To further explore the observed findings of these two potential players in the regulation of appetite and feeding[97,98], we next performed RNAscope in situ hybridization of *Pnoc* and *Nod2* in the ARC.

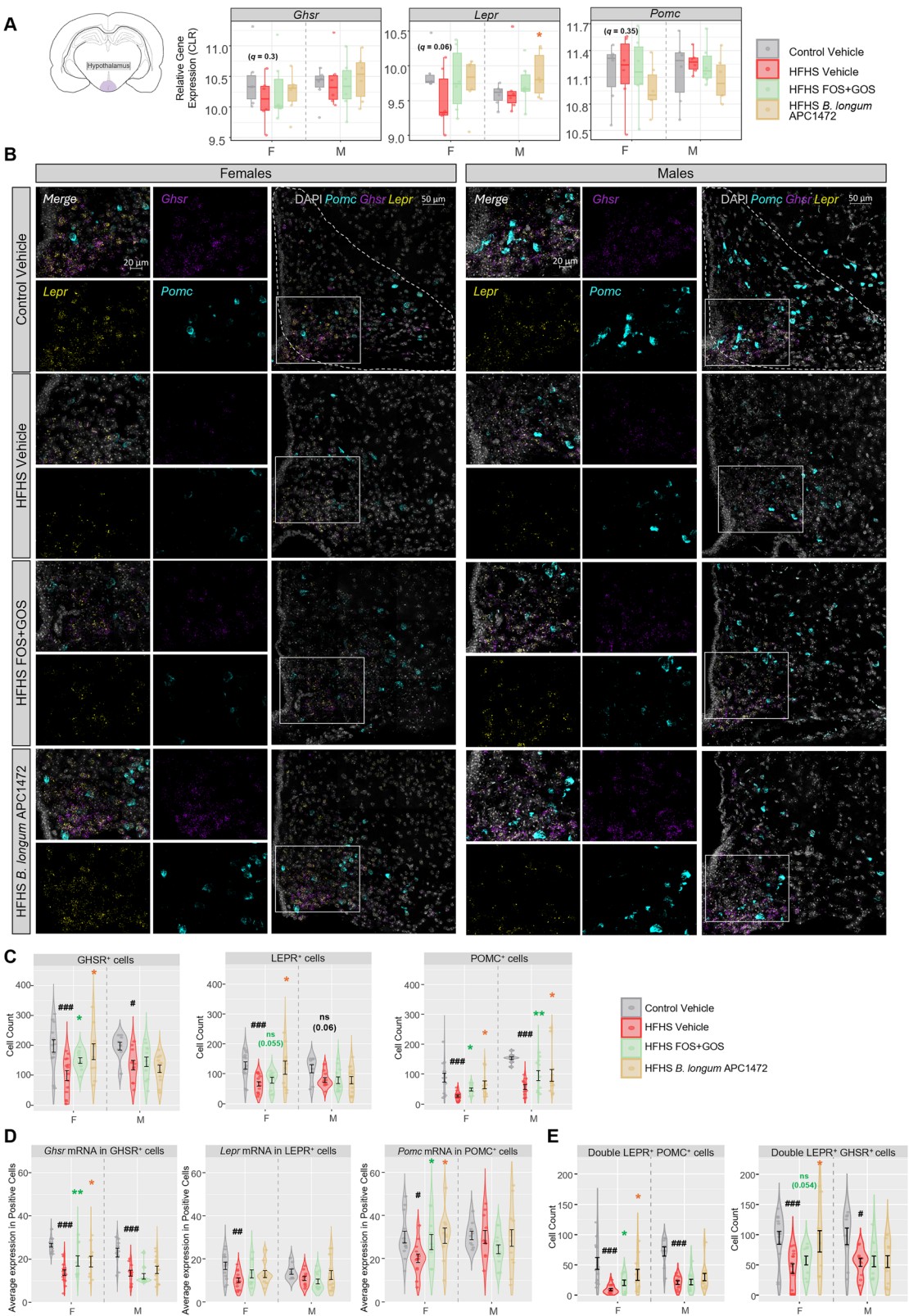

RNAscope analysis revealed significant changes in the number of PNOC[+] and NOD2[+] cells in the ARC nucleus of early-life HFHS-exposed mice (Fig. 5B, C and Supplementary Fig. S11). Furthermore, a significant reduction in the number of *Pnoc*-expressing cells was observed in the ARC of adult female and male mice exposed to early-life HFHS, which was restored by both microbiota-targeted interventions in both sexes (Fig. 5C). While both sexes showed a decrease in the number of ARC

cells expressing *Pnoc* (PNOC[+] cells), only male mice had a reduced average expression of *Pnoc* in PNOC[+] cells compared to control, which was restored by microbiota-targeted interventions (Fig. 5C).

The number of ARC cells expressing *Nod2* (NOD2[+] cells) was reduced in adult males and females exposed to early-life HFHS compared to control, which was restored by both microbiota-targeted interventions in males and by *B. longum* APC1472 in females (Fig. 5C).

**Fig. 4 | Exposure to a high-fat/high-sugar diet during early-life has long-term sex-specific effects on neuronal POMC cells and cells expressing ghrelin and leptin receptor in the arcuate nucleus of the hypothalamus of adult mice.** **A** Gene expression of ghrelin receptor (*Ghsr*), leptin receptor (*Lepr*) and pro-opiomelanocortin (*Pomc*) in the hypothalamus of adult female and male mice using RNA sequencing (12 weeks of age). Boxplots of centered log-ratio (CLR)-transformed values showing all data points (*n* = 6 per group/sex), with the median (central line), 25th and 75th percentiles (box), and whiskers extending to the minimum and maximum values within 1.5× interquartile range. Pairwise comparisons using two-sided t-tests between the two dietary interventions within Vehicle-treated animals, and between high-fat/high-sugar (HFHS)-exposed animals, including HFHS Vehicle-treated animals and each microbiota intervention (prebiotic FOS + GOS and putative probiotic *B. longum* APC1472) for *P*-adjusted values (*q*) using Benjamini–Hochberg correction for multiple comparison (*n* = 6/group/sex). Restoration effects of microbiota-targeted interventions were evaluated when both diet and intervention effects were significant and the effect sizes showed opposing directions (†*q* < 0.2, †† *q* < 0.1, #*q* < 0.05, ##*q* < 0.01, ###*q* < 0.001 for diet effect; †*q* < 0.2, ††*q* ^0.1, *q* < 0.05, **q* < 0.01, ***q* < 0.001 in green for restoration effects of FOS + GOS, and, orange of *B. longum* APC1472). **B** Representative images of the ARC (dotted white line) and magnification with RNAscope for *Pomc* mRNA (cyan), *Ghsr* mRNAs (magenta), and *Lepr* mRNA (yellow) with nuclear DNA counterstained with DAPI (light gray) for 12-week-old female and male mice exposed to either control diet, HFHS diet during early-life, HFHS diet during early-life and supplemented with either the prebiotic FOS + GOS or *B. longum* APC1472. Scale bar of 50 μm for full image and of 20 μm for the magnifications, in the Control Vehicle images for each sex. **C** Number of single positive cells for *Pomc* mRNA, *Ghsr* mRNAs and *Lepr* mRNAs, in the ARC of adult female and male mice (*n* = 4-5 group/sex). **D** Average expression (mRNA or puncta) of *Pomc* mRNA in single POMC$^+$ cells, of *Ghsr* mRNAs in single GHSR$^+$ cells and of *Lepr* mRNAs in single LEPR$^+$ cells in the ARC of adult female and male mice (*n* = 4–5 group/sex). **E** Quantification of number of double positive cells for *Lepr* and *Pomc*, or *Lepr* and *Ghsr* in the ARC of adult female and male mice (*n* = 4–5 group/sex). **C**–**E** Values represented as group average showing the data of the sections of different bregma (*n* = 2–4/mouse; bregma −1.455 to −1.955 mm) per each biological replicate and error bar represented as mean ± SEM. Data analyzed for diet effects by 3-way ANOVA followed by Tukey's two-sided pairwise post-hoc test, between Control Vehicle and HFHS Vehicle (#*p* < 0.05, ##*p* < 0.01, ###*p* < 0.001), and restoration effects by each individual microbiota-targeted intervention (FOS + GOS or *B. longum* APC1472) analyzed by planned orthogonal contrast (pairwise comparison of Control Vehicle–HFHS Vehicle, followed by comparison of HFHS Vehicle–HFHS FOS + GOS or HFHS Vehicle–HFHS *B. longum* APC1472) (*\**p* < 0.05, ***p* < 0.01, ****p* < 0.001, in green for restoration effects of FOS + GOS and orange of *B. longum* APC1472). For sex effect and interaction effects and additional detailed statistical analysis, see https://github.com/CristinaCuesta-Marti/earlylifeHFHS_feeding_microbiota_2025/tree/outputs/overall_stats_lm. Precise sample sizes (*n* numbers) per condition and outcome are detailed in Supplementary Data 4. Mouse brain slice illustration in **A** was obtained from SciDraw (https://scidraw.io/drawing/646, by Roberta Schellino, CC BY 4.0, DOI: 10.5281/zenodo.10390020) and was adapted using Affinity Designer 2. F females, M males.

The average expression of *Nod2* in NOD2$^+$ cells in the ARC was reduced in male mice that were exposed to the early-life HFHS diet compared to control, which was restored by *B. longum* APC1472 (Fig. 5C).

Here, we identified that GABAergic PNOC$^{ARC}$ cells (PNOC$^+$ cells in the ARC) can co-express *Nod2* (Fig. 5B, C and Supplementary Fig. S11). We found that early-life HFHS diet reduced the number of PNOC$^+$ cells co-expressing *Nod2* in the ARC of adult female and male mice (Fig. 5C). These effects were restored by the supplementation with the prebiotic FOS + GOS and *B. longum* APC1472 in both sexes (Fig. 5C). Male but not female mice exposed to the early-life HFHS diet exhibited reduced average *Pnoc* expression in PNOC$^+$ cells co-expressing *Nod2*, which was restored by the microbiota-targeted interventions (Fig. 5C). No differences were found in average *Nod2* expression in PNOC$^+$ cells co-expressing *Nod2* in the ARC of female or male mice (Fig. 5C).

We also found that early-life HFHS diet exposure reduced the total number of *Nod2*-expressing cells in the ARC alongside the third ventricle in adult male mice, which was restored by the microbiota-targeted interventions (Fig. 5B, D). The early-life HFHS diet did not alter the average *Nod2* expression in NOD2$^+$ cells alongside the third ventricle in adult female or male mice (Fig. 5D). Together, these findings suggest that exposure to early-life HFHS diet has enduring effects on GABAergic neurons expressing *Pnoc* and on *Nod2*-expressing cells in the ARC, which can be ameliorated by microbiota-targeted interventions.

## Discussion

Our study revealed that exposure to a HFHS diet during early-life has enduring sex-specific effects on feeding behaviors in adulthood (Fig. 6). These effects occur even when animals are later maintained on a control diet and body weight has normalized, suggesting that the observed long-term molecular and behavioral effects of early-life HFHS diet are independent of body weight differences or metabolic changes. These behavioral changes are associated with sex-specific alterations in serum metabolites and gene expression in the hypothalamus of adult mice. We provide evidence that the detrimental effects of an early-life HFHS exposure can be attenuated by supplementation with two microbiota-targeted interventions (a prebiotic combination, FOS + GOS, and a putative probiotic strain, the *B. longum* APC1472), via divergent mechanisms and in a sex-specific manner.

Our study revealed distinct sex-specific vulnerabilities to early-life HFHS exposure in feeding behavior and differential sex-specific mechanisms of restoration by the prebiotic FOS + GOS and the putative probiotic *B. longum* APC1472. Adult female mice demonstrated a larger number of altered hypothalamic genes in response to early-life HFHS exposure than males. Additionally, females showed altered *Lepr* expression in the ARC, while in males, early-life HFHS exposure reduced PNOC$^+$ and NOD2$^+$ cells. This is in line with larger changes of the functional capacity of the gut microbiota in adult females compared to males in response to the early-life HFHS diet. However, adult males showed higher number of altered genera and circulating metabolites compared to females. We propose that FOS + GOS administration acted in both sexes through extensive microbiome remodeling. However, the restoration of the functional microbiome capacity (GBMs) and gut microbiota composition was sex-specific in the same line as the sex-specific effects of the early-life HFHS diet. Metabolically, FOS + GOS restored metabolic pathways involved in amino acid and fatty acid metabolism in females, whereas in males, we found restoration of steroid metabolism. The putative probiotic *B. longum* APC1472 had stronger restorative effects than FOS + GOS with regards to the early-life HFHS-induced alterations in feeding behavior in females and males. In contrast to FOS + GOS, *B. longum* APC1472 did not act through inducing major microbiome compositional shifts, although there were sex-specific persistent restoration effects in *Lachnospiraceae* genus and in circulating tryptophan in females. In adult males, *Christensenellaceae* R-7 group and *Fournierella* genera, as well as the metabolite 4-guanidinobutyric acid, were restored.

Food crumbling, also known as food grinding, is a food manipulation behavior that has previously been described in laboratory rodents[83,99]. Here, we show that the food crumbling behavior in mice exposed to HFHS only in early-life persisted into adulthood, i.e., even after having been on a control diet for 5–6 weeks. Interestingly, both microbiota-targeted interventions reduced food crumbling behavior, particularly in female animals, suggesting that the microbiota (or microbiota-derived metabolites) plays a role in regulating these behaviors. However, to date, the central and peripheral mechanisms, including the neurocircuitry of food crumbling behavior, are incompletely understood. *B. longum* ACP1472 was able to normalize the heightened food preference induced by the early-life HFHS exposure in both male and female adults, while FOS + GOS only restored this

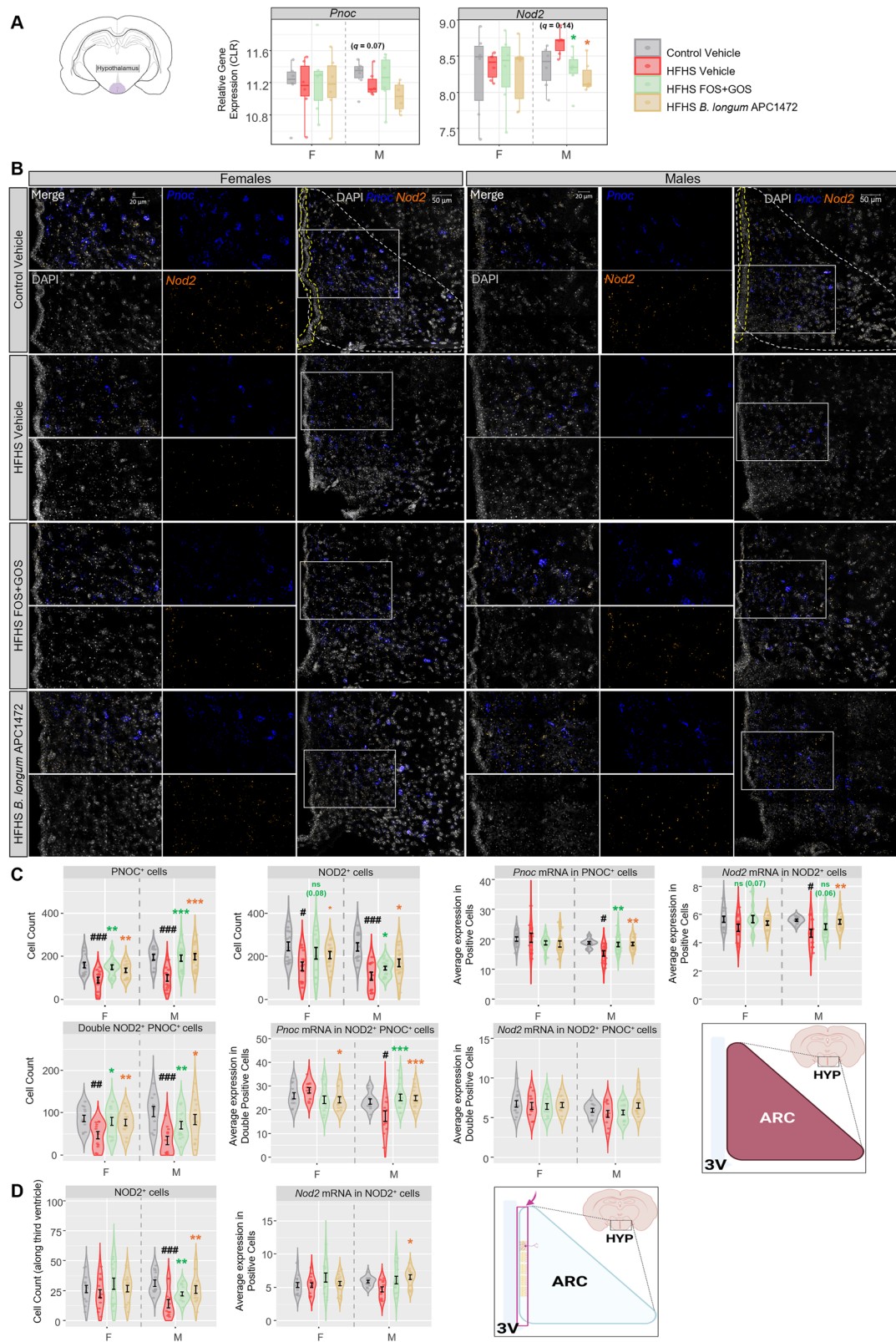

alteration in adult males. Both microbiota-targeted interventions were able to normalize the heightened saccharin preference by the early-life HFHS in adult males. These alterations in saccharin preference in early-life HFHS-exposed adult males may reflect early-life programmed alterations in sweet taste perception, in line with previous findings demonstrating that high-fat and other unhealthy diets can modify taste receptor function and sweet taste sensitivity[10,12,100].

Our data suggest that the reduction in *Bifidobacterium* abundance is critical for the development of disrupted feeding behaviors induced by early-life HFHS diet. In line with this, microbiota-targeted interventions that increased either *Bifidobacterium* abundance, including *B. pseudolongum* (FOS + GOS) or *Bifidobacterium longum* (*B. longum* APC1472), improved feeding behaviors in adulthood. Interestingly, increased *B. longum* abundance in feces following

**Fig. 5 | Exposure to a high-fat/high-sugar diet during early-life has long-term, sex-specific effects on GABAergic PNOC neurons and cells expressing NOD2 receptor in the arcuate nucleus of the hypothalamus of adult female and male mice. A** Gene expression of prepronociceptin (*Pnoc*) and nucleotide-binding oligomerization domain 2 (*Nod2*) in the hypothalamus of female and male adult mice using RNA sequencing (12 weeks of age). Boxplots of centered log-ratio (CLR)-transformed values showing all data points (*n* = 6 per group/sex), with the median (central line), 25th and 75th percentiles (box), and whiskers extending to the minimum and maximum values within 1.5× interquartile range. Pairwise comparisons using two-sided t-tests between the two dietary interventions within Vehicle-treated animals, and between high-fat/high-sugar (HFHS)-exposed animals, including HFHS Vehicle-treated animals and each microbiota intervention (pre-biotic FOS + GOS and putative probiotic *B. longum* APC1472) for *P*-adjusted values (*q*) using Benjamini–Hochberg correction for multiple comparison (*n* = 6/group/sex). Restoration effects of microbiota-targeted interventions were evaluated when both diet and intervention effects were significant and the effect sizes showed opposing directions (†*q* < 0.2, †† *q* < 0.1, #*q* < 0.05, ##*q* < 0.01, ###*q* < 0.001 for diet effect; †*q* < 0.2, ††*q* < 0.1, *q* < 0.05, **q* < 0.01, ***q* < 0.001 in green for restoration effects of FOS + GOS, and, orange of *B. longum* APC1472). **B** Representative images and magnifications of the ARC (dotted white line) with RNAscope for *Pnoc* mRNA (dark blue), and *Nod2* mRNAs (orange), with nuclear DNA counterstained with DAPI (light gray) for 12-week-old female and male mice exposed to either control diet, HFHS diet during early-life, HFHS diet during early-life and supplemented with the prebiotic FOS + GOS or *B. longum* APC1472. Yellow discontinuous line highlights the cells alongside the third ventricle. Scale bar of 50 μm for full image and of 20 μm for the magnifications, in the Control Vehicle images for each sex. **C, D** Number of single positive cells for *Pnoc* mRNA, and *Nod2* mRNAs, and average expression for each gene, as well as number of cells co-expressing *Pnoc* and *Nod2*, and average expression of each gene in the double positive cells in the ARC. **D** Number of single positive cells for *Nod2* mRNAs, and average expression of *Nod2* in the NOD2⁺ cells alongside the third ventricle (3V) of adult female and male mice (*n* = 4–5 group/sex), and schematic representation of the **C** ARC and **D** the area alongside the third ventricle in the hypothalamus, where the analysis have been performed. Values represented as group average showing the data of the sections of different bregma (*n* = 2–4/mouse; bregma −1.455 to −1.955 mm) per each biological replicate and error bar represented as mean ± SEM. Data analyzed for diet effects by 3-way ANOVA followed by Tukey's two-sided pairwise post-hoc test, between Control Vehicle and HFHS Vehicle (#*p* < 0.05, ##*p* < 0.01, ###*p* < 0.001), and restoration effects by each individual microbiota-targeted intervention (FOS + GOS or *B. longum* APC1472) analyzed by planned orthogonal contrast (pairwise comparison of Control Vehicle–HFHS Vehicle, followed by comparison of HFHS Vehicle–HFHS FOS + GOS or HFHS Vehicle–HFHS *B. longum* APC1472) (**p* < 0.05, **p* < 0.01, ***p* < 0.001, in green for restoration effects of FOS + GOS, and orange, of *B. longum* APC1472). For sex effect and interactions effects and additional detailed statistical analysis, see https://github.com/CristinaCuesta-Marti/earlylifeHFHS_feeding_microbiota_2025/tree/outputs/overall_stats_lm. Precise sample sizes (*n* numbers) per condition and outcome are detailed in Supplementary Data 4. Mouse brain slice illustration in **A** was obtained from SciDraw (https://scidraw.io/drawing/646, by Roberta Schellino, CC BY 4.0, DOI: 10.5281/zenodo.10390020) and was adapted using Affinity Designer 2. Schematic diagrams in **C, D** were created with BioRender: Schellekens, H. (2025) https://BioRender.com/w4ypyhz. F females, M males.

---

*B. longum* APC1472 administration, even without affecting overall *Bifidobacterium* abundance, was sufficient to drive behavioral changes and elicit specific microbiome and metabolite changes. The critical role of bifidobacteria in microbiota establishment is well described[67,101] as are their benefits for overall host health[67,102]. For example, *Bifidobacterium* species have been shown to reduce circulating glucose levels, increase the expression of proteins key for insulin signaling, and improve the adipokine profile in a mouse model of diabetes[103].

One potential mechanism underlying microbiome-mediated modulation of feeding behavior is via microbial metabolites and modulation of gut-brain-signaling[104–107]. For example, butyrate administration has been shown to reduce food intake and improve HFD-associated obesity and metabolic disorders[107]. We showed that exposure to early-life HFHS diet had enduring effects on circulating metabolites such as amino acids (β-alanine, arginine, proline), pyruvate, or nicotinamide, which are critical for metabolic and cellular processes as well as neurotransmitter production, including glutamate (GABA precursor), potentially influencing feeding behavior[108]. Interestingly, these dietary-induced changes were attenuated by the microbiota-targeted interventions in a sex- and intervention-specific manner. Specifically, *B. longum* APC1472 restored the enduring alterations in metabolites related to tryptophan metabolism in adult females exposed to early-life HFHS. Our previous in-vitro work demonstrates that *B. longum* APC1472 can produce tryptophan and acetate when grown in bacterial media[109]. Tryptophan-derived metabolites (indoles, serotonin) have been shown to modulate feeding behavior[110–112]. Indoles can induce GLP-1 secretion from L cells[110,111], whereas serotonin has been shown to directly inhibit the activity of AgRP neurons[113] and activate POMC neurons[114] in the hypothalamus, influencing feeding[55,115]. High dietary tryptophan content or exogenous administration has been shown to enhance food consumption by increasing circulating ghrelin[116], serotonin, NPY, gastrin, or growth hormone-insulin-like growth factor (GH-IGF) signaling[117,118]. Early-life HFHS diet significantly reduced all analyzed SCFAs and BCFAs in 5-week-old males but not in females, with FOS + GOS and *B. longum* APC1472 restoring acetate levels. However, these alterations observed at week 5 were not persistent at week 12. Even though the SCFA acetate and butyrate have been shown to regulate feeding behavior through modulation of hypothalamic

neuropeptide expression and neuronal activity[55,107], other metabolites may play a bigger role in the current study.

Peripheral signals, including metabolites and gut hormones, can reach the brain and influence food intake, with the hypothalamus being a key integration center for homeostatic and non-homeostatic feeding[19,21,31]. We observed that early-life HFHS diet induced enduring sex-specific changes of the hypothalamic transcriptome in adult mice. Specifically, female mice had a higher number of genes altered by the early-life HFHS diet than males (>5000 vs. <2000 genes), including key genes involved in energy metabolism and nutrient processing, neurodevelopment and synaptic regulation, response to nutrients and environmental cues, hormonal regulation, as well as gene and molecular regulation and signaling. This aligns with our food behavior assessment in PheCOMP cages, where only females exposed to early-life HFHS diet displayed increased food intake and crumbling. Others have also shown sex differences in the hypothalamic transcriptome of offspring receiving a HFD through the maternal diet. Huang et al. demonstrated that maternal HFD induced an expansion of astrocytes and tanycytes in female offspring, whereas there was an expansion of neurons in males. In this study, offspring were sacrificed at postnatal day 13-15 (PND13-PND15)[39], thus precluding any effect of HFD consumption via solid food and indicating that the observed effects were mediated via maternal milk rather than direct dietary consumption.

As the hypothalamus comprises several nuclei that, together, regulate homeostatic and non-homeostatic feeding, we used RNAscope to assess enduring alterations in the expression of key genes in the ARC. Within the ARC, anorexigenic POMC and orexigenic AgRP neurons have opposing effects on food intake, decreasing or increasing it, respectively[35,36]. We found that early-life exposure to a HFHS diet reduces the number of cells in the ARC expressing *Pomc*, which was normalized by both microbiota-targeted interventions, consistent with higher food intake in HFHS-exposed offspring and normalization in treated animals. A reduction of POMC-expressing neurons has also been shown in offspring from dams fed a HFD[8], which the authors associated with impaired neurogenesis, a pathway that was also identified in our transcriptomic analysis as altered by early-life HFHS diet. In addition, we found that GHSR-expressing cells were reduced in both male and female mice, while LEPR-expressing cells were reduced

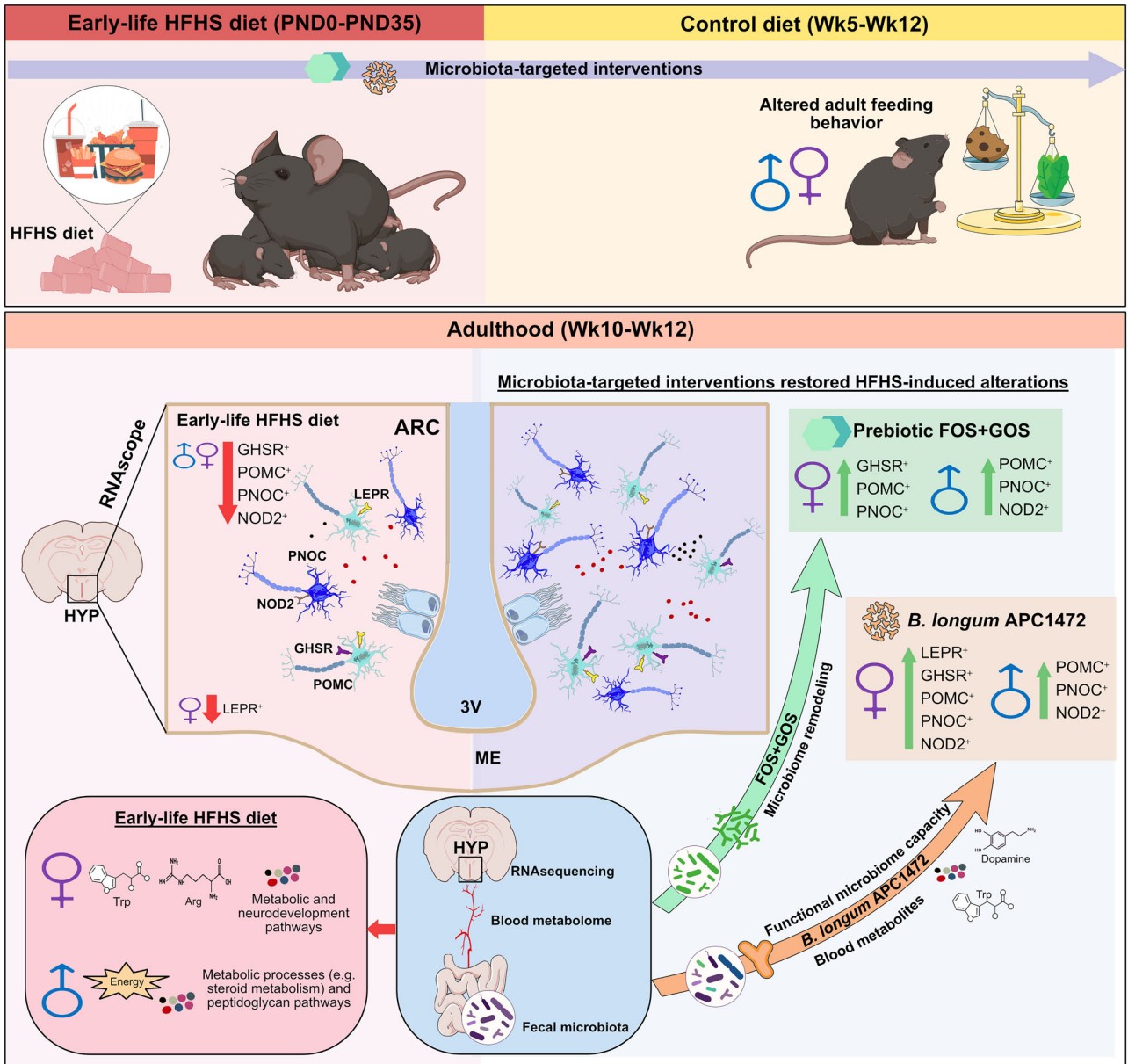

**Fig. 6 | Summary figure *Bifidobacterium longum* and prebiotic interventions restore early-life high-fat/high-sugar diet-induced alterations in feeding behavior in adult mice.** Offspring were exposed to either control or early-life high-fat/high-sugar (HFHS) diet from birth or postnatal day (PND) 0 to PND35 or week (Wk5), to investigate whether exposure to HFHS throughout the entire early life period (lactation and direct dietary exposure) affected adult feeding behavior (Wk10-Wk12) in female and male mice. Microbiota-targeted interventions (the prebiotic combination FOS + GOS or the putative probiotic *Bifidobacterium longum* APC1472) were administered in drinking water throughout the study to investigate their therapeutic potential for the early-life HFHS diet-induced feeding alterations in adult mice. Adult females showed greater hypothalamic vulnerability and alterations in feeding behavior than adult males. Using RNAscope, we found that adult male and female mice exposed to the early-life HFHS diet had reduced numbers of GHSR⁺, POMC⁺, PNOC⁺, NOD2⁺ cells in the arcuate nucleus (ARC) of the hypothalamus. Adult females also showed reduced LEPR⁺ cells in the ARC and disrupted metabolic pathways, including arginine and tryptophan metabolism, while males showed impaired peptidoglycan sensing and steroid metabolism.

Using fecal microbiota analysis, blood metabolomics and integrative analysis (multi-omics of hypothalamic transcriptomics, fecal microbiota analysis and blood metabolomics), we show that microbiota interventions restore these effects via distinct mechanisms. FOS + GOS induced extensive microbiome compositional shifts and sex-specific restoration of gut-brain pathways, while *B. longum* APC1472 induced greater behavioral restoration in both sexes with minimal microbiome compositional changes. We suggest that *B. longum* APC1472 effects are likely occurring through targeted restoration of specific metabolic pathways, such as tryptophan metabolism and energy homeostasis pathways in adult females, and through DOPAC-dopaminergic pathways and metabolic homeostasis (AMPK signaling, glucagon signaling) in adult males. FOS, fructo-oligosaccharides; GOS, galacto-oligosaccharides; ME, median eminence; Trp, tryptophan. Schematic diagrams were obtained from BioRender (Schellekens, H. (2025) https://BioRender.com/w4ypyhz) and SMART Servier Medical Art (https://smart.servier.com, licensed under CC BY 4.0 (https://creativecommons.org/licenses/by/4.0/), and were adapted using Affinity Designer 2.

only in females, exposed to early-life HFHS diet, potentially implying reduced sensitivity to peripheral signals reflecting nutritional status and energy balance[119–122]. Interestingly, microbiota-targeted interventions had a greater restoration effect on the reduction of GHSR-, LEPR-,

and POMC-expressing cells in females, which could explain the improvement in the feeding behavior in these adult female mice.

Recently, a population of ARC neurons has been implicated in the regulation of food intake. Activation (either optogenetically or using

3-day exposure to HFD) of these GABAergic PNOC^ARC neurons inhibits POMC neurons, thereby increasing food intake[97,123]. Here, we show that exposure to an early-life HFHS diet reduced the number of the GABAergic PNOC neurons in the ARC in both adult female and male mice, which, according to the above paradigm, might imply less inhibition of POMC neurons and thereby reduced food intake. However, we also see reduced numbers of POMC neurons, implying that the ratio of PNOC and POMC neurons, as well as POMC sensitivity to peripheral and central inhibitory stimuli, is paramount to understanding the overall effect on food intake[97,123]. Both microbiota-targeted interventions restored the enduring alterations induced by the early-life HFHS diet in the number of PNOC^ARC neurons, providing evidence for microbiota modulation of this neuronal subpopulation. To date, the physiology of PNOC^ARC neurons is not well understood. We found that a significant number of PNOC^ARC neurons expressed NOD2, a cytosolic receptor for peptidoglycans, an important component of the bacterial cell wall, suggesting that these cells play an important role in bacterial sensing. Previously, this receptor has been shown to be expressed in antigen-presenting cells, including microglia, and epithelial cells, where it regulates inflammatory processes, i.e., through activation of NF-kB[124,125]. Interestingly, we found that the "response to peptidoglycan" was the top differentially expressed pathway in the hypothalamus between control adult male mice and those exposed to the early-life HFHS diet. Moreover, we show that both microbiota-targeted interventions restored this pathway, as well as the reduced number of NOD2^+ and PNOC^+ cells in both sexes in the ARC. In line with our findings, another study has demonstrated NOD2 expression in GABAergic neurons co-expressing Vgat^+ and NPY^+ in the ARC but not in non-GABAergic POMC neurons[98]. Absence of *Nod2* expression in these GABAergic neurons delayed satiety, increased food intake and body weight gain, and altered nesting behavior, a stereotypic behavior in older female mice, but not in males[98]. Other studies have shown that whole body depletion of NOD2 exacerbates metabolic inflammation, insulin resistance and overall metabolic dysfunction in an obesogenic environment, suggesting that impaired NOD2 signaling increases food intake, consistent with our findings.

In addition to NOD2^+ cells in the whole ARC, we also found a reduction of NOD2^+ cells lining the third ventricle in the early-life HFHS-exposed adult male mice compared to controls. In this region, non-neuronal cells, including tanycytes and ependymal cells, are located. Tanycytes are specialized cells that can differentiate into orexigenic or anorexigenic neurons, and impact on feeding regulation[126]. They are also involved in nutrient sensing (particularly amino acids, arginine, lysine, and alanine)[127] and adult hypothalamic neurogenesis[126]. This is particularly noteworthy as blood metabolomic analysis revealed arginine metabolism as one of the main pathways impaired by early-life HFHS diet. Previous studies have shown that the number of tanycytic processes and end feet in the ARC and in the median eminence is reduced in offspring from dams receiving HFD during pregnancy[42]. This was associated with increased blood-brain barrier permeability[42] and suggests that increased sensing of bacterial and host metabolites by NOD2-expressing tanycytes may impair their regulatory function.

To conclude, we have shown that early-life exposure to a HFHS diet has enduring effects on adult feeding behavior and induces metabolic and transcriptomic changes in key hypothalamic pathways involved in the regulation of feeding. Our findings reveal that these effects manifest through distinct sex-specific mechanisms: females demonstrate broader hypothalamic vulnerability with a decreased number of POMC neurons and decreased expression of multiple feeding-regulating receptors and neuropeptides (*Ghsr*, *Lepr*, and *Pomc*), while males primarily show alterations in *Pnoc* and *Nod2* expression with selective peptidoglycan sensing disruption. The differential protective mechanisms of FOS + GOS (primarily through microbiome compositional shifts) versus *B. longum* APC1472 (via more

targeted metabolic pathway restoration) act in sex-specific manners, suggesting that therapeutic interventions for early-life dietary programming may require sex-tailored approaches. These findings underscore the critical importance of dietary environments in early-life for shaping food intake and feeding behaviors. This predisposition may heighten susceptibility to metabolic disorders, such as obesity, especially upon re-exposure to calorie-dense, unhealthy diets prevalent in modern societies. We demonstrate that two microbiota-targeted interventions, likely via different mechanisms, are able to attenuate the pervasive effects of early-life HFHS diet, highlighting their potential as therapeutic strategies to address the global rise in consumption of Western-style diets, especially during pregnancy and early-life.

A limitation of our study design is that we cannot distinguish whether the observed effects of early-life HFHS diet and/or microbiome-targeted treatments occur during the lactation period (through maternal milk) and/or the period when pups consume solid food independently (pre- or post-weaning). This experimental design was a deliberate choice to capture the complete window for early-life dietary programming effects (early-life HFHS diet) and maximize potential beneficial effects (microbiome-targeted treatments). While our microbiota-targeted interventions demonstrated beneficial effects on feeding behavior, hypothalamic transcriptomics, and alterations in the ARC, it should be noted that our continuous intervention design throughout life was designed to assess both protective effects during early-life HFHS exposure and restorative effects in adulthood. This makes it challenging to distinguish whether the observed benefits represent prevention of initial programming defects, active reversal of established developmental alterations, or a combination of both mechanisms. We acknowledge that the terminology used in our statistical analysis (restoration effect) for cases where microbiome-targeted interventions significantly attenuated the early-life HFHS-induced alterations may be interpreted ambiguously with respect to the underlying mechanisms. However, the term "restoration" in this context is intended to describe the directionality of change, rather than to imply a specific mechanism of action. Our findings of altered meal size patterns (Supplementary Fig. S1G) also suggest an involvement of the NTS, and other brain regions involved in food reward. Future studies should examine the impact of early-life HFHS exposure and microbiota-targeted interventions on other brain regions involved in different aspects of feeding to provide a more comprehensive mechanistic understanding and should also investigate more selective exposure periods to HFHS diet and microbiome-targeted interventions.

## Methods
### Animals, diets, and ethical approval
All experiments were conducted in accordance with the European Directive 2010/63/EC, the requirements of the S.I. No 543 of 2012 and were approved by the HPRA (AE19130/P179) and the Animal Experimentation Ethics Committee of University College Cork, Cork, Ireland. Male and female C57BL/6J breeders (Envigo, UK) were mated after 2 weeks of acclimatization in specific pathogen-free (SPF) animal research facilities. All mice were housed in IVC cages (Tecniplast, UK) in a 55% ± 10% humidity- and 21 °C ± 1 °C temperature-controlled room. The holding room was maintained on a 12-h light/dark cycle, and food and water were provided ad libitum, except where specified. Sample size determination was based on a priori power calculations performed with G*Power (v.3.1.9.4) and accounting for litter effects. Both male and female animals were included in the study, and sex-based analyses were performed and reported (see Section "Bioinformatics and statistical analysis"). Dams were mated and individually housed at embryonic day 17. On the day of birth (PND0), dams were randomly assigned to experimental groups using a simple randomization procedure (7 dams/group, aiming to avoid any cage/litter effects). After

birth, the dams' food was switched from chow (Teklad 2918, Inotiv) to either control (10 kcal% fat and no sucrose; D12450K Ssniff Spezialdiäten. GmbH, Germany) or high-fat/high-sugar (HFHS) diet (45 kcal% fat and 17 kcal% sucrose; D12451 modified with reduction of sucrose from 20.17% to 17% and increase of starch from 7% to 10.17% (Product N S9912-E760) Ssniff Spezialdiäten. GmbH, Germany) according to the group (1. Control Vehicle, 2. HFHS Vehicle, 3. HFHS diet + prebiotic combination FOS + GOS, 4. HFHS diet + putative probiotic *B. longum* APC1472) (Fig. 1A). Vehicle consisted of drinking water provided using a system with a filter for solids and ultraviolet light water filters for bacteria (used throughout the study unless otherwise specified). All diets, including chow, were provided ad libitum (except where specified), and full compositions of the control and HFHS diets are available in Supplementary Table S4. After weaning (PND21), offspring were group-housed by sex with their littermates within treatment groups and dams were euthanized two days after weaning (PND23). At PND35, 5-week-old offspring were euthanized depending on sex and litter size to keep 2–3 mice per litter/sex (on average) to reach the target sample size $n = 12$ per group per sex at 12 weeks of age. At PND35, these offspring that had consumed the HFHS diet were switched to the control diet until the end of the study in order to investigate the impact of the HFHS diet during the entire early-life period. Male and female offspring were culled at week 12 ($n = 12$/sex/group, with half of the mice being perfused ($n = 6$/sex/group)). As further described in Section "Microbiota-targeted intervention", the prebiotic combination FOS + GOS and the *B. longum* APC1472 were administered in drinking water to the dams from the birth of offspring until PND21, and to offspring from PND21 until the end of the study.

## Preparation of bacteria for in vivo administration and identity confirmation

**Preparation of bacteria for in vivo administration.** All culturing of *Bifidobacterium longum* APC1472 (NCBI accession number: GCA_002833115.1) was performed under anaerobic conditions (PLAS-LABS Simplicity 888 Automatic Atmosphere Chamber with an anaerobic gas mixture ($N_2 = 80\%$, $H_2 = 10\%$, $CO_2 = 10\%$, temperature $= 37\,°C$)). Growth curve of *B. longum* APC1472 was performed manually, in duplicate, using a WPA CO8000 Cell Density Meter, and automatically, in triplicate, using a Cerillo™ Stratus Kinetic Microplate Reader. A glycerol stock of *B. longum* APC1472 was streak-plated on De Man, Rogosa and Sharpe (MRS) medium (BD Difco™ Lactobacilli MRS Broth, BD288130, Fisher Scientific) supplemented with 0.5% (w/v) L-cysteine hydrochloride (C1276, Sigma-Aldrich) (mMRS, from now onwards) and grown for ~48 h until single colonies were visible. A single colony was then inoculated in liquid mMRS medium and incubated until the culture reached early stationary phase (OD600 of 1.1-1.2, ~$1 \times 10^9$ CFU/ml). To prepare the inoculum, the bacterial cell pellet was centrifuged at 4500 rpm (2740 g) for 15 min and resuspended in phosphate-buffered saline (PBS) (P4417, Sigma-Aldrich) twice to wash out any remaining medium. After the washing steps, bacterial cell pellets were resuspended in PBS + 20% (v/v) glycerol (G7893, Sigma-Aldrich), and 1.5 ml aliquots were stored at −80 °C until use. The number of viable bacterial cells in the inoculum was determined via counts of colony-forming units (CFU/ml) for every batch prepared. The purity and identity of each batch of cultures was assessed via Sanger Sequencing of 16S rRNA gene (Eurofins Genomics).

**Confirmation of the purity and identity of the bacterial inoculum.** Total bacterial DNA was isolated using the QIAamp Fast DNA Stool Mini kit (51604, Qiagen) following the manufacturer's instructions with some modifications: bacterial cell pellets with Inhibitex Buffer and zirconia/silica beads (0.125 g of 0.1 mm zirconia/silica beads, 0.125 g of 1.0 mm zirconia/silica beads, and one single 3.5 mm glass bead; 11079101z, 11079110z, and 11079135, respectively; BioSpec) were placed in the FastPrep-24 bead beater (116004500, MP Biomedicals), and

heated in a thermo block (95 °C, 5 min) before continuing according to the manufacturer's instructions. The quality and quantity of the isolated bacterial DNA were assessed using the NanoDrop spectrophotometer (ND-1000, Nanodrop Technologies). DNA was used as a template for PCR targeting the bacterial 16S rRNA gene using the universal bacterial primers 27F2 (5'-AGA GTT TGA TCA TGG CTC A-3') and 1492R3 (5'-TAC GGT TAC CTT GTT ACG ACT T-3') (Integrated DNA Technologies) and the Phusion High-Fidelity PCR Master Mix with HF Buffer (F531, Thermo Fisher Scientific). PCR products were visualized by electrophoresis in 2% agarose gel (Ultra-Pure Agarose, 16500500, Invitrogen and 50X TAE Electrophoresis Buffer, B49, Thermo Scientific), stained with SYBR Safe DNA Gel Stain (533102, Invitrogen), and using loading dye (AM8546G, Invitrogen). PCR products were purified using the QIAquick PCR Purification Kit (28106, Qiagen), following manufacturer's instructions. Purified PCR products were sequenced by Sanger Sequencing of 16S rRNA gene (Eurofins Genomics). Obtained sequences were analyzed in BioEdit v7.2.5/Mega v11.0.10 and identified using Basic Local Alignment Search Tool (BLAST) server at the National Center for Biotechnology Information (NCBI) (v2.14.0). Alignment with the respective source sequence (deposited in NCBI, accession number: GCA_002833115.1) was performed using MultiAlin[128].

## Microbiota-targeted intervention

**In vivo prebiotic administration.** The prebiotic FOS + GOS was prepared by combining FOS Orafti P95 (Healy Group) and GOS90 (Tate & Lyle PLC) (Supplementary Table S5). FOS95 and GOS90 were dissolved under sterile conditions in water to a final concentration of 75 g/L (37.5 g/L of FOS95 and 37.5 g/L of GOS90), which was a dose chosen based on previous literature[129], and filter-sterilized through a 0.45 μm filter. Mice in the prebiotic group received 7.5% FOS + GOS in their drinking water throughout the study (23 days for dams, 12 weeks for offspring). The prebiotic solution, which is readily soluble in water, was prepared weekly under sterile conditions, aliquoted and stored at 4 °C to maintain stability. Water bottles with FOS + GOS were replenished daily to ensure freshness and were fully changed regularly (every 2–3 days maximum) to prevent any potential degradation or contamination.

**In vivo probiotic administration.** Aliquots with the *B. longum* APC1472 inoculum were defrosted daily just prior to the start of the dark phase and diluted in drinking water to administer ~$1.0 \times 10^9$ CFU/animal/day for dams and ~$5.0 \times 10^8$ CFU/animal/day for offspring throughout the experiment (23 days for dams, 12 weeks for offspring). For example, if a cage contained 2 pups, 2x ~ $5.0 \times 10^8$ CFU were diluted in 15 ml of drinking water. The dose administered to adult mice was based on previous pre-clinical and clinical studies in our group[72] and others[130–133], and following evidence-based recommendations[134], which were adapted for the offspring. The putative probiotic was administered for 12 h duration overnight and replaced with regular drinking water every morning.

**Survival *B. longum* APC1472.** The survival of the *B. longum* APC1472 in drinking water was assessed every 4 h over 24 h (Supplementary Fig. S1A). An aliquot of bacteria inoculum was defrosted and diluted in drinking water to the same concentration as administered to the offspring ( ~ $5 \times 10^8$ CFU/animal/day, equivalent to ~$1 \times 10^8$ CFU/mL, dose justified in Section "In vivo probiotic administration"). CFU/ml was determined in duplicates every 4 h during 24 h period (specifically, at time 0, 4, 8, 12, 16, 20, and 24 h). Mice were supplemented with *B. longum* APC1472 for 12 h during the dark phase consistent with survival of this strain (Supplementary Fig. S1A) and aligned to the protocol used in a previous study[72].

## Daily food and drink intake and body weight

Food intake was measured daily in the morning for every cage and averaged by the number of offspring per cage. Energy intake (kcal) was

calculated by multiplying the daily food intake (g) by the caloric content (kcal/g) of the corresponding diet (Supplementary Table S4) that the mice were receiving at the corresponding time. Drink intake was measured every 12 h throughout the experiment and averaged by the number of mice per cage to monitor daily and overnight consumption (Supplementary Fig. S1B). Body weight was individually measured weekly.

### Behavioral tests

**Saccharin preference test.** Female and male mice ($n = 12$/group/sex) undergoing the saccharin preference test at weeks 10 of age, were singly housed in new experimental cages with the presence of two bottles (50 ml Falcons) and pellets of control diet. Mice were habituated to single housing and the presence of two water bottles for three days (3 h, 12 h, 3 h) before the content of one bottle was replaced with a 0.2% saccharin solution (109185, Sigma-Aldrich). Saccharin and water were stored in 50 ml Falcon tubes. The saccharin preference test was carried out using the following modified protocols from previous studies in our group[135–137] and others[138–140]. Accordingly, intake (volume) of water and saccharin solution in both bottles was recorded every 12 h for 48 h in adult female and male mice. The positions of the two bottles were switched after 12 h initially and 24 h afterwards, to reduce any confounding produced by a side or day/nighttime bias. After the last measurement, mice were returned to their original group housing. Preference for saccharin (sweet solution) over water was calculated as follows: (ml of saccharin/[ml of saccharin + ml of water]) × 100.

**Food preference test.** The food preference test was carried out using the following modified protocols from previous studies[10,141–143]. Accordingly, adult female and male mice (10–11 weeks of age, $n = 12$/group/sex) were habituated to single housing for 48 h in a new home cage. After this habituation period, mice were returned to their original group housing. The following day, adult female and male mice were singly housed in a new home cage and presented with control diet in one hopper and HFHS diet in the other, for 48 h. Food intake of control and HFHS diet was recorded every 12 h. Positions of the two hoppers were switched after 12 h initially and 24 h afterwards, to reduce any confounding produced by a side or day/nighttime bias. After the last measurement, mice were returned to their original group housing. Food preference was measured by quantifying the control and palatable diets removed from the respective food hoppers, which in corresponding cases included both the food intake and crumbling of the palatable diet.

**Meal pattern.** Feeding behavior, including meal pattern, food crumbling, and total intake was characterized using PheCOMP cages (Panlab–Harvard Instruments), which allow the precise detection of water and food intake and meal patterns as they continuously record the weight of the two food and two drink hoppers. Adult female and male mice (10–11 weeks of age, $n = 12$/group/sex) were habituated to single housing in the PheCOMP cage for 3 h. After this habituation period, mice were returned to their original group housing. On the experimental day, adult female and male mice (10–11 weeks of age) were placed (singly housed) in the PheCOMP cages with drinking water available ad libitum (two hoppers were used for drinking water to minimize side bias) and HFHS diet in one hopper. Meal pattern was recorded every 5 s for 12 h during the light phase.

Exact feeding pattern was analyzed and determined using the COMPULSE software v1.1 (Panlab–Harvard Instruments) as follows: a meal needed to be larger than 0.2 g and consumed for longer than 10 s, and two distinct meals were recorded when separated by an interval ≥10 min[144–146]. Parameters characterizing feeding patterns, such as relative total food intake, ingestion rate, number of meals, average meal size and duration, interval between meals, eating rate, first meal

size, and latency of first meal, were obtained (data only shown for total intake, food crumbled, eating rate, first meal size, and average meal size). Importantly, while food crumbling and total food intake were measured separately, meal pattern analysis captured the temporal distribution of all food hopper removal events, encompassing both eating and crumbling behaviors.

**Locomotor activity.** Locomotor activity was analyzed using the two-dimensional infrared frames in the PheCOMP cages (Panlab–Harvard Instruments). Adult female and male mice (10–11 weeks of age, $n = 12$/group/sex) were habituated to single housing for 3 h. After habituation, mice were returned to their original group housing. On the experimental day, adult female and male mice (10–11 weeks of age) were placed (singly housed) in the experimental PheCOMP cages with drinking water available ad libitum (two hoppers were used for drinking water to minimize side bias) and HFHS diet in one hopper, for 12 h during the light phase. Locomotor activity was analyzed using the ActiTrack software v2.7.13 (Panlab-Harvard Instruments) with a speed threshold between 2 cm/s and 5 cm/s and a maximum speed at 10 cm/s.

### Intraperitoneal glucose tolerance test

Intraperitoneal glucose tolerance (IpGTT) was assessed on 11-week-old female and male offspring ($n = 12$/group/sex). Briefly, mice were fasted for 6 h during the light phase, with free access to water. An intraperitoneal injection of 2 g/kg glucose ((G8270, Sigma-Aldrich) of body weight dissolved in PBS) was performed, and glucose levels were measured in samples taken from tail vein blood using the ACCU-CHEK glucometer (9221786018, Roche) and strips (7819382, Roche) immediately before and 15, 30, 60, 90, and 120 min after the intraperitoneal injection of glucose, as previously performed[72], with the minor modifications detailed.

### Sacrifice and murine tissue sampling

At the end of the study, 12-week-old female and male ($n = 6$/group/sex) mice were euthanized by decapitation. Trunk blood was collected in tubes (454020, VACUETTE, Greiner Bio-One) containing 500 KIU/ml aprotinin (PI78432, Thermo Fisher Scientific) and 1X cOmplete Protease Inhibitor Cocktail (5892970001, Roche). Blood was then centrifuged at 2000 g for 15 min at 18 °C, following instructions of blood tubes' manufacturers. Plasma was aliquoted and 0.05 N hydrochloric acid (H/1185/PB17, Fisher Scientific) was added into one aliquot for future biochemical analyses of circulating ghrelin. Adipose depots (epididymal/gonadal white adipose tissue, subcutaneous white adipose tissue, mesenteric white adipose tissue and brown adipose tissue), cecum, and liver, were dissected and weighed. Colon and small intestine were dissected and measured. Hypothalamus and striatum (including dorsal and ventral striatum and nucleus accumbens core and shell) were dissected with a forceps (macropunch) and collected. All samples were frozen on dry ice and subsequently stored at −80 °C for further analysis.

Dam fecal samples were collected when their offspring reached weaning age (PND21) ($n = 7$/group). Offspring fecal samples were collected at 5 and at 10 weeks of age, before starting the second round of behavioral tests and the IpGTT ($n = 12$/group/sex per time point).

### Perfusion

12-week-old female and male offspring ($n = 6$/group/sex) were anesthetized with an intraperitoneal injection of Pentobarbital Sodium (Abbeyville Vet Hospital, 90 mg/kg of body weight) and perfused transcardially with PBS followed by 4% w/v paraformaldehyde (PFA) (P/0840/53, Fisher Scientific). Brains and livers were collected and kept in PFA overnight at 4 °C, and then transferred to 25% (w/v) sucrose (S/8600/60, Fisher Scientific) for at least 48 h at 4 °C. Once the brains were saturated, they were snap frozen in isopentane (2-Methylbutane,

294524E, VWR Chemicals BDH), cooled using liquid nitrogen, and stored at −80 °C.

## RNA isolation from the hypothalamus and striatum

Total RNA was extracted from the hypothalamus and striatum ($n = 6$/group/sex) using the High Pure RNA Isolation Kit (11828665001, Roche) with the following modifications from manufacturer's recommendations: instead of the Lysis/Binding Buffer, Qiazol Lysis Reagent (79306, Qiagen) was used together with zirconia/silica beads (~0.125 g of 1.0 mm zirconia/silica beads, and 3–5 2.3 mm zirconia/silica beads, 11079101z and 11079125z, respectively; BioSpec) and, placed 1 min in the FastPrep-24 bead beater (116004500, MP Biomedicals) for disruption and homogenization steps. After the addition of Chloroform (C2432, Sigma-Aldrich), the aqueous phase was used to continue with manufacturer's recommendations. Quality and quantity of the isolated RNA were assessed on a NanoDrop spectrophotometer (ND1000, Nanodrop Technologies) and Agilent Bioanalyser 2100 (5067-1511, Agilent Technologies), and kept at −80 °C until further analysis.

## Transcriptomic sequencing

mRNA from the hypothalamus (~2 μg) was used for the RNA-seq analyses ($n = 6$/group/sex). RNA sample quality control, mRNA library preparation (poly-A enrichment), NovaSeq X Plus (PE150) (6 G raw data per sample) sequencing, and data quality control as well as Fastq-file generation, were performed by Novogene. A reference genome for the obtained sequences was generated using the following reference annotation: *Mus musculus* (organism), GRCm38, UCSC genome browser (GRCm38.p6), Ensemble[147]. Reads were assessed and filtered for quality with Fastqc (v0.12.1) using default parameters. Genes were annotated and counted with Kallisto (v0.48.0)[148] using the default parameters.

## Differential gene expression and gene ontology term enrichment analyses.

Transcriptomic data analysis, after genes were annotated and counted, were performed in R Studio (2023.12.1). Low-expression genes (with counts less than 10 in 75% or more of the samples) were filtered out from the dataset to minimize noise and improve downstream analysis. Differential expression, representations, and gene ontology (GO) term enrichment analyses were performed on centered log-ratio (CLR)-transformed values, as further described in Section "Bioinformatics and statistical analysis ('Omics datasets)". Employing the base R stats phyper implementation of the hypergeometric test on a priori selected Gene Ontology (GO) terms[96] relevant to the objectives and scope of the study design, enrichment analysis was performed on genes altered by the diet (Benjamini-Hochberg correction, P-adjusted ($q$) <0.2), and restored by each microbiota-targeted intervention (Benjamini–Hochberg correction, P-adjusted ($q$) <0.2) on each sex. Figures were generated using ggplot2 (v3.5.2) and patchwork (v1.3.1) in R with these selected genes.

## RNAscope (fluorescent in situ hybridization)

Perfused brains were cut using a cryostat and 20 μm thick section were stored in tissue storage solution (25% glycerol (G7893, Sigma-Aldrich), 25% ethylene glycol (293237, Sigma-Aldrich), 50% PBS (P4417, Sigma-Aldrich), autoclaved for RNAscope) at −20 °C until further processing. We focused on genes expressed by neurons, including receptors for hormones or bacterial peptidoglycan to sense peripheral signals, in the arcuate nucleus that are linked to energy balance and feeding regulation[19,21,97,98]. Thus, we selected probes for POMC, GHSR, LEPR, prepronociceptin (PNOC), and Nucleotide-binding oligomerization domain 2 (NOD2). On the day before the RNAscope assay, sections containing the ARC of the hypothalamus (bregma −1.455 to −1.955 mm) were mounted onto SuperFrost Plus Adhesion Microscape slides (J1800AMNZ, Epredia), dried at room temperature and baked at 60 °C o/n in the ACD HybEZ oven (PN 321710/321720, Advanced Cell Diagnostics) ($n = 4$–5/group/sex). Fluorescent RNAscope (ACD; Advanced Cell Diagnostics Inc., Hayward, CA) was performed to analyze simultaneous *Ghsr* (426141, BioTechne), *Lepr* (402731, BioTechne) and *Pomc* (314081, BioTechne) on one set, and simultaneous *Nod2* (433391, BioTechne) and *Pnoc* (437881, BioTechne), on another set, in the ARC. Incubation steps were performed at 40 °C using the humidified chamber and oven from the HybEZ II System (Advanced Cell Diagnostics). The pre-treatment protocol followed manufacturer's instructions with the following modifications: 7 min incubation in hydrogen peroxide (322381, Advanced Cell Diagnostics), and 7 min incubation with target retrieval (322000, Advanced Cell Diagnostics). The detection protocol followed the manufacturer's protocol for the RNAscope Multiplex Fluorescent v2 kit. For detection of the probes directed towards *Ghsr* and *Nod2* mRNA, Opal650 tyramide (FP1496001KT, Akoya Biosciences), was used diluted 1:2000. For *Lepr* mRNA, Opal570 tyramide (FP1488001KT, Akoya Biosciences) was used diluted 1:2000. For detection of *Pomc* mRNA, Opal520 tyramide (FP1487001KT, Akoya Biosciences) was used diluted 1:3000. The same Opal520 tyramide was used diluted 1:500 for detection of the probe directed towards *Pnoc*.

All the sections were afterwards counterstained with DAPI (1:5,000) (62248, Thermo Fisher Scientific), coverslipped with Pro-Long™ Diamond Antifade Mountant (P36970, Invitrogen) mountant, and stored in the dark at 4 °C, until imaging acquisition. Images were acquired using a laser scanning confocal microscope (LSM 700 inverted, Zeiss), at the Centre for Cellular Imaging at the University of Gothenburg. A Plan-Apochromat 40×/1.3 Oil DIC objective was used, with 3 × 3 tiling and z-stack (7 optical sections) settings, to image the RNAscope in the ARC subregion. The Z-stack images were processed using the maximum intensity projection function in the Zen Black software (Zeiss, version 2012 SP5). Channels were merged and DAPI-identified cells with ≥2 dots/cell were defined as being positive for *Nod2*, ≥3 dots/cell for *Ghsr*, and *Lepr*, ≥4 dots/cell for *Pomc* and *Pnoc* using QuPath software[149] (v0.5.1).

## Monoamine neurotransmitter measurement in striatum using HPLC and electrochemical detection

For their involvement in the non-homeostatic regulation of feeding[16,90,150,151], striatal samples were processed for the measurement of monoamine concentrations, including noradrenaline, dopamine, L-3,4 dihydroxyphenylalanine (L-DOPA), homovanillic acid (HVA), 3,4-Dihydroxyphenylacetic acid (DOPAC), 5-Hydroxytryptamine or serotonin (5-HT), and 5-hydroxyindoleacetic acid (5-HIAA), using reverse-phase high-performance liquid chromatography (HPLC) as previously described[152]. In brief, 500 μL of homogenization buffer, composed of the mobile phase and an internal standard (2 ng/20 μL of N-methyl 5-HT, M1514, Sigma Aldrich), was added to the previously weighed tissues ($n = 6$/group/sex). The samples were then sonicated (Sonopuls HD2070, Bandelin), and centrifuged at 21,910 g for 20 min at 4 °C using a Hettich Mikro 22 R centrifuge (AGB). The supernatant was carefully collected afterwards, and 20 μL was used as the injection volume. The HPLC system (Shimadzu) included an SCL 10-Avp system controller, LC-10AS pump, and SIL-10A autoinjector (kept at 4 °C), along with a CTO-10A column oven set to 30 °C. Detection was performed using a Measurement Cell Sputnik electrochemical detector (EC3000, ClinLab), operating at +0.8 V with a range of ±50 nA. The column used was a Kinetex 2.6 μm C18, 100 Å, 100 × 4.6 mm (Phenomenex), with a flow rate of 0.9 ml/min. The mobile phase was prepared (0.1 mol/L citric acid (c0759, Sigma Aldrich), 0.1 mol/L sodium dihydrogen phosphate monohydrate (102455S, VWR Chemicals BDH), 5.6 mmol/L 1-octane sulfonic acid (1.18307.0100, Sigma Aldrich), and 0.01 mmol/L EDTA disodium salt dihydrate (e4884, Sigma Aldrich)) in MQ water. Finally, HPLC-grade methanol ((9% v/v), m/4059/pb17, Fisher Chemical) was gradually added, and the pH was adjusted to 2.8, using 1 M NaOH (S8045, Sigma Aldrich).

## Analysis of plasma ghrelin, corticosterone, and metabolic signature

A potential impact of the early-life HFHS diet or microbiota-targeted interventions on plasma active ghrelin levels was analyzed by enzyme-linked immunosorbent assay (ELISA), because of its role in the regulation of food intake and feeding behavior[153–155], and its suggested involvement with the gut microbiota[78,156]. Corticosterone levels were also analyzed by ELISA, to assess any potential impact of the early-life HFHS diet or microbiota-targeted interventions on the stress response and hypothalamic-pituitary-adrenal (HPA) axis, which the microbiota has been shown to regulate[74].

Accordingly, active ghrelin levels in plasma were measured using the Rat/Mouse Ghrelin (Active) ELISA kit (EZRGRA-90K, Merck Millipore) according to the manufacturer's instructions ($n = 6$/group/sex). Free corticosterone levels were assayed using the Corticosterone ELISA kit (ADI-901-097, Enzo Life Sciences) according to the manufacturer's instructions, diluting the sample with assay buffer without steroid displacement reagent ($n = 6$/group/sex). Plasma levels of triglycerides, glucose, insulin and cholesterol, were analyzed by chemoluminiscence in the Cobas Integra 400 (Roche Diagnostics) at the Estación Biológica de Doñana (EBD-CSIC, Seville, Spain) ($n = 6$/group/sex).

## Biochemical analysis of metabolic signature in the liver

Liver biochemical composition, including cholesterol, glucose, protein, alanine transaminase (ALT), and aspartate aminotransferase (AST), was analyzed by chemiluminiscence using the Cobas Integra 400 (Roche Diagnostics) at the Estación Biológica de Doñana (EBD-CSIC, Seville, Spain) ($n = 6$/group/sex).

## Untargeted metabolomic analyses in plasma and integrated pathway analysis

Blood metabolomics was conducted by MS-OMICS (Vedbaek, Denmark) ($n = 6$/group/sex). Blood collection, processing, and plasma storage are described in Section "Sacrifice and murine tissue sampling". Semi-polar metabolite profiling (reverse phase) was analyzed with MS-Omics' semi-polar metabolites method, a slightly modified version of the protocol described by Doneanu et al.[157] (Doneanu et al., 720004042en). Samples were extracted using Phree Phospholipid Removal cartridges (Phenomenex) with an equal mixture of acetonitrile and methanol (1:1 vol/vol) containing stable isotope-labeled internal standards, centrifuged ($15,000 \times g$ for 5 min at 4 °C), and dried. Samples were reconstituted in a 9:1 (v/v) mixture of mobile phase eluents A and B containing additional stable isotope-labeled standards, resulting in a total dilution factor of 5.5 prior to injection. A quality control (QC) sample was prepared by pooling ~30 μL aliquots from each plasma sample to create a representative average. The QC was processed alongside the experimental samples and analyzed after every sixth experimental sample throughout the sequence. The pooled QC sample was analyzed 13 times across the analytical sequence to assess technical variation, with acceptance criteria for detected compounds including signal-to-noise ratio >5, precision <20% (relative standard deviation across pooled QCs), QC-to-sample signal ratio between 0.5 and 2, and correlation between signal and QC dilution factor >0.8 to assess matrix effects. No technical replicates of individual samples were performed; technical variation was instead assessed through the pooled QC samples analyzed throughout the sequence. The analysis was carried out in a randomized sample order using a UPLC system (Vanquish, Thermo Fisher Scientific) equipped with a C18 column ($2.1 \times 150$ mm, 1.8 μm, HSS T3) coupled with a high-resolution quadrupole-orbitrap mass spectrometer (Orbitrap Exploris 240 MS, Thermo Fisher Scientific). Mobile phases consisted of 10 mM ammonium formate with 0.1% formic acid in water (A) and in methanol (B), with a 0–90% B gradient over 15 min at 300 μL/min and a 50 μL injection volume. Ionization was achieved with an electrospray ionization interface operated in positive and negative mode under polarity switching, and a QC sample was analyzed in MS/MS fragmentation mode for compound identification. MS acquisition parameters included a scan range of m/z 65–975 with a resolving power of 60,000 and mass tolerance of 3 ppm for compound detection. Data were processed, and peak areas were extracted using Compound Discoverer v3.3 (Thermo Fisher Scientific) and Skyline v23.1[158]. Out of the three levels of annotation for LC-MS/MS data, level 1 (annotations based on accurate mass, MS/MS spectra and known retention time compared against in-house standards analyzed on the same system) and level 2a (based on accurate mass and known retention time compared against in-house standards analyzed on the same system) were selected for further analyses, with the exclusion of non-dietary plant metabolites. The resulting count table identifying level 1 and 2a metabolites was CLR-transformed to account for the inherent compositionality of multi-omic data[159] using the R package vegan[160]. Metabolites that were found to be altered by the early-life exposure to a HFHS diet and restored by the microbiota-targeted interventions (individually by FOS + GOS or *B. longum* APC1472) were mapped to their corresponding Human Metabolome Database identifier and subjected to enrichment analysis using the MetaboAnalyst online pipeline for metabolites altered by the diet (Benjamini-Hochberg correction, $P$-adjusted ($q$) <0.2), and restored by each intervention (Benjamini-Hochberg correction, $q < 0.2$) on each sex[161] with additional controls with False Discovery Rate (FDR), choosing the murine KEGG library as a reference[162].

Integrated pathway analysis was performed incorporating the lists of differentially abundant circulating metabolites and differentially expressed genes in the hypothalamus of 12-week-old female and male mice ($q < 0.2$) using the MetaboAnalyst software V6.0 with default parameter settings on all KEGG pathways[163]. This allowed the identification of potential pathways of interest that were modulated by diet alteration and by restoration through each microbiota-targeted intervention (FOS + GOS or *B. longum* APC1472) via multi-omics investigation[164]. This analysis considers both the enrichment ratio of components of the pathway as well as the centrality in that pathway to generate statistical significance values and pathway impact metrics[163]. For each sex, three separate integrated analyses were performed using differentially expressed genes and abundant metabolites in 12-week-old mice: (1) those altered by dietary intervention, (2) those restored by FOS + GOS administration, and (3) those restored by *B. longum* APC1472 administration.

## Short-chain and branched-chain fatty acid (SCFA and BCFA) analysis preparation and analysis GC-FID operation

Extraction and analysis of the short-chain fatty acids (SCFAs) acetate, butyrate, propionate and valerate, and branched-chain fatty acids (BCFA) isobutyrate and isovalerate, followed a slightly modified version of the protocol described by Lynch et al.[165]. Accordingly, cecal samples (0.2 g) were homogenized in 1 mL acidified water (0.008 mL/L HCl in Milli-Q water) and centrifuged at $16,000 \times g$ for 30 min at 4 °C. The exact mass of each cecum sample was accounted for when calculating concentration of SCFA/BCFA post-GC. The supernatant was transferred to a new 2 mL tube and centrifuged at $16,000 \times g$ for 30 min at 4 °C. This centrifugation step was repeated twice more for a total of three cycles. The final supernatant was sequentially filtered through 5 μm Ultrafree-MC-SV centrifugal filters (Merck Millipore) at $12,000 \times g$ for 10 min, followed by 0.22 μm COSTAR SpinX centrifuge tube filters (Corning) at $12,000 \times g$ for 10 min. The filtrate was divided into two 270 μL aliquots, and 30 μL of 10 mM 2-ethylbutyric acid (internal standard) was added to each aliquot. The mixture was centrifuged at $15,000 \times g$ for 10 min at 4 °C, and 250 μL was transferred into 2 mL amber glass GC vials (Agilent), containing polymer feet inserts (Agilent). A standard curve was prepared prior to running samples on GC, using acetate, propionate, butyrate, and valerate

(0.1–10 mM), and isobutyrate and isovalerate (0.01–1 mM) (all acids were purchased from Sigma-Aldrich).

SCFA and BCFA analysis was performed by gas chromatography with flame ionization detection (GC-FID) using an Agilent 8860 system (Agilent) equipped with a DB-FFAP column (30 m × 0.32 mm ID × 0.25 µm film thickness; Agilent). Samples were injected (1 µL) in splitless mode using an automated sampler. Helium (Air Products) served as the carrier gas at a constant flow rate of 1.3 mL/min. The oven temperature program was: initial temperature 50 °C held for 0.5 min, increased to 140 °C at 10 °C/min and held for 0.5 min, then increased to 240 °C at 20 °C/min and held for 5 min (total run time 20 min). Injector and detector temperatures were maintained at 240 and 300 °C, respectively. Peak integration and quantification were performed using OpenLab software (Agilent). Abundance (µmol/g) was plotted for acetate, propionate, butyrate, valerate, isobutyrate and isovalerate for cecal samples from offspring that were collected at week 5 ($n = 6$–14/sex/group) and 12 ($n = 6$–7/sex/group) of age. Of note, the week 12 female HFHS FOS + GOS group had a reduced sample size ($n = 2$) due to insufficient remaining cecal filtrate for GC-FID analysis. The SCFA and BCFA abundance was calculated as the average of two technical replicates per sample, and outliers were removed as described in Section "Bioinformatics and statistical analysis".

## Fecal microbiome 16S sequencing

Fecal DNA from dams ($n = 7$/group) and 5- and 10-week-old female and male offspring ($n = 12$/group/sex/timepoint) was isolated using the QIAamp Fast DNA Stool Mini kit (51604, Qiagen), following the modified manufacturer's instructions explained in Section "Confirmation of the purity and identity of the bacterial inoculum". The quality and quantity of the isolated DNA were assessed on a NanoDrop spectrophotometer (ND1000, Thermo Scientific) and kept at −20 °C until further analysis. The bacterial composition of the samples was determined through sequencing of the hypervariable V3–V4 region of the bacterial 16S rRNA gene following the protocol outlined in the Illumina 16S Metagenomic Sequencing Library Preparation guide[166]. The universal bacterial primers were selected from those described by Klindworth et al.[167]. PCR amplification of V3–V4 region was performed using the forward primer 5′- TCGTCGGCAGCGTCAGATGTGTATAA-GAGACAGCCTACGGGNGGCWGCAG-3′ and reverse primer 5′-GTCT CGTGGGCTCGGAGATGTGTATAAGAGACAGGACTACHVGGGTATCTA ATCC-3′ (Merck, Sigma Aldrich). Each 25 µl PCR reaction contained 5 ng/µl microbial genomic DNA, 1 µM of each primer, and 12.5 µl 2× KAPA HiFi HotStart ReadyMix (Roche Diagnostics). The PCR conditions were as follows: 5 min at 95 °C; 25 cycles of 30 s at 95 °C, 30 s at 55 °C and 30 s at 72 °C; and 72 °C for 5 min. Purification of amplicons was carried out by Agencourt AMPure XP beads (Beckman Coulter). A subsequent limited-cycle PCR step was performed to add multiplexing indices and Illumina sequencing adapters. Index primers were supplied as part of the Nextera XT Index Kit v2, Set A (15052163, Illumina Inc.). Each 50 µL indexing PCR reaction contained 5 µL amplified DNA, 5 µL Nextera XT Index Primer 1 (Illumina Inc.), 5 µL Nextera XT Index Primer 2 (Illumina Inc.), 25 µL 2× KAPA HiFi HotStart ReadyMix, and 10 µL PCR grade molecular water. The Agencourt AMPure XP beads were used for purification of index PCR products. Amplicons were quantified using the Qubit® dsDNA HS Assay Kit (Life Technologies) and 4200 Tapestation system (Agilent Technologies), and subsequently, normalized, and pooled. Sequencing was carried out by Genewiz Azenta Life Sciences (GENEWIZ) employing 2 × 300 base-pair sequencing on the Illumina MiSeq platform.

### Analysis of microbiome sequencing data. Using FastQC (v0.12.1), raw sequences underwent quality checks and were further processed using the DADA2 pipeline (v1.30.0)[168]. After plotting quality profiles, the following filtering and trimming parameters were chosen to best handle the data: trimLeft = c(17,21), truncLen = c(285,210), maxN = 0, maxEE = c(2,4), truncQ = 2. Error rates were learned, and the core sample inference algorithm was applied to the dereplicated data. Forward and reverse reads were merged, and a sequence table generated, made up of amplicon sequence variants. Since two MiSeq sequencing runs were required to sequence the data, each run was processed through the above DADA2 pipeline separately. Once the sequence tables were produced for runs 1 and 2, they were then merged. Chimeras were removed from the merged sequence table, and taxonomy was assigned using the Silva taxonomy database v138[169]. A taxonomy table was subsequently generated. The sequence table and taxonomy table produced were imported to RStudio (v2024.04.2 (with Build 764)) using R v4.4. A phyloseq object was created with these outputs using the phyloseq package (v1.5.2). Further packages required for data analysis were ggplot2, dplyr (v1.1.4), tidyverse (v2.0.0), ggpubr (v0.6.2), vegan (v2.7.1), adonis2 function from the vegan package, car (v3.1.3), and rstatix (v0.7.2). Raw counts were normalized by transforming them to relative abundances using the transform_sample_counts function.

Microbial functional potential was inferred from 16S rRNA gene sequences of microbiome samples using PICRUSt2 v2.5.2 applied to the processed DADA2 output, generating KEGG orthologue predictions[94]. Additionally, we filtered the pathways identified by PICRUSt2 to retain pathways involved in the metabolism of metabolites that were significantly altered by the early-life HFHS diet or significantly restored by each microbiota-targeted intervention (FOS + GOS or *B. longum* ACP1472) ($q < 0.2$). This approach linked microbial functional potential to observed metabolic changes through biologically relevant pathways, mechanistic insights into the gut-brain-metabolome axis. The abundances of pathways were CLR-transformed before conducting statistical analyses (see Section "Omics datasets"). A pseudocount of 2/3 was used to replace zero values. In addition, the R package omixerRpm (v0.3.3) was employed to compute Gut-Brain Modules (GBMs) from the functional annotation data[170].

## Bioinformatics and statistical analysis

All data, including 16S rRNA, transcriptomic, metabolomic, and other molecular and behavioral data, were analyzed using R Studio (v2023.12.1). In outputs where multiple technical replicates were performed (for instance, circulating active ghrelin or HPLC analysis), the mean per individual was used with standard error of the mean (SEM). Statistical analyses were assessed using linear models using as implemented in base R, and linear mixed-effects models using the package lme4 (v1.1.37)[171]. Functions and scripts are available online (https://github.com/thomazbastiaanssen/Tjazi). The alterations by the early-life exposure to the HFHS diet were analyzed by 3-way ANOVA followed by Tukey's two-sided pairwise post-hoc test, between Control Vehicle (or also referred to as Control diet group) and HFHS Vehicle (unless otherwise noted, #$p < 0.05$, ##$p < 0.01$, ###$p < 0.001$). The restoration by each individual microbiota-targeted intervention (FOS + GOS or *B. longum* APC1472) of features altered by the early-life exposure to the HFHS diet was assessed using orthogonal contrasts in a generalized linear model, weighing either "Control Vehicle" and "HFHS FOS + GOS" or "Control Vehicle" and "HFHS *B. longum* APC1472" groups against the "HFHS Vehicle" group (unless otherwise noted, *$p < 0.05$, **$p < 0.01$, ***$p < 0.001$ for restoration effects of each intervention). Complete statistical analyses, including main effects (sex, diet, intervention and timepoint for longitudinal data) and interaction effects (before and after removing statistically significant outliers), are available at https://github.com/CristinaCuesta-Marti/earlylifeHFHShttps://github.com/CristinaCuesta-Marti/earlylifeHFHS_feeding_microbiota_2025/tree/outputs/overall_stats_lm.

Orthogonal contrasts were employed to test specific hypotheses regarding restoration effects, providing greater statistical sensitivity

for these pre-planned comparisons compared to ANOVA approaches with multiple comparison adjustments. Therefore, restoration effects may reach statistical significance even when the pairwise diet effects (Control Vehicle vs HFHS Vehicle) assessed by Tukey's post-hoc tests do not reach significance.

Additional random effects (including litter and cohort effects) were assessed using linear mixed-effects models using the lmer function in R and results of these statistical analyses are available at https://github.com/CristinaCuesta-Marti/earlylifeHFHS_feeding_microbiota_2025/tree/outputs/overall_stats_lmer. Technical outliers were analyzed using the Grubbs test in R Studio with the "outliers" package in R (v0.15)[172] and a maximum of one outlier per group/sex (if any) was excluded. Data distribution was assumed to be normal, but this was not formally tested. Analyses were conducted in a manner blinded to the experimenter.

**'Omics datasets.** 'Omics data analysis, which includes metabolomics, transcriptomics and 16S metagenomics, was performed on CLR-transformed counts[173]. Pairwise comparisons were performed using t-tests between the two dietary interventions within Vehicle-treated animals, and between HFHS-exposed animals, including HFHS Vehicle-treated animals and each microbiota intervention (prebiotic FOS + GOS and putative probiotic *B. longum* APC1472) for *P*-adjusted (*q*) values using Benjamini–Hochberg correction (unless otherwise noted, with a *q*-value of 0.2 as a threshold) for multiple comparisons using the rstatix package in R (v0.7.2). Restoration effects of microbiota-targeted interventions were evaluated when both diet and intervention effects were significant, and the effect sizes showed opposing directions. Additionally, alpha diversity was analyzed using Simpson, Shannon, and Chao1 indices. Beta diversity was assessed using Aitchinson distances (i.e., Euclidean distance over the CLR-transformed values) with principal coordinates analysis (PCoA). PERMANOVA implementation from the vegan library (v2.7.1) with the Euclidean distance matrix was used to test for significant differences in microbial diversity between groups, followed by post-hoc pairwise comparisons. For sex effect and interactions effects and additional detailed statistical analysis, see https://github.com/CristinaCuesta-Marti/earlylifeHFHS_feeding_microbiota_2025/tree/outputs/overall_stats_lm. Additional random effects (including litter and cohort effects) were assessed, and results of these statistical analyses can be found in https://github.com/CristinaCuesta-Marti/earlylifeHFHS_feeding_microbiota_2025/tree/outputs/overall_stats_lmer.

### Reporting summary
Further information on research design is available in the Nature Portfolio Reporting Summary linked to this article.

## Data availability
The transcriptomic data generated in this study have been deposited in the European Nucleotide Archive under accession code PRJEB89033. The metagenomic data generated in this study have been deposited in the European Nucleotide Archive under accession code PRJEB89407. The metabolomic data generated in this study have been deposited in MetaboLights under accession code MTBLS12714. The raw molecular and behavioral data have been deposited in the public GitHub repository (https://github.com/CristinaCuesta-Marti/earlylifeHFHS_feeding_microbiota_2025/tree/data) and archived via Zenodo [https://doi.org/10.5281/zenodo.17831151][174].

All statistical analyses have been deposited in the public GitHub repository (https://github.com/CristinaCuesta-Marti/earlylifeHFHS_feeding_microbiota_2025/tree/outputs/) and archived via Zenodo [https://doi.org/10.5281/zenodo.17831151][174].

## Code availability
The code used in this study has been deposited in the public GitHub repository (https://github.com/CristinaCuesta-Marti/earlylifeHFHS_feeding_microbiota_2025/tree/scripts) and has been archived via Zenodo [https://doi.org/10.5281/zenodo.17831151][174].

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

## Acknowledgements

The authors wish to thank Tara Foley, Dr Gerry Moloney, Patrick Fitzgerald and Dr. Kenneth J. O'Riordan for their invaluable technical assistance and support. The authors also wish to thank Dr Linda Engström Ruud for her help with RNAscope. The research was conducted and primarily funded by APC Microbiome Ireland, a research center funded by Research Ireland [Grant Number 12/RC/2273_P2] (formerly Science Foundation Ireland). H.S. received funding from the Biostime Institute for Nutrition & Care - BINC Foundation, which also contributed to the research directly, and Food for Health Ireland through the EI Technology Centre award (TC20180025). The authors wish to acknowledge the Centre for Cellular Imaging at the University of Gothenburg and the National Microscopy Infrastructure, NMI (VR-RFI 2016-00968), for providing access to their confocal microscope and for their assistance. C.C.-M. is funded by the Irish Research Council (Research Ireland) with a Government of Ireland Postgraduate Scholarship (GOIPG/2023/4836). E.P.E. was supported by the VI Program of Inner Initiative for Research and Transfer of the University of Seville (VI PPIT-US). F.U. was supported by a Research Ireland Pathway program [Grant Number 22/PATH-S/10882].

## Author contributions

H.S. conceived the overall study and designed the experiments with S.O.M., C.C.-M., and F.U. C.C.-M., E.P.-E., and F.U. conducted the in vivo experiments. C.C.M., I.S., and S.D. designed the RNAscope experiments. C.C.-M., E.P.-E., F.U., I.S., L.O.-R., and C.R.-C. carried out and contributed to additional experimental procedures and analyses. C.C.-M., K.V., L.W., and L.K. contributed to the design and carried out in vitro experiments. C.C.-M., T.F.S.B., E.P.-E., G.S.S.T., B.V., D.H., and A.L. conducted and contributed to the biostatistics and bioinformatics analyses. H.S., S.O.M., G.C., and S.D. coordinated and supervised the experiments. T.G.D. contributed domain expertise on *Bifidobacterium* strain development. H.S., C.C.-M., J.F.C., C.S., R.P.R., and S.D. contributed to the funding. C.C.-M. wrote the manuscript. H.S., S.O.M., G.C., F.U., S.D., I.S., C.R.-C., E.P.-E., B.V., and K.V. revised the manuscript. E.P.-E. and F.U. contributed equally to this work and reserved the right to list

themselves as first co-second authors in their CVs. All authors have critically reviewed the manuscript and consented to the final submitted version.

## Competing interests

The authors declare no competing non-financial interests. The following competing financial interests are declared: G.C. received honoraria from Janssen, Probi, Ingelheim Boehringer and Apsen, as an invited speaker, and research funding from Pharmavite, Reckitt, Tate & Lyle PLC, Nestle, Fonterra and is or has been a paid consultant for Heel Pharmaceuticals, Bayer Healthcare, Yakult and Zentiva; H.S. has obtained research funding from Pharmavite, Cremo, Tate & Lyle PLC, Pepsico, and Fonterra. This support did not influence or constrain the study design, data analysis, collection, data interpretation, writing or overall content of this manuscript. H.S., T.D., J.F.C., and C.S. are inventors on a patent for other findings previously identified with the in-house bacterial strain used in the present work (filed and pending, EP17187989.3, entitled: "*Bifidobacterium longum* APC1472 for treating obesity and weight management"). The remaining authors declare no competing interests.

## Additional information

[1]Department of Anatomy and Neuroscience, University College Cork, Cork, Ireland. [2]APC Microbiome Ireland, Cork, Ireland. [3]Instituto de Biomedicina de Sevilla, IBiS, Universidad de Sevilla, HUVR, Junta de Andalucía, CSIC, Sevilla, Spain. [4]Departamento de Bioquímica Médica y Biología Molecular e Inmunología, Facultad de Medicina, Universidad de Sevilla, Sevilla, Spain. [5]Department of Pharmacology and Therapeutics, University College, Cork, Ireland. [6]Department of Physiology/Endocrine, Institute of Neuroscience and Physiology, The Sahlgrenska Academy at the University of Gothenburg, Gothenburg, Sweden. [7]The Life Sciences Interface Group, Tyndall National Institute, Lee Maltings Complex, Cork, Ireland. [8]Teagasc Food Research Centre, Moorepark, Fermoy, Ireland. [9]Department of Psychiatry and Neurobehavioral Science, University College Cork, Cork, Ireland. [10]These authors contributed equally: Eduardo Ponce-España, Friederike Uhlig. ✉e-mail: CCuestaMarti@ucc.ie; h.schellekens@ucc.ie

