## [Transparent Peer Review file · Nature Communications]

Bifidobacterium longum and prebiotic interventions restore early-life high-fat/high-sugar diet-induced alterations in feeding behavior in adult mice

Corresponding Author: Dr Harriet Schellekens

Version 0:

Reviewer comments:

Reviewer #1

(Remarks to the Author)

The work from Cuesta-Marti et al., explores the enduring effects of early life high-fat/high sugar diet exposure on feeding behavior. They find that HFHS diet indeed have lasting effects on feeding behavior in sex dependent manner, and this effect can be attenuated by prebiotic mixture (FOS+GOS) or B. longum administration. The findings are novel, and very impactful with clinical implications. The conclusions are based on sound experimental design. However, I have a few concerns and suggestions for clarity:

1. FOS+GOS and B. longum has enduring effects on feeding behavior for both female and males mice but the effects are not necessarily through same mechanisms, authors do not discuss or explore what would be the discrepancies in the mechanisms. We see more changes in female mice in metabolome and transcriptome in the brain however there are some missing connections to provide information how these affects connects from gut to brain. For that, there are couple things to consider:

- a. Is B. longum observed in the intestine, cecum, colon, small intestine even? Are they passengers in the GI tract that's why they are found in the feces abundantly? The mechanism of action is not clear. It does not shift microbiome composition. Does it change the metabolome in the microbiome? An overtime abundance analysis would confirm whether their abundance is increased. How does B. longum restores genes in the brain unclear. Through metabolite production? But then it needs to be confirmed that it colonizes because it seems like it does not change the microbiome composition.
- b. It would be good to include unpublished in vitro work where B. longum produces tryptophan. Did B. longum increase serotonin levels in the brain, how about enterochromaffin cells? It increases turnover but not necessarily serotonin levels. Were tryptophan levels affected in the gut. A metabolome analysis of the gut is needed.
- c. The study needs correlation and/or multiassociation analysis to bring blood metabolome and brain transcriptome together and even gut microbiome and metabolome if those experiments are done.
- d. A picrust analysis of the microbiome that connects blood metabolome to microbiome is needed.

2. There needs some clarity in the experimental design

- a. Is it clear the offspring is not exposed to the HFHS diet during pre-weaning period or are the effects through mothers milk? The Fig 1A is not clear when does the offspring start eating HFHS diet. Are there any measurements that they consume HFHS crumbs during postnatal period?
- b. What is the composition of mothers milk during postnatal period? Does B. longum travel through mothers milk?
- c. Are there any changes in dams rearing behavior between treatment groups and control groups that might affect feeding behavior?
- d. Why is the reason the administration of prebiotic mixture and B. longum continues even after switching to control diet. If the reason looking at early life exposure, shouldn't they also be stopped? It does not necessarily meant early life exposure because the treatment continues to adulthood.
- e. Why aren't there any prebiotic+Blongum group? What about synbiotic effects of FOS+GOS+ B. longum.
- f. Was the group housing done randomly in the weaning period or did siblings go to the same cage?
- g. There will be cage to cage or cohort to cohort difference in the microbiome analysis, was that considered?
- h. How many dams per group was recruited in the beginning to reach offspring n=12 per group?

3. Authors only address Euclidean distance in beta-diversity. Whereas Euclidean is not enough for compositional data due to its inability to account constant sum. Therefore Aitchison distance that takes log-ratio transformation into account is

recommended and this will give more information about prevalence. In addition, weighted unifrac for calculating beta diversities will give information about abundance based on phylogenetic taxa observed. In addition, authors should discuss alpha diversity in the context of species evenness and richness. This is important especially when B. longum administration does not increase the number of total bifidobacterium but B. longum numbers increase in feces.

- a. Authors don't discuss that B. longum administration does not necessarily increase total Bifidobacterium abundance whereas it increases in B. longum. That seems like a potential competition between other bifidobacterium and B. longum or B. longum found in feces not necessarily colonized in the gut. See my earlier comment on this suggestion.
 - b. Which Bifidobacterium species were enriched during FOS+GOS administration?
4. The conclusion will need to be tailored, that could also be reflected in the graphical abstract. I think that authors did not pay much attention in the discussion about the different effects/mechanisms prebiotic vs B. longum and male vs female. Early life HFHS has different effects male vs female and FOS+GOS and B. longum attenuates these effects different for male and female mice. The abstract should reflect that finding, which mechanisms were found in female and which are found in male. In male mice, PNOC neuron and Nod2-expressing cells are altered whereas in female mice, POMC neurons and Ghnr and Lepr receptors are affected. "a reduction in the number of cells expressing Ghnr, Lepr and Pomc individually in the ARC of adult early-life HFHS exposed female mice while Ghnr and Pomc were only reduced in adult early-life HFHS exposed males compared to control
5. Authors should also focus on the mechanisms of FOS+GOS and B. longum more clearly. Because they have different effects on the HFHS induced alterations however whether the mechanisms are different is unclear. It seems like FOS+GOS makes shifts in the microbiome composition whereas no clear shifts were observed for B. longum however B. longum displays more lasting effects. This can be also addressed by looking at metabolomics in fecal or cecal samples.
6. There are some typos listed:
Line 36 not symbiotics, synbiotics..
Line 644 should be POMC. typo

(Remarks on code availability)
the code is not deposited to github

Reviewer #2

(Remarks to the Author)

This is a well-written paper and a comprehensive study design that included four experimental groups, two microbiota-targeted interventions (prebiotic, probiotic) and two dietary control groups, and multiple analyses (including behavioural, metabolomic, transcriptomic, microbiota profiling, and fluorescent in situ hybridization). The inclusion of both male and female mice and consistent presentation of results for both sexes highlighting sex-specific effects is an important contribution and an often-overlooked factor. The detailed inclusion of supplementary data including body weight and gain in both graph and table format and details of various tissue measures and weights is commendable. The measurement of food intake and food crumbling also adds to the precision of the food intake recording.

The quality of the data appears technically sound and is presented in detail. The conclusions made are not excessive and are supported by the evidence provided. These findings are relevant and add to the evidence for early life effects of diet on later food intake behaviours in mice while the sex differences shown highlight the importance of including both male and female mice and reporting their data separately.

Comments

There are some points that could improve the manuscript:

1. The control diet used is detailed as "D12451 modified (Product N S9912-E760" but the modifications made to the D12451 diet are not clear. Can these modifications be clarified.
2. How was the probiotic dosing decided upon? The dosage used in drinking water is based upon that used in previous research and adapted for the offspring but how was this adapted for the offspring?
3. The methods state that mice were distributed evenly across experimental groups. How were these allocations decided and was any randomisation method used? If so please state the method used to allocate mice into experimental groups.
4. Are the long-term changes observed in the arcuate nucleus relevant to later control diets? This point could be mentioned in the discussion, that the changes in responses to food later in the study appeared to be only in response to further experimental tests and that body weight normalised in all groups when on control diet in the later part of the study period despite earlier exposures.
5. Does the FOS/GOS in drinking water remain adequately dissolved and suspended during the time the water is supplied? Has this been measured in a similar manner to the conformation of the survival of the probiotic in the drinking water to ensure consistent dosage?
6. Was the amount of water the mice drink checked and was this used to confirm that the mice were receiving the expected dose of the FOS/GOS or probiotic?
7. Mice pups appeared to show increased body weight gain during the suckling period when dams were fed High-fat/High-

sugar diet. Is there a potential explanation for this effect through the milk given the same control diet was used during pregnancy?

8. In supplementary figure 1D the food and energy intake tend to be higher in High-fat/High- sugar diet fed mice given FOS/GOS in their drinking water. Are there any suggested reasons for this?

9. The body weight gain appears higher in mice supplemented higher in FOS/GOS? Is this due to increased cecum weight and intestinal length? This could be included as a discussion point as this physiological effect of fermentable fibre in mice is quite significant as this difference (Supplementary Figure 2A) appeared to persist after the mice were returned to the control diet.

10. It is stated that "early-life HFHS diet reduces the abundance of Bifidobacterium genus" and that "restored by FOS+GOS administration". But it is not mentioned that the HFHS diet is deficient in any fermentable fibre that can be metabolised by Bifidobacterium. While the D12451 control diet is lacking in added fibre it contains a lot of corn starch that can act as resistant starch and is a fermentable carbohydrate to Bifidobacterium. It could be mentioned that adding FOS+GOS to the diet is restoring the loss of any fermentable carbohydrate in the HFHS diet.

(Remarks on code availability)

Reviewer #3

(Remarks to the Author)

The paper investigates the effects of early exposure to a high fat high sugar diet on feeding behavior, blood and brain neurochemistry. The potential therapeutic effect of microbiota-based interventions during and after diet exposure are also tested. These microbiota-targeted interventions improved feeding behavior, brain neurochemistry and restore some altered blood metabolomics parameters. Although it is unclear if changes are prevented by the interventions during diet exposure or if restoration / normalization happen after diet exposure. Similarly, the dietary switch happened at PND35, blurring the distinction between early-life exposure and post-weaning /adolescence exposure

Additional comments:

English needs to be thoroughly edited in the abstract

The introduction gives a broad overview of the homeostatic and hedonic systems involved in regulating feeding, a more focused introduction on the perinatal influence and the role of the microbiota in modulating both systems (and how they are connected) would be more appropriate.

The rationale for focusing on the hypothalamus is not well justified as microbiota has been shown to influence vagal /NTS signaling in the homeostatic regulation of feeding and the references provided show effect of the reward areas (NAc and VTA), which are also under vagal influence. Data presented in the supp materials also show a strong effect on meal size, again more a vagal /NTS regulated behavior. Maternal studies have shown strong effect on hypothalamic factors but these are not cited in the manuscript.

Fig1

- Text refers to figure 2 panels that are actually figure 1
- Preference for saccharin may be related to different threshold in sweet perception, as HF feeding has been shown to alter sweet testing, this should be mentioned
- Does food crumbling directly correlate to food intake? If this is the case, the observed reduction in crumbling may be result of the overall reduction in intake. It may also be reflective of exploratory behavior or stress
- Results state that control and treated group have a higher preference for saccharin (males), this appears to be contradictory to the data displayed on the graph?

The offspring are only exposed to the HFHS diet for a limited period of time, after which their body weight is normalized. This is a key piece of data as it supports that observed changes in adulthood are not due to differences in food intake or adiposity. Body weight and food intake data should be included in the main results.

It is still unclear however whether the effects observed are due to 1) changes in milk composition during lactation or 2) changes in feeding during that period or 3) effects post-weaning. This limitation should be addressed.

Similarly, despite the interventions having beneficial effects on hypothalamic transcriptomics, the animals appear to overeat when exposed to the HFHS diet. This should be discussed, it is unclear whether the changes are prevented by the interventions or reversed post- diet exposure.

Minor

Line 11 -References are different format

Line 30 – Vagal afferents do not terminate in proximity to the median eminence or the hypothalamus. Ref 42 focuses on the NAc and is out of place here

Line 30 – Implication of the microbiota in reward driven feeding should include Kim et al. Molecular Metabolism, 2023 Supp. Food intake should be presented in kcal not grams as diets have different kcal density
Was any power analysis conducted ? Omics outcomes display high variances, and the groups may be underpowered

(Remarks on code availability)

Reviewer #4

(Remarks to the Author)

This paper explores the effects of early-life high-fat/high-sugar (HF/HS) diet and microbiota modulations by a prebiotic mix or putative probiotic on later life feeding behavior, microbiota composition, and transcriptomic differences in the arcuate nucleus of the hypothalamus. Additionally, it explores the differences in male and female offspring showing that the effect of early-life HF/HS diet is sex dependent.

The manuscript adds interesting results to the existing literature about early-life diet alterations impact on later life but addresses it with the male/female difference point of view (which is often left out).

The methodology is clearly explained and stated, to the exception of a few minor details, and allow reproducibility of the experiments conducted.

The graphs are clear for the most part and the graphical abstract/protocol visual description help with the comprehension.

I enjoyed the statistical approach taken (difference between the two diet – vehicle groups followed by measure of the restoration of the parameter by the microbiota intervention), it was a nice way of analysing the data, that should be used more often.

The introduction and discussion places the article in context with good reference to current literature. These two sections are very clear and nice to read.

However, some significant changes are required before publication are listed below.

Overall, a re-organisation of the figures and a proper labeling of them throughout the manuscript are a needed change as it was difficult to match the figure number with the number in the text.

Overall comment

- Reorder the figures and supplementary figures to make them appear in a logical order either in the text or in the figure files. Having them so disordered makes it hard to follow the manuscript. Additionally in the text, some figures are misnumbered (ex: Fig 2B (line 468) is actually Fig 1B).

- Statistics need to be described in a fairer and clearer way. While the statistical tests are really adapted to represent the restoration by both interventions, sometimes the changes induced by HFHS vehicle are not significant compared to the control vehicle group, but the restoration effect is significant. While this non significance doesn't take anything away from the trend/biological effect, it is sometimes stated clearly and sometimes not in the text. A more consistent depiction of the statistics would benefit the manuscript.

For example:

o line 490 "Interestingly, both the prebiotic FOS+GOS and B. longum APC1472 restored the higher 491 preference towards saccharin in adult male mice that were exposed to an early-life HFHS", the difference between the control vehicle group and the HFHS vehicle group is not significant ($p=0.12$) although the restoration effect by both microbiota interventions are.

Abstract:

The abstract could benefit from breaking down some of the sentences into multiple sentences as it is hard to follow as it is written now.

Material and methods

"Animals, diets and ethical approval"

- Indicate how many dams were housed per cage

- Indicate if inside of a same treatment group offspring were mixed after weaning to avoid cage effect

- Figure 1A and S1B are both very clear and helpful but redundant, chose one to keep in the manuscript

Bioinformatics & statistical analysis

- "Additional random effects (including 426 litter and cohort effects) were assessed using linear mixed-effects models using the lmer function in R" - I couldn't find the results of these tests in document S2. If it is not, please add it, if it actually is contained in this file, please make it clearer so the reader can find the information.

Results

- Please add a sentence to describe Fig S1 E and F

- Line 459 "Interestingly, we found that early-life HFHS-exposed offspring did not consume all food removed from the food hopper but crumbled part of it" - - Please show data for this claim if possible as the graphs represent cumulative consumption and crumbling behaviour.

- Fig S4B: harmonize the serotonin names between the text and the figures + discuss the 9 graphs that are not mentioned in the text.

- Line 503 to 512, please re-write this section to make the differences between the IpGTT results in male and female clearer.

- Line 511, for the basal glucose graph description, "restored" has been used in the rest of the manuscript and should be used here as well to keep the results clear and consistent as it is going back to control vehicle group levels. For the males, please add additional statistics and discussion of the effect of FOS+GOS vs control vehicle on basal glucose.

- Line 517: what about the other organs in Fig S5?
- Line 522 to 536: please mention the alpha-diversity results
- FigS3D to I are not described
- Line 577, there is no mention on the effect/absence of effect of B. longum APC1472 on the males' metabolome.
- For the section "Early-life HFHS has an enduring and sex-specific impact on hypothalamic transcriptomic profile in adult mice", it would be interesting the mention the gene commonly restored by both male and female FOS+GOS supplementation in fig.3D
- Line 647, the difference is not significant for Lepr
- Line 651, Figure 4F doesn't exists (or I couldn't match the text with the figures)
- Line 669: It is only true for females
- Line 669: the POMC+ Ghsr expressing cells count graph doesn't exist (or I couldn't match the text with the figures)
- Line 692: not significant for Pnoc
- In the section "Enduring molecular impact of an early-life HFHS diet on PNOG GABAergic neurons and Nod2-expressing cells in the arcuate nucleus of the hypothalamus of adult male and female mice" please harmonize the writing conventions of the genes expressed and cell populations, this section is very confusing for the reader.

(Remarks on code availability)

The code was not provided so I could not review it

Version 1:

Reviewer comments:

Reviewer #1

(Remarks to the Author)

I have no additional comments. The authors addressed all my previous comments and concerns very thoroughly. I recommend this manuscript for publication.

(Remarks on code availability)

The code is not available. The link provided leads to a "Page not found" error.

Reviewer #2

(Remarks to the Author)

(Remarks on code availability)

The link has been provided but the code does not appear to be uploaded yet.

https://github.com/CristinaCuesta-Marti/earlylifeHFHS_feeding_microbiota_2025

Reviewer #4

(Remarks to the Author)

The work done by the authors to answer the reviewer's comments both in the manuscript and in the rebuttal is very nice and adds a lot of clarity to the paper. It really helps to understand their work that comprises multiple techniques, analyses and statistical tests. Thank you for taking the time to perform all the changes and answer our comments with clarity and detail and congratulations on this very interesting study.

(Remarks on code availability)

Unfortunately, although the Github link to the code and other supplementary material didn't work on my side and gave me a 404 error. If it is a general problem, please review that there is no typos in the link so the material becomes accessible.

REVIEWER COMMENTS

We appreciate the time and effort that the reviewers have dedicated to providing valuable feedback on our manuscript. We are grateful to the reviewers for their insightful comments on our paper.

We have been able to incorporate changes to reflect most of the suggestions provided by the reviewers. We also have performed significant number of additional works:

1. Analysis of short-chain and branched-chain fatty acids in the caecum of 5-week and 12-week-old female and male mice.
2. Microbial functional potential analysis using PICRUSt2 to generate KEGG orthologue predictions, pathway filtering and analysis focused on metabolism of significantly altered metabolites, and computation of Gut-Brain Modules (GBMs) in 10-week-old mice.
3. Integrated pathway analysis incorporating differentially abundant circulating metabolites and differentially expressed genes in the hypothalamus using MetaboAnalyst to identify modulated pathways.
4. ASV-level analysis to examine strain-specific changes within the *Bifidobacterium* genus between Control Vehicle and HFHS *B. longum* APC1472 female and male mice at 5 and 10 weeks of age.
5. Abundance analysis of all *Bifidobacterium* species identified in fecal samples (*B. pseudolongum*, *B. longum*, and *B. breve*) across treatment groups in 5-week and 10-week old mice.
6. Analysis and plotting the daily drink intake throughout the study.
7. Analysis of distance traveled and time spent in the external zone (10%) of the open field test to assess thigmotaxis and exploratory behavior.

We have submitted the new version with the manuscript with tracked changes, following the journal guidelines. In addition, we have updated the access code to the repository but they are not publicly available yet.

A point-by-point response to the reviewers' comments can be found below.

Reviewer #1

(Remarks to the Author):The work from Cuesta-Martí *et al.*, explores the enduring effects of early life high-fat/high sugar diet exposure on feeding behavior. They find that HFHS diet indeed have lasting effects on feeding behavior in sex dependent manner, and this effect can be attenuated by prebiotic mixture (FOS+GOS) or *B. longum* administration. The findings are novel, and very impactful with clinical implications. The conclusions are based on sound experimental design. However, I have a few concerns and suggestions for clarity:

1. FOS+GOS and *B. longum* has enduring effects on feeding behavior for both female and males mice but the effects are not necessarily through same mechanisms, authors do not discuss or explore what would be the discrepancies in the mechanisms. We see more changes in female mice in metabolome and transcriptome in the brain however there are some missing connections to provide information how these affects connects from gut to brain. For that, there are couple things to consider:

1.1. Is *B. longum* observed in the intestine, cecum, colon, small intestine even? Are they passengers in the GI tract that's why they are found in the feces abundantly?

We thank the reviewer for these insightful questions regarding *B. longum* localization, and colonization patterns following administration.

As shown in Fig 1H (Fig 1J now), administration of *B. longum* APC1472 increases the relative abundance of *B. longum* species in fecal samples in 5- and 10-weekold male and female offspring, which might indeed suggest they are predominantly passengers or colonize transiently. To date, we have not performed extensive colonization studies for our *B. longum* APC1472 strain. Therefore, we cannot conclude whether its effects are through (transient) colonization. However, there is ample evidence that colonization is not required for beneficial effects of probiotics for the host (ISAPP, 2022). Tofani et al. (2025) administered a single dose of the probiotic strain, *L. reuteri* ATCC PTA 6475 and showed to modulate corticosterone release (Tofani et al. 2025).

Different strains of *B. longum* have been shown to persistently or transiently colonize the entire gastrointestinal tract – including the small intestine, cecum, and colon—with the highest abundances typically detected in the colon (Chaplin et al., 2015; Lv et al., 2015; Gutierrez et al., 2023). For instance, *B. longum* BBMN68 colonized the gut of rats within one day of oral administration, reaching populations exceeding 10^8 cells per gram in both cecal and colonic contents (Lv et al., 2015). In humans *B. longum* strain AH1206 persisted in the intestinal tract of 30 % of individuals for at least six months following a single oral administration, while other *B. longum* strains have been detected for up to 5-10 years in children's intestines (Maldonado-Gómez et al., 2016; Chaplin et al., 2015; Shkoporov et al., 2013).

To provide context regarding gut colonization of *B. longum* and its relevance for host effects, we have added this in the introduction (lines 53-55).

The ISAPP quick guide to probiotics for health professionals: History, efficacy, and safety. ISAPP <https://isappscience.org/for-clinicians/resources/probiotics/> (2022).

Tofani, G. S. S. et al. Gut microbiota regulates stress responsivity via the circadian system. *Cell Metab.* 37, 167-185.e8 (2025).

Chaplin, A.V. et al. Intraspecies Genomic Diversity and Long-Term Persistence of *Bifidobacterium longum*. *PLOS ONE* 10(8): e0135658 (2015).

Lv, Y. et al. Biodistribution of a Promising Probiotic, *Bifidobacterium longum* subsp. *longum* Strain BBMN68, in the Rat Gut. *J Microbiol Biotechnol.* 25(6):863-71 (2015).

Gutierrez, A. et al. *Bifidobacterium* and the intestinal mucus layer. *Microbiome Res Rep.* 2(4):36 (2023).

Maldonado-Gómez, M.X. et al. Stable Engraftment of *Bifidobacterium longum* AH1206 in the Human Gut Depends on Individualized Features of the Resident Microbiome. *Cell Host Microbe* 20(4):515-526 (2016).

Shkoporov, A.N. et al. Draft Genome Sequences of Two Pairs of Human Intestinal *Bifidobacterium longum* subsp. *longum* Strains, 44B and 1-6B and 35B and 2-2B, Consecutively Isolated from Two Children after a 5-Year Time Period. *Genome Announc.* 1(3):e00234-13 (2013).

- 1.2. The mechanism of action is not clear. It does not shift microbiome composition. Does it change the metabolome in the microbiome? An overtime abundance analysis would confirm whether their abundance is increased. How does *B. longum* restores genes in the brain unclear. Through metabolite production? But then it needs to be confirmed that it colonizes because it seems like it does not change the microbiome composition.

We thank the reviewer for these insightful questions.

With regards to the microbiome composition, we apologize for oversimplification of our findings. We have now revised the section to highlight that while early-life HFHS diet did not induce persistent changes in alpha and beta diversity in adult male and female mice (10 weeks), we observed differences at genus and species level, some of which were sex-specific (lines 649-664 and 730-748, and Supplementary Table S7).

We have only undertaken blood metabolomic analysis in adult (12 weeks) mice because this is the timepoint most closely aligned with our behavioral tests. However, since changes in week 5 metabolome may indeed have long-term effects, we have now undertaken functional analysis (PICRUSt2 and Gut-Brain modules (GBMs), see response to point 1.4) of the gut microbiome (16S) at both timepoints (week 5 and 10) as well as cecal SCFAs and BCFAs abundance analysis at week 5 and 12 of age.

We have selected SCFAs and BCFAs because these microbial metabolites, particularly acetate and butyrate, can cross the blood-brain barrier and directly influence hypothalamic POMC neurons, key regulators of appetite and energy homeostasis (Frost *et al.*, 2014; Perry *et al.*, 2016). Briefly, we found that early-life HFHS diet induced sex-specific alterations of SCFA/BCFA in 5-week-old pups. All analyzed SCFA/BCFA were significantly reduced by early-life HFHS diet at week 5 in males while no changes were observed in females. FOS+GOS and *B. longum* APC1472 restored acetate levels in 5-week-old males. Interestingly, regardless of dietary alteration, administration of *B. longum* APC1472 increased acetate in male mice, and butyrate, propionate, and valerate levels in females at 5 weeks of age. With regards to week 12, there were no significant effects of early-life HFHS diet on SCFA/BCFA levels. However, FOS+GOS administration showed trends toward increased butyrate, propionate, and acetate levels compared all the groups, though these findings should be interpreted cautiously due to low sample size (n=2 for FOS+GOS females at week 12) and technical limitations (see below). We describe these findings results section (lines 718-729, Supplementary Fig. S5, see below) and included the methodology (lines 410-439; statistical analysis in https://github.com/CristinaCuesta-Marti/earlylifeHFHS_eating_microbiota_2025/outputs/overall_stats).

Unfortunately, technical limitations (procedural error) during sample preparation affected the SCFA/BCFA analysis for week 12 samples. To mitigate this, we reanalyzed the remaining cecal filtrate; however, the available volumes were considerably below optimal, requiring proportional adjustments to the internal standard. As a result, quantification accuracy and data quality at week 12 were impaired. These results are therefore presented descriptively in the supplementary materials for completeness but are not emphasized as primary findings or integrated with other datasets.

New Supplementary Fig. S5

With regards to the mechanism of action of *B. longum* APC1472, the new SCFA/BCFA analysis thus suggests these microbiota-derived metabolites to be a contributing factor to the *B. longum* APC1472 beneficial effects. In addition, as shown in the functional microbiome analysis (see point 1.4), SCFAs were identified as one of the potential pathways by which *B. longum* APC1472 may be exerting its restorative effects. In addition, we found that early-life HFHS depleted SCFA-producing genera in feces of week 10 mice, some of which were restored sex-specifically by *B. longum* APC1472 (lines 730-748). However, other metabolites, such as tryptophan for females, may also play a role, and we have described this in the discussion.

As discussed in point 1.1, changes in microbiota-derived metabolites can occur whether or not colonization occurs. Therefore, increased detection of *B. longum* species in the feces confirms its presence with minimal disruption to the overall microbiome composition and is consistent with its potential to influence metabolite levels.

Frost, G. et al. The short-chain fatty acid acetate reduces appetite via a central homeostatic mechanism. *Nat. Commun.* 5, 3611 (2014).

Perry, R. J. et al. Acetate mediates a microbiome–brain– β -cell axis to promote metabolic syndrome. *Nature* 534, 213-217 (2016).

1.3. It would be good to include unpublished *in vitro* work where *B. longum* produces tryptophan. Did *B. longum* increase serotonin levels in the brain, how about enterochromaffin cells? It increases turnover but not necessarily serotonin levels. Were tryptophan levels affected in the gut. A metabolome analysis of the gut is needed.

Thanks for this comment. We have included below the work showing that *B. longum* APC1472 can produce tryptophan *in vitro* in late stationary phase, which has been recently accepted in *npj Biofilms and Microbiomes* and publicly available (edited from Main Figure 1b; Cuesta-Marti et al. 2025, <https://doi.org/10.1038/s41522-025-00820-9>). As shown in this work, cell-free conditioned supernatants (CCSs) of *B. longum* APC1472 harvested at late stationary phase demonstrated a higher relative abundance of tryptophan compared to control (see figure below adapted from this publication, demonstrating the higher abundance of the late stationary phase in conditioned cell-free supernatants (CCSs)).

a) *In silico* analysis of Gut-Brain Modules

b) Semi-Polar Metabolites

Regarding the *in vivo* effects, *B. longum* APC1472 significantly increased the non-significant reduction in serotonin (5-HT) levels in the striatum of adult male mice following early-life HFHS diet exposure (updated in lines 605-611). In female mice, *B. longum* APC1472 normalized the diet-induced reduction in serotonin turnover (5-HIAA/5-HT ratio) rather than increasing absolute serotonin levels. Additionally, *B. longum* APC1472 restored the alteration in circulating tryptophan levels in adult female mice.

Our study focused on central serotonin rather than evaluating peripheral serotonin synthesis. While we acknowledge that both enterochromaffin cell analysis (as the primary source of peripheral serotonin) and gut metabolome profiling would provide valuable complementary data, our research prioritized the central nervous system. It is known that serotonin produced by enterochromaffin cells cannot readily cross the blood-brain barrier due to its hydrophilic nature (Matthes *et al.*, 2018). Additionally, our research question is focused on understanding the central serotonin levels as the more direct indicator of potential central effects relevant to the observed feeding behaviors. Additionally, our *in vitro* data demonstrated *B. longum* APC1472's capacity to produce tryptophan, and we sought to determine whether this translated to measurable central nervous system effects rather than characterizing the complete peripheral metabolic landscape.

Cuesta-Marti, C., Valderrama, B., Bastiaanssen, T. et al. *In vitro* assessment of bacterial supernatants on hypothalamic gene expression: implications for appetite regulation. *npj Biofilms Microbiomes* 11, 192 (2025). <https://doi.org/10.1038/s41522-025-00820-9>

Matthes, S. & Bader, M. Peripheral serotonin synthesis as a new drug target. *Trends Pharmacol. Sci.* 39, 560-572 (2018).

1.4. The study needs correlation and/or multiassociation analysis to bring blood metabolome and brain transcriptome together and even gut microbiome and metabolome if those experiments are done. A picrust analysis of the microbiome that connects blood metabolome to microbiome is needed.

We appreciate the reviewer's suggestion for integration analysis between our multi-omics datasets to investigate potential sex-specific and intervention-specific mechanisms underlying the observed behavioral improvements.

We have now performed integrated pathway analysis incorporating differentially abundant circulating metabolites (blood metabolomics) and differentially expressed genes (transcriptomics) in the hypothalamus of 12-week-old female and male mice ($q < 0.2$), as detailed in the updated methods section 2.14 (lines 399-409) and presented in Supplementary Fig. S8 and results section (lines 841-871 and 874-876). This analysis used MetaboAnalyst to integrate KEGG pathways that are altered by early-life HFHS diet and restored by microbiota-targeted interventions in both blood metabolomics and hypothalamic transcriptomics. This analysis revealed EGFR tyrosine kinase inhibitor resistance and mTOR signaling altered by diet in females and AMPK signaling and beta-alanine metabolism in males. FOS+GOS restored pyrimidine and purine metabolism in females and fatty acid degradation and elongation in males. *B. longum* APC1472 restored pantothenate and CoA biosynthesis and DNA replication in females and growth hormone synthesis, secretion and action as well as valine, leucine and isoleucine degradation in adult males. This further underlines sex-specific and intervention-specific mechanisms.

New Supplementary Fig. S8

Following the reviewer's recommendation, we also performed PICRUSt2 analysis and incorporated Gut Brain Modules (GBMs) analysis to examine predicted functional capacity of the microbiome in relation to neuroactive compounds. Following that, to integrate microbiota and blood metabolomics data we filtered PICRUSt2-derived pathways to retain only those involved the metabolism of blood metabolites significantly altered by early-life HFHS diet or restored by interventions in adult mice (12-week-old). These results are described in lines 759-770 and Supplementary Fig. S7 (see below). Further statistical information of GBMs can be found on https://github.com/CristinaCuesta-Marti/earlylifeHFHS_eating_microbiota_2025/outputs. The multi-omics findings have been also discussed regarding implications for dietary and microbiota-targeted interventions effects in a sex-specific manner in the Discussion (lines 971-989).

Briefly, these results suggest that early-life HFHS induces persistent changes in 13 gut-brain modules (GBMs) for example neurotransmitter metabolism and tryptophan degradation in (females), as well as 5 GBMs related to propionate synthesis and neurotransmitter metabolism (males) (Supplementary Fig. S7A). FOS+GOS and *B. longum* APC1472 restored 10 and 2 out of 13 GBMs, respectively, in females, and 4 and 1 out of 5 in males.

Regarding the integrative analysis of PICRUSt2 and metabolomics, we found that the pathways altered by early-life HFHS diet in females (13 pathways including arginine, fatty acid and tryptophan metabolism) and males (4 pathways including fatty acid and arginine metabolism) in blood were also present and significantly altered in the functional fecal microbiota analysis (Supplementary Fig. S7B). FOS+GOS administration restored 5 out of the 13 altered pathways (arginine and fatty acid metabolism) in blood and fecal microbiota in females and 1 out of 4 pathways (fatty acid β -oxidation) in males. Further, *B. longum* APC1472 restored 1 pathway in both female and male (fatty acid β -oxidation) in blood metabolome and fecal microbiome.

These integrative approaches allowed us to identify consistent changes across blood metabolome, brain transcriptome, and gut microbiome data, providing mechanistic insights into sex- and intervention-specific effects on the gut-brain-metabolome axis.

New Supplementary Fig. S7

2. There needs some clarity in the experimental design
 2.1. Is it clear the offspring is not exposed to the HFHS diet during pre-weaning period or are the effects through mothers milk? The Fig 1A is not clear when does the offspring start eating HFHS diet. Are there any measurements that they consume HFHS crumbs during postnatal period?

We thank the reviewer for this important clarification request. We have updated Figure 1A to more clearly indicate the timeline of diet exposure.

Based on literature, mouse pups begin eating solid food around postnatal day 16, coinciding with eye opening and functional vision (MGI; Bailoo *et al.* 2020). Therefore, we expect that

direct consumption of diets (solid food) may occur prior to weaning (PND21), but it was not feasible to distinguish this from the dam's food intake.

Additionally, our research aimed to investigate the effects of HFHS diet during the *complete* early-life period, encompassing both the lactation period (through maternal milk) and the period when pups consume solid food independently (either pre- or postweaning). We intentionally selected this comprehensive early-life period, where maternal and direct dietary influences overlap because it reflects the window where early-life dietary programming effects occur.

Mouse Genome Informatics (MGI). THE POSTNATAL PERIOD. Available at: <http://www.informatics.jax.org/silver/chapters/4-4.shtml>.

Bailoo, J. D., Voelkl, B., Varholick, J., Novak, J., Murphy, E., Rosso, M., Palme, R. & Würbel, H. Effects of weaning age and housing conditions on phenotypic differences in mice. *Sci. Rep.* 10, 11998 (2020).

2.2. What is the composition of mothers milk during postnatal period? Does *B. longum* travel through mothers milk?

We appreciate the reviewer's question regarding maternal milk composition and potential bacterial/prebiotic transfer. The scope of the current study however was to investigate whether exposure to HFHS throughout the entire early life period (lactation and direct dietary exposure) affected adult feeding behavior, and whether this is sensitive to microbial intervention.

We agree that understanding the composition of maternal milk and the potential transfer of *B. longum* during the postnatal period is an interesting and relevant question. We had considered collecting breastmilk as it would indeed be a potential route of transmission. However, our study design did not allow us to collect mothers milk as dams were euthanized at PND22-23 where pups had been weaned and transitioned to solid foods, and dams were no longer in active lactation. Therefore, we can not examine whether the composition of the mother milk is affected by treatments (it is likely it was) or whether *B. longum* or FOS+GOS may be transferred through this mechanism. We have added this limitation to the Discussion section (lines 1129-1139).

Regarding the possibility of *B. longum* APC1472 transfer through maternal breastmilk, while we cannot directly assess this in our study, researchers at APC Microbiome Ireland have demonstrated that *Bifidobacterium* strains, including *B. longum*, can indeed be transmitted from mother to infant via breast milk (Feehily et al., 2023). Additionally, an earlier randomized controlled trial confirmed that *Bifidobacterium* strains, including *B. breve*, can transfer from maternal gut to breast milk and subsequently to the infant gut (Moore et al., 2023). Other studies found maternal-to-infant transmission of bacterial strains belonging to different genera including *Bifidobacterium* through breastfeeding (Murphy et al., 2017; Martín et al., 2012).

Feehily, C., O'Neill, I. J., Walsh, C. J., Moore, R. L., Killeen, S. L., Geraghty, A. A. et al. Detailed mapping of *Bifidobacterium* strain transmission from mother to infant via a dual culture-based and metagenomic approach. *Nat. Commun.* 14, 3015 (2023).

Moore, R. L., Feehily, C., Killeen, S. L., Yelverton, C. A., Geraghty, A. A., Walsh, C. J. et al. Ability of *Bifidobacterium breve* 702258 to transfer from mother to infant: the MicrobeMom randomized controlled trial. *Am. J. Obstet. Gynecol. MFM* 5, 100994 (2023).

Murphy, K., Curley, D., O'Callaghan, T. F., O'Shea, C-A., Dempsey, E. M. & O'Toole, P. W. et al. The composition of human milk and infant faecal microbiota over the first three months of life: a pilot study. *Sci. Rep.* 7, 40597 (2017).

Martín, V., Maldonado-Barragán, A., Moles, L., Rodríguez-Baños, M., Del Campo, R. & Fernández, L. et al. Sharing of bacterial strains between breast milk and infant feces. *J. Hum. Lact.* 28, 36–44 (2012).

2.3. Are there any changes in dams rearing behavior between treatment groups and control groups that might affect feeding behavior?

We did not assess dam rearing behavior as part of our study design. While we acknowledge that maternal rearing behavior could potentially influence offspring feeding behavior, systematically assessing dam behavior during the lactation period would have presented significant methodological challenges and potentially introduced confounders for our study objectives.

Specifically, assessment of maternal rearing behavior would require repeated handling and separation of dams from pups during the lactation period which is often used in animal models as postnatal stressor and can alter both maternal care patterns and offspring neurodevelopment (Rombaut et al., 2011). For example, maternal separation during the lactation period produces long-lasting alterations in offspring behavior, including altered feeding behavior (de Souza et al., 2020), increased anxiety and depressive-like behaviors (O'Mahony et al., 2011), as well as disrupted stress responses and cognitive function (Rincel & Darnaudéry, 2020).

Rombaut, C., Roura-Martinez, D. & Lepolard, C. Brief and long maternal separation in C57Bl6J mice: behavioral consequences for the dam and the offspring. *Psychopharmacology* 214, 71–88 (2011).

de Souza, J. A. et al. Early life stress induced by maternal separation during lactation alters the eating behavior and serotonin system in middle-aged rat female offspring. *Pharmacol. Biochem. Behav.* 198, 172908 (2020).

O'Mahony, S. M., Hyland, N. P., Dinan, T. G. & Cryan, J. F. Maternal separation as a model of brain-gut axis dysfunction. *Psychopharmacology* 214, 71–88 (2011).

Rincel, M. & Darnaudéry, M. Maternal separation in rodents: a journey from gut to brain and nutritional perspectives. *Proc. Nutr. Soc.* 79, 113-132 (2020).

2.4. Why is the reason the administration of prebiotic mixture and *B. longum* continues even after switching to control diet. If the reason looking at early life exposure, shouldn't they also be stopped? It does not necessarily meant early life exposure because the treatment continues to adulthood.

We thank the reviewer for requesting this important clarification regarding our intervention protocol. The "early-life exposure" component of our study refers specifically to the HFHS diet exposure during the critical developmental period, not to the putative probiotic/prebiotic intervention. As now noted in our revised manuscript, our study design maintained the *B. longum* APC1472 and FOS+GOS administration throughout the experimental period to test

the hypothesis that *B. longum* APC1472 or FOS+GOS have the potential to improve early-life HFHS-induced alterations. Therefore, we administered throughout life as this would provide the largest window of opportunity encompassing both neuronal development (prevention) and adulthood (restoration). Whether administration during select periods of life (early-life or post-weaning) also exerts similar effects will need to be investigated in future studies, now that it is evident that life-long consumption has beneficial effects. We included this limitation in the Discussion section of the manuscript (lines 1129-1139).

2.5. Why aren't there any prebiotic+*B. longum* group? What about synbiotic effects of FOS+GOS+ *B. longum*.

We appreciate the reviewer's interest in a combined prebiotic FOS+GOS and *B. longum* APC1472 group. This was a deliberate experimental decision to first establish the individual efficacy and mechanisms of each intervention to improve early-life HFHS-induced alterations before investigating their synergistic effects.

The analysis of the individual interventions allows us to first attribute effects to specific interventions without confounding synergistic interactions. Next, it allows us to investigate the distinct pathways through which each intervention acts independently.

The data from this foundational study will inform the design of future synbiotic investigations, including our planned follow-up investigation that will specifically examine the synbiotic effects of FOS+GOS+ *B. longum* APC1472 combination.

2.6. Was the group housing done randomly in the weaning period or did siblings go to the same cage?

We thank the reviewer for this methodological question. Our randomization occurred at the dam level at postnatal day 0 (PND0), immediately after delivery. Dams were individually and randomly assigned to experimental groups using a simple randomization procedure. Specifically, each dam was allocated sequentially to an experimental group by drawing from a shuffled list of group identifiers (Control Vehicle, HFHS Vehicle, HFHS+FOSG+GOS, HFHS+*B. longum* APC1472). In cases where target sample size was achieved, the next randomly assigned group was selected.

After weaning (PND21), offspring were group-housed by sex with their littermates within treatment groups. Random allocation of treatment groups after weaning was not possible since the offspring were already assigned to their respective treatment group from birth. We have now edited section "2.1. Animals, diets and ethical approval" to clarify the reviewer's questions.

2.7. There will be cage to cage or cohort to cohort difference in the microbiome analysis, was that considered?

In our initial analyses, cage-to-cage and cohort-to-cohort variability were not explicitly modeled as random effects. To address this, we have re-run the statistical analyses using linear mixed-effects models (lmer function in R) that incorporated cage/litter and cohort as random factors.

The variance partitioning from these models indicated that cage and cohort effects contributed modestly to the overall variance across the different comparisons, with residual variance remaining the primary source of variation in most models. Importantly, accounting for these random effects did not substantially alter our main findings regarding diet and treatment effects.

The results for statistical analysis including cage and cohort effects are now available at the following repository: https://github.com/CristinaCuesta-Marti/earlylifeHFHS_eating_microbiota_2025/outputs/overall_stats_lmer and https://github.com/CristinaCuesta-Marti/earlylifeHFHS_eating_microbiota_2025/BINC_PROTECT_16S and we have updated the methods section "2.17.1. 'Omics datasets" accordingly.

2.8. How many dams per group was recruited in the beginning to reach offspring n=12 per group?

We have now updated "2.1. Animals, diets and ethical approval" to clarify that we utilized 7 dams per group to reach the sample size at endpoint. In line 86-102 it states: "*Dams were mated and individually housed at embryonic day 17. At day of birth (PND0), dams were randomly assigned to experimental groups using a simple randomization procedure (7 dams/group, aiming to avoid any cage/litter effects). [...]. After weaning (PND21), offspring were group-housed by sex with their littermates within treatment groups and [...]. At PND35, 5-week-old offspring were euthanized depending on sex and litter size to keep 2-3 mice per litter/sex (on average) to reach the target sample size n=12 per group at 12 weeks of age.*"

3. Authors only address Euclidean distance in beta-diversity. Whereas Euclidean is not enough for compositional data due to its inability to account constant sum. Therefore Aitchison distance that takes log-ratio transformation into account is recommended and this will give more information about prevalence. In addition, weighted unifrac for calculating beta diversities will give information about abundance based on phylogenetic taxa observed. In addition, authors should discuss alpha diversity in the context of species evenness and richness. This is important especially when *B. longum* administration does not increase the number of total bifidobacterium but *B. longum* numbers increase in feces.

We thank the reviewers for the thoughtful comment. We agree with the suitability of Aitchison distance to account for the compositional nature of -omics datasets. Indeed, we used Aitchison's distance which incorporates Euclidian distance over CLR-transformed data in our analysis and we have clarified this in the Methods Section "2.17.1. 'Omics datasets). For a discussion, please refer to this relevant review <https://www.frontiersin.org/journals/microbiology/articles/10.3389/fmicb.2017.02224/full> and the documentation of the R package vegan, which was used to calculate the Aitchison's distance in our article: <https://rdr.io/cran/vegan/man/vegdist.html>.

Regarding alpha diversity, we have included comprehensive analysis of species richness (Chao1 index) and evenness (Shannon and Simpson indexes) in the Results section (lines 649--654), to highlight the diet and intervention effects. We have included in discussion the point regarding the increase of *B. longum* species without impacting in overall

Bifidobacterium genus abundance in feces by the putative probiotic administration (lines 1007-1010).

- 3.1. Authors don't discuss that *B. longum* administration does not necessarily increase total *Bifidobacterium* abundance whereas it increases in *B. longum*. That seems like a potential competition between other *Bifidobacterium* and *B.longum* or *B.longum* found in feces not necessarily colonized in the gut. See my earlier comment on this suggestion.

We have now included additional context regarding the effect of *B. longum* APC1472 administration on microbiome composition, particularly regarding the lack of effect on *Bifidobacterium* genus abundance (lines 985-987 and 1007-1010). We agree that this finding likely reflects strain-level dynamics where the introduced strain competes for specific niches with other strains without dramatically altering overall genus abundance. This is consistent with a (potentially transient) colonization of the gut by *B. longum* APC1472.

It is indeed not surprising that we did not observe differences at genus level upon treatment with a probiotic strain from the same genus. In this line, recent research by Khatib *et al.* demonstrates genus-level stability can mask species- and strain-specific changes when analyzed at higher taxonomic levels (Khatib *et al.*, 2025). Additionally, even within species, at a finer taxonomic level, there can be differential abundance patterns across time and sex. We have now performed Amplicon Sequence Variant (ASV)-level analysis of ASVs within the *B. pseudolongum* species (within *Bifidobacterium* genus) to better illustrate these sub-species dynamics (see figure below). This demonstrates for example, that in control animals ASV2 remains stable in females over time whereas it decreases in males. Contrarily, ASV56 abundance reduces in both sexes over time in control mice. In mice treated with *B. longum* APC1472, neither ASV2 nor ASV56 showed any changes across time or sex.

This analysis reveals that even within a single species, individual ASVs can show differential and sometimes opposing abundance patterns within a group and timepoint, confirming that genus-level measures can mask important species- and strain-level dynamics and competitive interactions within *Bifidobacterium* genus.

Khatib, L. et al. A three-country analysis of the gut microbiome indicates taxon associations with diet vary by taxon resolution and population. *mSystems* 10, e00544-25 (2025).

3.2. Which *Bifidobacterium* species were enriched during FOS+GOS administration?

We thank the reviewer for this important question seeking clarification on the specific *Bifidobacterium* species affected by FOS+GOS treatment.

The murine gut microbiome typically harbors limited *Bifidobacterium*, in both species diversity and abundance (Hugenholtz et al., 2018; Lkhagva et al., 2021; Weldon et al., 2015). The relative abundance of this genus typically ranges from undetectable to 1% of total composition. Among the few *Bifidobacterium* species normally present in mice, *B. pseudolongum* and *B. animalis* are most commonly detected (Shin et al., 2016).

To address the reviewer's question directly, we have now plotted the absolute abundance of all *Bifidobacterium* species identified in our dataset by treatment group, sex, and timepoint (figure below). This analysis reveals three *Bifidobacterium* species in the offspring microbiome: *B. pseudolongum*, *B. longum*, and *B. breve*. *Bifidobacterium breve* was found only sporadically in a small number of pups with almost negligible counts and showed no apparent enrichment with FOS+GOS supplementation, as shown in the figure above.

As shown in the figure, the only species affected by FOS+GOS was *B. pseudolongum*. *B. pseudolongum* was the most abundant species and showed specific enrichment in response to FOS+GOS supplementation. Statistical analysis revealed that early-life HFHS diet significantly reduced *B. pseudolongum* relative abundance in 5-week-old female ($q = 0.011$) and male mice ($q = 0.0002$), effects that did not persist at 10 weeks of age. FOS+GOS administration restored the HFHS-induced reduction in *B. pseudolongum* relative abundance in 5-week-old females ($q = 0.004$) and 5-week-old males ($q = 0.001$). FOS+GOS administration restored the non-significant reduction in *B. pseudolongum* relative abundance in 10-week-old females (diet effect, $q = 0.14$; FOS+GOS restoration, $q = 0.025$). This finding aligns with previous studies demonstrating that FOS and GOS selectively increase the prevalence of this genus in mouse models (Mao *et al.*, 2018; Monteagudo-Mera *et al.*, 2016).

Hugenholtz, F., & de Vos, W. M. (2018). Mouse models for human intestinal microbiota research: a critical evaluation. *Cellular and Molecular Life Sciences*, 75(1), 149-160.

Lkhagva, E., Chung, H. J., Hong, J., Tang, W. H. W., Lee, S. I., Hong, S. T., & Lee, S. (2021). The regional diversity of gut microbiome along the GI tract of male C57BL/6 mice. *BMC microbiology*, 21(1), 44.

Weldon, L., Abolins, S., Lenzi, L., Bourne, C., Riley, E. M., & Viney, M. (2015). The gut microbiota of wild mice. *PLoS One*, 10(8), e0134643.

Shin, J., Lee, S., Go, M. J., Lee, S. Y., Kim, S. C., Lee, C. H., & Cho, B. K. (2016). Analysis of the mouse gut microbiome using full-length 16S rRNA amplicon sequencing. *Scientific reports*, 6(1), 29681.

Mao, B., Gu, J., Li, D., Cui, S., Zhao, J., Zhang, H., & Chen, W. (2018). Effects of different doses of fructooligosaccharides (FOS) on the composition of mice fecal microbiota, especially the *Bifidobacterium* composition. *Nutrients*, 10(8), 1105.

Monteagudo-Mera, A., Arthur, J. C., Jobin, C., Keku, T., Bruno-Barcena, J. M., & Azcarate-Peril, M. A. (2016). High purity galacto-oligosaccharides enhance specific *Bifidobacterium* species and their metabolic activity in the mouse gut microbiome. *Beneficial microbes*, 7(2), 247-264.

4. The conclusion will need to be tailored, that could also be reflected in the graphical abstract. I think that authors did not pay much attention in the discussion about the different effects/mechanisms prebiotic vs *B. longum* and male vs female. Early life HFHS has different effects male vs female and FOS+GOS and *B. longum* attenuates these effects different for male and female mice. The abstract should reflect that finding, which mechanisms were found in female and which are found in male. In male mice, PNOC neuron and Nod2-expressing cells are altered whereas in female mice, POMC neurons and Ghsr and Lepr receptors are affected. “a reduction in the number of cells expressing Ghsr, Lepr and Pomc individually in the ARC of adult early-life HFHS exposed female mice while Ghsr and Pomc were only reduced in adult early-life HFHS exposed males compared to control.

We have substantially revised and updated the discussion (lines 971-989) and conclusion (lines 1105-1112), as well as the abstract and graphical abstract, to explicitly detail these sex-specific effects of early-life HFHS diet and intervention-specific mechanisms of restoration. Specifically, we now highlight that in females, early-life HFHS exposure induced broader hypothalamic vulnerability with reductions in GHSR+, LEPR+, and POMC+ cells in the ARC, alongside extensive disruption of tryptophan metabolism, arginine metabolism, and developmental programming pathways. In contrast, males primarily showed reductions in PNOC+, and NOD2+ cells in the ARC, with disruptions in peptidoglycan sensing pathways and neurotransmitter metabolism.

Furthermore, we now clearly delineate the differential mechanisms of the two interventions (lines 978-989). Whilst there were sex-specific effects, FOS+GOS acted primarily through extensive microbiome compositional shifts (restoring multiple gut-brain modules and SCFA-producing bacteria), while *B. longum* APC1472 induced minimal microbiome compositional changes but achieved robust and lasting improvements in feeding behaviors, which we hypothesize occurs through restoration via targeted metabolic pathway, for instance via tryptophan metabolism in females. These intervention- and sex-specific findings are now highlighted throughout the revised manuscript.

5. Authors should also focus on the mechanisms of FOS+GOS and *B. longum* more clearly. Because they have different effects on the HFHS induced alterations however whether the mechanisms are different is unclear. It seems like FOS+GOS makes shifts in the microbiome composition whereas no clear shifts were observed for *B. longum* however *B. longum* displays more lasting effects. This can be also addressed by looking at metabolomics in fecal or cecal samples.

We thank the reviewer for this insightful observation regarding the distinct mechanisms underlying FOS+GOS versus *B. longum* APC1472 interventions. To further substantiate the differential mechanisms of action between FOS+GOS and *B. longum* APC1472, we have conducted extensive additional analyses of our existing data and conducted SCFA/BCFA analysis in cecal samples (week 5 & 12) (Supplementary Fig. S5). Additionally, we have (1) analyzed the functional capacity of the microbiome using PICRUSt to investigate gut-brain-modules and common pathways of altered/restored metabolites in blood (Supplementary Fig. S7) and (2) integrated hypothalamic transcriptomics and blood metabolomics, that provide more mechanistic clarity for each microbiota-targeted intervention in a sex-specific manner (Supplementary Fig. S8).

FOS+GOS induced substantial microbiome compositional shifts, restoring multiple gut-brain modules (10 in females, 4 in males) and SCFA-producing bacteria, which aligned with broad metabolic pathway restoration. In contrast, *B. longum* APC1472 induced minimal microbiome compositional and functional (1 GBM in both sexes) changes but achieved a greater lasting improvement in feeding behaviors in adult females and males. Our integrative analyses revealed that this likely occurs through targeted restoration of specific metabolic pathways, including tryptophan metabolism and energy homeostasis pathways in adult females, and through DOPAC-dopaminergic pathways and metabolic homeostasis (AMPK signaling, glucagon signaling) in males. We have included these additional analysis in results (lines 718-770 and 841-871) and discussion (lines 971-989).

6. There are some typos listed:

- ✓ Line 36 not symbiotics, synbiotics..
- ✓ Line 644 should be POMC. Typo

We thank the reviewer for identifying these typographical errors. We have corrected both instances.

Reviewer #1 (Remarks on code availability):

the code is not deposited to github

We thank the reviewer for this comment. We have updated the "Code Availability" section of the manuscript to include the GitHub URL where all the code can be accessed. The repository now contains all scripts used for statistical analyses, data processing, visualization and the new integrative multi-omics analyses including the multi-association analyses between blood metabolome and brain transcriptome, and gut microbiome data and blood metabolome.

Reviewer #2

(Remarks to the Author):

This is a well-written paper and a comprehensive study design that included four experimental groups, two microbiota-targeted interventions (prebiotic, probiotic) and two dietary control groups, and multiple analyses (including behavioral, metabolomic, transcriptomic, microbiota profiling, and fluorescent in situ hybridization). The inclusion of both male and female mice and consistent presentation of results for both sexes highlighting sex-specific effects is an important contribution and an often-overlooked factor. The detailed inclusion of supplementary data including body weight and gain in both graph and table format and details of various tissue measures and weights is commendable. The measurement of food intake and food crumbling also adds to the precision of the food intake recording.

The quality of the data appears technically sound and is presented in detail. The conclusions made are not excessive and are supported by the evidence provided. These findings are relevant and add to the evidence for early life effects of diet on later food intake behaviors in mice while the sex differences shown highlight the importance of including both male and female mice and reporting their data separately.

Comments

There are some points that could improve the manuscript:

1. The control diet used is detailed as “D12451 modified (Product N S9912-E760” but the modifications made to the D12451 diet are not clear. Can these modifications be clarified.

We thank the reviewer for this important clarification request. We have now clearly specified the modifications that were made to the D12451 diet (lines 92-93).

The HFHS diet used in our study was a modified version of the D12451 diet, with specific compositional changes requested to achieve our desired macronutrient profile. According to the manufacturer (Ssniff Spezialdiäten GmbH, Germany), the modifications made to the standard D12451 formulation included a reduction of sucrose from 20.17% to 17% (w/w) and a corresponding increase in corn starch from 7% to 10.17% (w/w) to compensate for the sucrose reduction.

The 17% sucrose content was specifically selected to align with established HFHS diet formulations widely used in the literature investigating metabolic programming and obesity-related outcomes. This composition has been validated in numerous studies examining early-life dietary effects, including investigations of metabolic dysfunction (Mishra et al., 2022; Pennington et al. 2017), gut microbiome alterations and feeding behavior (de Wouters d'Oplinter, et al. 2021, 2023).

We have also corrected the diet description terminology in our methods section “2.1. Animals, diets and ethical approval” (lines 90-91) to specify "kcal%" rather than "%" when referring to the energy contribution from macronutrients, as this more accurately reflects the compositional data presented in Supplementary Table 1.

Pennington, K. A., et al. Effects of acute exposure to a high-fat, high-sucrose diet on gestational glucose tolerance and subsequent maternal health in mice. Biol. Reprod. 96, 435–445 (2017).

Mishra, A., et al. A. A novel model of gestational diabetes: Acute high-fat, high-sugar diet results in insulin resistance and beta cell dysfunction during pregnancy in mice. *PLoS One* 17, e0279041 (2022).

de Wouters d'Oplinter, A., et al. Gut microbes participate in food preference alterations during obesity. *Gut Microbes* 13, 1959242 (2021).

de Wouters d'Oplinter, A., et al. A. Obese-associated gut microbes and derived phenolic metabolite as mediators of excessive motivation for food reward. *Microbiome* 11, 94 (2023).

2. How was the probiotic dosing decided upon? The dosage used in drinking water is based upon that used in previous research and adapted for the offspring but how was this adapted for the offspring?

We thank the reviewer for this important question regarding our probiotic dosing strategy.

The probiotic dosing regimen was designed using body weight scaling principles alongside water consumption patterns, with practical considerations for long-term study implementation. The maternal dose ($\sim 1.0 \times 10^9$ CFU/animal/day for dams weighing an average of 29g dams) and offspring dose ($\sim 5.0 \times 10^8$ CFU/animal/day) were calculated based on both body weight scaling and average daily water intake to provide comparable exposure per gram body weight when pups began direct supplementation of the bacterial strain at 5-10g.

These doses were selected based on previous studies from our own laboratory and others in which a dose of 10^8 - 10^9 CFU/day induced both metabolic and neurobehavioral effects in rodents (Carasi *et al.*, 2015; Mariman *et al.*, 2015; Bai *et al.*, 2016; Wang *et al.*, 2012). Notably, our group has previously demonstrated that the same *Bifidobacterium longum* strain (APC1472) improves metabolic health in HFD-fed mice (at $\sim 2 \times 10^8$ CFU/mL in drinking water) as well as in overweight and obese humans (at 1×10^{10} CFU/day) (Schellekens *et al.*, 2021). Furthermore, multiple probiotic strains administered at comparable doses (10^8 - 10^{10} CFU/day) have shown beneficial effects on metabolic parameters in both rodent models and in humans (Patterson *et al.*, 2019; Schellekens *et al.*, 2021; Gomes *et al.*, 2017; Chang *et al.*, 2011; Bernini *et al.*, 2016).

For offspring, we maintained a fixed-dose approach as is commonly used in preclinical studies. Since water intake mainly scales with body weight in growing mice, we modified accordingly the amount to drinking water in which the inoculum was resuspended to maintain a fixed dose (CFU). We have now clarified our strategy for probiotic dose administration in methods section "2.2.2. In vivo probiotic administration" (lines 166-167).

Schellekens, H. *et al.* *Bifidobacterium longum* counters the effects of obesity: Partial successful translation from rodent to human. *EBioMedicine* 63, 103176 (2021).

Patterson, E. *et al.* Gamma-aminobutyric acid-producing lactobacilli positively affect metabolism and depressive-like behaviour in a mouse model of metabolic syndrome. *Sci. Rep.* 9, 16323 (2019).

Gomes, A. C. *et al.* The additional effects of a probiotic mix on abdominal adiposity and antioxidant Status: A double-blind, randomized trial. *Obesity* 25, 30–38 (2017).

Chang, B. J. *et al.* Effect of functional yogurt NY-YP901 in improving the trait of metabolic syndrome. *Eur. J. Clin. Nutr.* 65, 1250–1255 (2011).

Bernini, L. J. et al. Beneficial effects of Bifidobacterium lactis on lipid profile and cytokines in patients with metabolic syndrome: A randomized trial. Effects of probiotics on metabolic syndrome. Nutrition 32, 716–719 (2016).

3. The methods state that mice were distributed evenly across experimental groups. How were these allocations decided and was any randomisation method used? If so please state the method used to allocate mice into experimental groups.

We thank the reviewer for this important methodological question. We have updated the Methods section “2.1 Animals, diets and ethical approval”, to clearly describe our randomization procedure (lines 86-102).

The section now states: "Dams were mated and individually housed at embryonic day 17. At day of birth (PND0), dams were randomly assigned to experimental groups using a simple randomization procedure (7 dams/group, aiming to avoid any cage/litter effects). [...]. After weaning (PND21), offspring were group-housed by sex with their littermates within treatment groups and [...]."

4. Are the long-term changes observed in the arcuate nucleus relevant to later control diets? This point could be mentioned in the discussion, that the changes in responses to food later in the study appeared to be only in response to further experimental tests and that body weight normalized in all groups when on control diet in the later part of the study period despite earlier exposures.

We thank the reviewer for this important point. We did not explicitly test the effect of early-life HFHS diet on food intake (chow) in behavioral tests (PheCOMP). However, our food intake data (daily chow intake in home cage) in adulthood suggests that there were no major differences prior to reintroduction of the HFHS diet. As can be seen in main Figure 1C, early-life HFHS exposed males overall tended to eat less chow than control vehicle mice between week 8 and 10 of age (adulthood). However, there was no significant difference in daily intake on any specific timepoint, nor any microbiota-intervention effect.

We have now included a sentence in the discussion (lines 963-965) to clarify that the molecular changes observed in the arcuate nucleus and the altered feeding behaviors in adulthood occur in the absence of body weight differences between groups. As the reviewer correctly noted, body weight normalized in all groups when switched and maintained on control diet in the later part of the study period, despite the earlier dietary exposures.

5. Does the FOS/GOS in drinking water remain adequately dissolved and suspended during the time the water is supplied? Has this been measured in a similar manner to the conformation of the survival of the probiotic in the drinking water to ensure consistent dosage?

FOS (fructooligosaccharides) and GOS (galactooligosaccharides) are both readily soluble in water, which ensures consistent dissolution and homogeneous distribution throughout the drinking water.

To ensure consistent dosage delivery, we implemented strict preparation and replacement protocols. The prebiotic solution was prepared weekly under sterile conditions, aliquoted and stored at 4°C to maintain stability. Water bottles with FOS+GOS were replenished daily

to ensure freshness and were fully changed/renewed regularly (every 2-3 days maximum) to prevent any potential degradation or contamination.

These protocols ensure that the FOS+GOS concentration and dosing remained consistent throughout the study period. The high solubility of both prebiotics minimizes the potential for precipitation or uneven distribution that can be associated with less soluble compounds, thereby supporting consistent dosing across all animals. We have updated Section 2.2.1. *In vivo* prebiotic administration in Methods to clarify this methodology.

6. Was the amount of water the mice drink checked and was this used to confirm that the mice were receiving the expected dose of the FOS/GOS or probiotic?

We thank the reviewer for this important question. Daily drink intake of the mice was monitored throughout the study to confirm that they were receiving the expected doses of FOS+GOS and *B. longum* APC1472. We have now included this data in Supplementary Fig. S1B and show the graph below in this rebuttal document, displaying the daily drink intake by group and sex, with measurements recorded during both light and dark phases. As can be seen, mice readily consumed water throughout the study regardless of treatment. If anything, mice tended to drink more when the drinking water contain *B. longum* APC1472 (both sexes). For this reason, the amount of water containing inoculum was adapted to the consumption during dark phase to ensure mice had enough water and received the desired dose throughout the study. Based on the average drink intake of females and males and a concentration of 7.5 g/L of FOS+GOS, females received 36 mg and males 43.5 mg of FOS+GOS across the entire day. Statistical analysis for drink intake can be found in https://github.com/CristinaCuesta-Marti/earlylifeHFHS_eating_microbiota_2025/outputs/overall_stats.

7. Mice pups appeared to show increased body weight gain during the suckling period when dams were fed High-fat/High-sugar diet. Is there a potential explanation for this effect through the milk given the same control diet was used during pregnancy?

We thank the reviewer for this important observation. To clarify, the dams were maintained on standard chow diet during the entire pregnancy. The dietary interventions (control versus high-fat/high-sugar diet) started at birth (PND0). The increased body weight gain observed in pups during the suckling period is most likely explained by metabolic effects transmitted through the maternal milk composition.

Because maternal diet is reflected in breastmilk (Aumeistere, L. *et al.*, 2019; Ellsworth, L. *et al.*, 2020), we assume that these metabolic effects in offspring are transmitted through this pathway. However, we were unable to collect milk due to our study design (dams were euthanized on PND23) and therefore, could not analyze its composition to conclusively examine this.

Aumeistere, L. *et al.* *Impact of maternal diet on human milk composition among lactating women in Latvia. Nutrients* 11, 1216 (2019).

Ellsworth, L. *et al.* *Impact of maternal overweight and obesity on milk composition and infant growth. Matern. Child Nutr.* 16, e12979 (2020).

8. In supplementary figure 1D the food and energy intake tend to be higher in High-fat/High-sugar diet fed mice given FOS/GOS in their drinking water. Are there any suggested reasons for this?

We thank the reviewer for this important observation regarding the food and energy intake patterns in Supplementary Figure 1D (main Fig. 1C and Supplementary Fig. 1G now). While the HFHS diet-fed mice administered with FOS+GOS showed higher food intake on certain punctual days during early-life compared to HFHS Vehicle females, this was not consistently higher compared to other HFHS groups throughout the study period.

The apparent higher food intake observed on specific days in the FOS+GOS group may be attributed to increased food crumbling behavior rather than genuine increased consumption. We observed that the FOS+GOS group exhibited more food crumbling (also observed during the food preference test, Figure 1D now), which could lead to overestimation of real food intake when measured by food removal from the food hoppers on those particular days.

9. The body weight gain appears higher in mice supplemented higher in FOS/GOS? Is this due to increased cecum weight and intestinal length? This could be included as a discussion point as this physiological effect of fermentable fiber in mice is quite significant as this difference (Supplementary Figure 2A) appeared to persist after the mice were returned to the control diet.

We thank the reviewer for this observation.

FOS+GOS group indeed showed slightly higher body weight compared to the other groups both the period where they were exposed to the HFHS diet and especially later in life when they were switched to the control diet. Since metabolic markers were not negatively impacted by FOS+GOS administration and we observed increased caecum weight and colon/small intestine length (when not normalized to body weight), we ascribed this weight difference to adaptive changes associated with increased fermentable fiber intake, as also supported by previous literature (Bryk, G. *et al.*, 2015; Pan, X. *et al.*, 2009). We highlighted this observation now in results (lines 639-642) where we believe it fits more appropriately than in the Discussion.

Bryk, G. *et al.* *Effect of a combination GOS/FOS® prebiotic mixture and interaction with calcium intake on mineral absorption and bone parameters in growing rats. Eur. J. Nutr.* 54, 913-923 (2015).

Pan, X. *et al.* *Prebiotic oligosaccharides change the concentrations of short-chain fatty acids and the microbial population of mouse bowel. J. Zhejiang Univ. Sci. B* 10, 258-263 (2009).

10. It is stated that “early-life HFHS diet reduces the abundance of *Bifidobacterium* genus” and that “restored by FOS+GOS administration”. But it is not mentioned that the HFHS diet is deficient in any fermentable fiber that can be metabolized by *Bifidobacterium*. While the D12451 control diet is lacking in added fiber it contains a lot of corn starch that can act as resistant starch and is a fermentable carbohydrate to *Bifidobacterium*. It could be mentioned that adding FOS+GOS to the diet is restoring the loss of any fermentable carbohydrate in the HFHS diet.

We thank the reviewer for this insightful observation regarding the fermentable carbohydrate content of the diets.

HFHS diet does indeed have significantly lower amount of fermentable substrates. Control diet (D12450K) contained 48.85% corn starch, while the HFHS diet (D12451) had only 10.17% (Supplementary Table S1). In addition to the reduction of corn starch, HFHS diet had reduced total carbohydrate content to accommodate the increase in fat content. Therefore, both of these factors may contribute to the reduction of bacteria like *Bifidobacterium* that use fermentable fiber in addition to other carbohydrates. FOS+GOS supplementation may increase not only fermentable fiber but also overall carbohydrate content of the diet.

We have now included this important point in our results section (lines 669-671).

Reviewer #3

(Remarks to the Author):

The paper investigates the effects of early exposure to a high fat high sugar diet on feeding behavior, blood and brain neurochemistry. The potential therapeutic effect of microbiota-based interventions during and after diet exposure are also tested. These microbiota-targeted interventions improved feeding behavior, brain neurochemistry and restore some altered blood metabolomics parameters.

Although it is unclear if changes are prevented by the interventions during diet exposure or if restoration / normalization happen after diet exposure. Similarly, the dietary switch happened at PND35, blurring the distinction between early-life exposure and post-weaning /adolescence exposure.

We thank the reviewer for this important comment regarding the distinction between protective versus restorative effects of our microbiota-targeted interventions.

We acknowledge that the dietary switch at PND35 encompasses both the early-life lactation period (birth through weaning at PND21) and the adolescent period (post-weaning through PND35). We selected this design to capture the entire critical developmental window during which dietary programming effects may occur. This also reflects the human reality in which children born to parents that consume an unhealthy diet will follow generally this until early adulthood.

Regarding, the restoration/normalization intervention effects, we agree that our study design encompassing microbiome-targeting interventions throughout life makes it challenging to distinguish prevention from restoration. We appreciate that the terminology used for our statistical analysis (restoration effect) for cases where microbiome-targeted interventions significantly attenuated the early-life HFHS-induced alterations may be misleading in regards to the suggested mechanisms. We have therefore acknowledged this in the Discussion section (lines 1130-1134) and clarified our terminology regarding the statistical analysis in Section "2.17. Bioinformatics and statistical analysis".

Additional comments:

English needs to thoroughly edited in the abstract. We have now revised and updated the abstract.

The introduction gives a broad overview of the homeostatic and hedonic systems involved in regulating feeding, a more focused introduction on the perinatal influence and the role of the microbiota in modulating both systems (and how they are connected) would be more appropriate.

We acknowledge the reviewer's suggestion for a more focused introduction on how the perinatal microbiota influences feeding systems. While extensive research demonstrates that gut microbiota modulates both homeostatic and hedonic feeding in *adulthood* and that the microbiota has a critical role in host early-life development, there is currently limited direct evidence specifically addressing how early-life microbiota influences homeostatic and hedonic feeding systems. This important knowledge gap was the key rationale for our study.

We have highlighted this knowledge gap in the introduction while acknowledging the current evidence from adult studies (lines 62-65).

The rationale for focusing on the hypothalamus is not well justified as microbiota has been shown to influence vagal /NTS signaling in the homeostatic regulation of feeding and the references provided show effect of the reward areas (NAc and VTA), which are also under vagal influence. Data presented in the supp materials also show a strong effect on meal size, again more a vagal /NTS regulated behavior. Maternal studies have shown strong effect on hypothalamic factors but these are not cited in the manuscript.

We thank the reviewer for this important point. We acknowledge that the microbiota influences feeding through multiple interconnected pathways, including vagal-NTS signaling (which regulates meal size as observed in our supplementary data) and reward circuits that are also under vagal influence. We have updated our introduction to better justify our focus on hypothalamic markers by citing studies that demonstrate direct programming effects of maternal diet on hypothalamic neuropeptides during early development (lines 28-30). We have added the need for future studies to examine vagal-NTS pathways and brainstem nuclei to complement our hypothalamic findings and provide a more comprehensive mechanistic understanding of microbiota-mediated feeding control in our discussion (lines 1134-1139). Additionally, we would like to highlight that we looked into dopamine, serotonin and noradrenalin (and their metabolites) in the striatum, a brain region involved in food reward, and found minor non-significant long-term effects of early-life HFHS diet (Supplementary Figure S2A).

Fig1

1. Text refers to figure 2 panels that are actually figure 1.

We thank the reviewer for this observation. We apologize for the oversight and we have corrected it now.

2. Preference for saccharin may be related to different threshold in sweet perception, as HF feeding has been shown to alter sweet testing, this should be mentioned

We thank the reviewer for this excellent point. We agree that the preference for saccharin observed in our study may indeed be related to alterations in sweet taste perception thresholds. Previous research has demonstrated that high-fat feeding as well as other unhealthy diets can significantly alter sweet taste sensitivity, and perception, potentially leading to changes in preference for sweet-tasting substances like saccharin (Choo *et al.*, 2022; Bayol *et al.* 2007; Maliphol *et al.* 2013).

We have now included this important consideration in our discussion (lines 997-1003), as it provides valuable context for interpreting the saccharin preference test results and highlights how early-life dietary exposures can have enduring effects on taste and sweet perception and feeding preferences that extend beyond the intervention period.

Choo, E., Wong, L., Chau, P., Bushnell, J. & Dando, R. Offspring of obese mice display enhanced intake and sensitivity for palatable stimuli, with altered expression of taste signaling elements. *Sci. Rep.* 10, 12776 (2020).

Bayol, S. A., Farrington, S. J. & Stickland, N. C. A maternal 'junk food' diet in pregnancy and lactation promotes an exacerbated taste for 'junk food' and a greater propensity for obesity in rat offspring. *Br. J. Nutr.* 98, 843–851 (2007).

Maliphol, A. B., Garth, D. J. & Medler, K. F. Diet-induced obesity reduces the responsiveness of the peripheral taste receptor cells. *PLoS One* 8, e79403 (2013).

3. Does food crumbling directly correlate to food intake? If this is the case, the observed reduction in crumbling may be result of the overall reduction in intake. It may also be reflective of exploratory behavior or stress

We thank the reviewer for this question regarding the relationship between food crumbling and food intake as well as the potential contribution of exploratory behavior and stress.

We have conducted correlation analyses to examine the direct relationship between food crumbling and food intake during the PheComp behavioral test at week 10, which allows measurable separation of food crumbling from food intake due to the specialized cage design without bedding interference. The overall correlation analysis revealed a moderate positive correlation for both sexes, with a slightly stronger relationship between crumbling and intake behaviors in males ($r = 0.53$, $p < 0.001$) compared to females ($r = 0.44$, $p = 0.002$), suggesting that food intake and food crumbling are indeed associated. Interestingly, control groups (not exposed to HFHS in early-life) and *B. longum* APC1472 treated HFHS-animals showed non-significant correlations (control females: $r = -0.58$, $p = 0.062$; control males: $r = -0.14$, $p = 0.693$, *B. longum* APC1472 females: $r = 0.41$, $p = 0.163$, *B. longum* APC1472 males: $r = 0.22$, $p = 0.571$). This is in line with these two groups overall consuming less food and displaying less food crumbling behavior (Fig. 1E, F).

Regarding relation to exploratory behavior, we found no evidence for increased exploratory behavior in adult male and female mice as measured by overall locomotor activity in the PheComp cages (Supplementary Fig. S2B). Regarding stress, we found that early-life HFHS exposure did not increase either corticosterone in the blood or noradrenaline levels in the striatum (Supplementary Fig. S4I, Fig. S2A) in both sexes. In fact, both were reduced in adult males. Regarding the potential influence of anxiety-like behavior, as this has previously been linked to stress and food crumbling behavior (Ang et al., 2024; Cameron & Speakman, 2010), we observed that, only in male mice, early-life HFHS diet increased the time spent in the external zone (10%) as shown in the figure below. Importantly, stress- and anxiety-related measures were exclusively observed in males while it was females who displayed significantly increased food crumbling and food intake after early-life HFHS exposure. Thus, overall these findings support the conclusion that food crumbling is primarily feeding-related and not stress- or exploration-related, especially in females.

Cameron, K. M. & Speakman, J. R. The extent and function of 'food grinding' in the laboratory mouse (*Mus musculus*). *Lab Anim.* 44, 298–304 (2010).

Tang, H., et al. Food Grinding Behavior: A Review of Causality and Influential Factors. *Animals (Basel)* 14, 1865 (2024).

- Results state that control and treated group have a higher preference for saccharin (males), this appears to be contradictory to the data displayed on the graph?

We thank the reviewer for pointing out this inconsistency in our manuscript. We have now clarified the sentence to accurately reflect our data, which shows that early-life HFHS exposure increased saccharin preference in adult male mice, and that both microbiota-targeted interventions (prebiotic FOS+GOS and *B. longum* APC1472) normalized this elevated preference back to control levels. The corrected sentence now properly aligns with the data presented in the corresponding figure (lines 592-596).

- The offspring are only exposed to the HFHS diet for a limited period of time, after which their body weight normalized. This is a key piece of data as it supports that observed changes in adulthood are not due to differences in food intake or adiposity. Body weight and food intake data should be included in the main results.

We thank the reviewer for this important observation. We agree that the normalization of body weight following the early-life HFHS exposure period is a crucial finding that supports our conclusion that the observed changes in adulthood are independent of differences in food intake or adiposity. In response to this comment, we have moved the body weight and food intake data from the supplementary materials to the main results and included it as part of Figure 1 (Figure 1B-1C) to emphasize the significance of this finding for interpreting our adult behavioral and molecular outcomes.

- It is still unclear however whether the effects observed are due to 1) changes in milk composition during lactation or 2) changes in feeding during that period or 3) effects post-weaning. This limitation should be addressed.

We thank the reviewer for raising this important point regarding the mechanistic distinction between maternal milk-mediated effects versus direct consumption effects (either during lactation or post-weaning). We have included a statement acknowledging this limitation in the discussion section (lines 1120-1133). We would like to highlight that our study was specifically designed to investigate the complex dietary effects in early-life, a period that encompasses both lactation and post-weaning periods. Additionally, collecting milk from dams was not possible since they were euthanized after weaning (PND23) in order to not interfere with normal maternal care. This has been addressed more in detail in response to Reviewer #1 point 2.1 and 2.2.

7. Similarly, despite the interventions having beneficial effects on hypothalamic transcriptomics, the animals appear to overeat when exposed to the HFHS diet. This should be discussed, it is unclear whether the changes are prevented by the interventions or reversed post- diet exposure.

We thank the reviewer for this important observation regarding overeating despite “normalization” of hypothalamic transcriptomics. We would like to highlight that despite some genes and pathways being restored by our microbiota interventions in a sex-specific manner, these do not include *all* genes altered by early-life HFHS diet (as quantitatively shown in Fig 4C, D). In addition, other brain regions might also play a role.

Regarding the potential mechanism of action (prevention and/or restoration) of our microbiome-targeting interventions, we acknowledge that our study design does not allow for distinguishing prevention of initial programming defects, active reversal of established developmental alterations, or a combination of both mechanisms. We also acknowledge that the terminology used for our statistical analysis (restoration effect) for cases where microbiome-targeted interventions significantly attenuated the early-life HFHS-induced alterations may be misleading in regards to the suggested mechanisms. However, this is not intended to suggest a mechanism of action but rather refer to the directionality of the change. This statement has now been included in the discussion (lines 1122-1134).

Overall our objective was to assess whether FOS+GOS or *B. longum* APC1472 can improve early-life HFHS-induced alterations in feeding, including overeating. We therefore tested continuous administration from PND0 through week 12 as the largest window of opportunity encompassing both neuronal development (prevention) and adulthood (restoration). Whether administration during specific periods (early-life only or post-weaning only) produces different effects requires future investigation.

Minor

- Line 11: References are different format.
We have corrected the reference formatting to ensure consistency throughout the manuscript according to the journal's style guidelines.
- Line 30: Vagal afferents do not terminate in proximity to the median eminence or the hypothalamus. Ref 42 focuses on the NAc and is out of place here.
We thank the reviewer for this important correction. We acknowledge our original wording was unclear and did not intend to suggest direct vagal termination in the hypothalamus. We have revised and updated this statement to accurately reflect the distinct pathways through which the gut microbiota can modulate hypothalamic function (lines 32-34).
We have also updated the references and particularly removed Ref 42 (now 47) in the context of hypothalamic involvement.
- Line 30: Implication of the microbiota in reward driven feeding should include Kim *et al.* Molecular Metabolism, 2023.
We appreciate the reviewer's suggestion to include the Kim et al. (2023) Molecular Metabolism reference regarding microbiota involvement in reward-driven feeding. We have incorporated this reference in line 36.

- Supp: Food intake should be presented in kcal not grams as diets have different kcal density.

We thank the reviewer for these important methodological point regarding food intake presentation.

We chose to display food intake measurements in grams in the main manuscript (Fig 1C), and the energy intake in Kcal in the supplementary (Supplementary Fig. S1G), because it more clearly represents the combined concept of both actual food intake and food crumbling behavior, which is methodologically important for interpreting our behavioral findings. We acknowledge that energy intake accounts for caloric content of the diet and would allow a more direct comparison and potential metabolic interpretation of HFHS diet and microbiome-targeting interventions. However, given the caveat of increased food crumbling in the home cage, we felt that “removal of grams from the food hopper” more accurately reflected our observation.

- Was any power analysis conducted ? Omics outcomes display high variances, and the groups may be underpowered.

A comprehensive power analysis was conducted prior to study initiation as part of our ethics application using G*Power 3.1.9.4 (effect size of = 0.43318, 80% power ($1-\beta$ error) and $\alpha = 0.05$ and accounting for litter effects and intra-cluster correlation (ICC = 0.05)). The effect size was chosen to calculate the required samples size to obtain significant effects in feeding behaviors and 16S microbiota analyses as our primary outcomes. This yielded a sample size of $n=12/\text{group}/\text{sex}$.

We acknowledge that RNA sequencing, and blood metabolomics, with $n=6/\text{group}/\text{sex}$ (from 4-6 different litters) may exhibit high variability which is also inherent to such datasets. The sample size for these two omics datasets was lower since endpoint samples were divided into two sets for further (non-primary) analysis: one for perfusion (RNAscope analyses) and one for fresh tissue collection. Nevertheless, $n=6/\text{group}/\text{sex}$ represents a reasonable sample size for omics studies (RNA sequencing and metabolomics). We have added this power analysis information to the Methods section (2.1. Animals, diets and ethical approval, lines 84-86).

Reviewer #4

(Remarks to the Author):

This paper explores the effects of early-life high-fat/high-sugar (HF/HS) diet and microbiota modulations by a prebiotic mix or putative probiotic on later life feeding behavior, microbiota composition, and transcriptomic differences in the arcuate nucleus of the hypothalamus. Additionally, it explores the differences in male and female offspring showing that the effect of early-life HF/HS diet is sex dependent. The manuscript adds interesting results to the existing literature about early-life diet alterations impact on later life but addresses it with the male/female difference point of view (which is often left out).

The methodology is clearly explained and stated, to the exception of a few minor details, and allow reproducibility of the experiments conducted. The graphs are clear for the most part and the graphical abstract/protocol visual description help with the comprehension.

I enjoyed the statistical approach taken (difference between the two diet – vehicle groups followed by measure of the restoration of the parameter by the microbiota intervention), it was a nice way of analyzing the data, that should be used more often.

The introduction and discussion places the article in context with good reference to current literature. These two sections are very clear and nice to read. However, some significant changes are required before publication are listed below. Overall, a re-organization of the figures and a proper labeling of them throughout the manuscript are a needed change as it was difficult to match the figure number with the number in the text.

Overall comment

1. Reorder the figures and supplementary figures to make them appear in a logical order either in the text or in the figure files. Having them so disordered makes it hard to follow the manuscript. Additionally in the text, some figures are misnumbered (ex: Fig 2B (line 468) is actually Fig 1B).

We thank the reviewer for this important feedback regarding the organization and referencing of our figures. We have now reordered all figures and supplementary figures to appear in a logical sequence that corresponds to their appearance in the text. Additionally, we have carefully reviewed the entire manuscript to correct all figure misnumbering errors, including the example noted where Fig 2B (line 468 in the original manuscript) was incorrectly referenced and should have been Fig 1B. We have ensured that all figure citations now accurately correspond to the correct figure panels and that the figure numbering follows a consistent and logical progression throughout the manuscript.

2. Statistics need to be described in a fairer and clearer way. While the statistical tests are really adapted to represent the restoration by both interventions, sometimes the changes induced by HFHS vehicle are not significant compared to the control vehicle group, but the restoration effect is significant. While this non significance doesn't take anything away from the trend/biological effect, it is sometimes stated clearly and sometimes not in the text. A more consistent depiction of the statistics would benefit

the manuscript. For example: o line 490 “Interestingly, both the prebiotic FOS+GOS and *B. longum* APC1472 restored the higher 491 preference towards saccharin in adult male mice that were exposed to an early-life HFHS”, the difference between the control vehicle group and the HFHS vehicle group is not significant ($p=0.12$) although the restoration effect by both microbiota interventions are.

We thank the reviewer for this important comment regarding the statistical presentation in our manuscript. We have now ensured that throughout the manuscript, when the diet effect did not reach statistical significance (but trended) and the interventions significantly altered those effects in the opposite direction (using orthogonal contrast), we explicitly state the non-significant diet effect to provide transparency and clarity. We rephrased these cases to avoid the word “restoration effect”, see for example the sentence highlighted by the reviewer (regarding saccharin preference, lines 592-596 now).

To clarify, we used two complementary approaches to statistically analyze our data: (1) ANOVA followed by Tukey's post-hoc tests to assess diet effects (Control Vehicle vs HFHS Vehicle; while considering the intervention), followed by orthogonal contrasts in a generalized linear model to specifically test the restoration effects of each microbiota-targeted intervention. (2) For omics datasets, diet effects were assessed using t-tests between Control Vehicle and HFHS Vehicle groups, while restoration effects were evaluated by comparing HFHS Vehicle against each intervention group (HFHS FOS+GOS or HFHS *B. longum* APC1472), with restoration defined as cases where both diet and intervention effects were significant with opposing effect sizes.

These different statistical approaches inherently possess different statistical power: orthogonal contrasts are designed to test specific, pre-planned directional hypotheses and thus, have greater sensitivity than the more conservative pairwise comparisons with multiple testing adjustments used in ANOVA/Tukey's tests. Consequently, it is statistically appropriate and not uncommon for restoration effects (assessed via orthogonal contrasts) to reach significance even when the initial diet effect (assessed via ANOVA/Tukey's) does not, as these tests address different questions with different levels of statistical power.

3. Abstract:

The abstract could benefit from breaking down some of the sentences into multiple sentences as it is hard to follow as it is written now.

We have now revised the abstract by breaking down the longer, complex sentences into multiple shorter sentences to improve readability and flow. We have also updated the abstract after new analysis performed after revisions.

4. Material and methods

a. “Animals, diets and ethical approval”

i. Indicate how many dams were housed per cage.

We have clarified this methodological section (lines 86-89). It now reads: *Dams were mated and individually housed at embryonic day 17. At day of birth (PND0), dams were randomly assigned to experimental groups using a simple randomization procedure (7 dams/group, aiding to avoid any cage/litter effects).*

- ii. Indicate if inside of a same treatment group offspring were mixed after weaning to avoid cage effect.

We have clarified this methodological section, which now reads (lines 97-102): *After weaning (PND21), offspring were group-housed by sex with their littermates within treatment groups and [...].*

Random allocation of offspring between litter was not feasible due to differences in pregnancy and birth dates. Further, combining pups from different litter may have resulted in fighting, especially in males.

- iii. Figure 1A and S1B are both very clear and helpful but redundant, chose one to keep in the manuscript.

We have kept Figure 1A, and S1B has been removed to eliminate redundancy.

b. Bioinformatics & statistical analysis

- i. "Additional random effects (including 426 litter and cohort effects) were assessed using linear mixed-effects models using the lmer function in R" - I couldn't find the results of these tests in document S2. If it is not, please add it, if it actually is contained in this file, please make it clearer so the reader can find the information.

We apologize for the oversight. The results of the linear mixed-effects models assessing litter and cohort effects were not contained in document S2. We maintained the results with original statistical analysis using linear model but included a document in https://github.com/CristinaCuesta-Marti/earlylifeHFHS_eating_microbiota_2025/outputs/overall_stats_lmer with the statistical outcomes using linear mixed-effects models to include the random effects (litter and cohort effects), as stated in our updated methods section "2.17. Bioinformatics and statistical analysis". We have separated these analyses from our main statistical results (which are found at https://github.com/CristinaCuesta-Marti/earlylifeHFHS_eating_microbiota_2025/outputs/overall_stats) to provide clear distinction between our primary linear model analyses and the additional random effects assessments. Importantly, the inclusion of random effects (litter and cohort) did not alter the pattern of statistical significance for our main findings, though in several cases the statistical power was enhanced, resulting in more robust effect estimates.

c. Results

- i. Please add a sentence to describe Fig S1 E and F

We have now added a sentence to describe Fig S1 E and F (lines 546-551).

- ii. Line 459 "Interestingly, we found that early-life HFHS-exposed offspring did not consume all food removed from the food hopper but crumbled part of it" - - Please show data for this claim if possible as the graphs represent cumulative consumption and crumbling behavior.

The observation mentioned in line 459 refers to behavioral observations made in the home cage environment, which did not allow us to distinguish between actual food consumption and food crumbling behavior in the daily food intake measurements. To further investigate

this phenomenon more systematically, we analyzed real food intake and food crumbling behavior separately using the PheCOMP cages during a 12-hour light phase period, with results presented in main Figure 1E and 1F (after reordering) respectively. These data provide quantitative support for the initial observation. We also have photographic documentation of the food hoppers showing the food pellets and crumbs that can be provided if the reviewer would find this additional visual evidence helpful.

- iii. Fig S4B: harmonize the serotonin names between the text and the figures + discuss the 9 graphs that are not mentioned in the text

We have now harmonized the serotonin names between the text and figures, and have included a description of the 9 graphs that were previously not mentioned in the Results section (lines 598-611).

- iv. Line 503 to 512, please re-write this section to make the differences between the IpGTT results in male and female clearer.

We have now rewritten the section from lines 503 to 512 to make the differences between the IpGTT results in male and female mice clearer and more accessible to readers (lines 618-630) and we have updated them accordingly in Supplementary Fig. S2C-S2E.

- v. Line 511, for the basal glucose graph description, “restored” has been used in the rest of the manuscript and should be used here as well to keep the results clear and consistent as it is going back to control vehicle group levels. For the males, please add additional statistics and discussion of the effect of FOS+GOS vs control vehicle on basal glucose.

We have chosen the word “significantly reduced” in lieu of “restored” in this case to maintain clarity and consistency throughout the manuscript (as requested by the reviewer in point 2) when diet-effects did not reach significance. Additionally, for this reason, we have removed reference to FOS+GOS effects on basal glucose in adult males. In this case, diet effect was not indicating a trend.

- vi. Line 517: what about the other organs in Fig S5?

We have now included reference to the other organs shown in Fig S5 that were missing from line 517 (lines 633-642).

- vii. Line 522 to 536: please mention the alpha-diversity results.

We have now included mention to the alpha-diversity results in the section from lines 522 to 536 (lines 649-654).

- viii. FigS3D to I are not described

We have now included mention to Fig. S3D in lines 654-664.

- ix. Line 577, there is no mention on the effect/absence of effect of *B. longum* APC1472 on the males' metabolome.

We apologize for not including the effect of *B. longum* APC1472 on the males' metabolome in line 577. We have now included this information to provide a complete picture of the metabolomic effects in both sexes (lines 703-704).

- x. For the section “Early-life HFHS has an enduring and sex-specific impact on hypothalamic transcriptomic profile in adult mice”, it would be interesting the mention the gene commonly restored by both male and female FOS+GOS supplementation in fig.3D.

We have now included information about the genes that are commonly restored by FOS+GOS supplementation (*Hcar1*) and *B. longum* APC1472 supplementation (*Fen1* and *Piezo2*) in both males and females as shown in Fig. 3D in results (lines 785-791). Additionally, we have included a supplementary table (Supplementary Table S8) detailing all commonly restored genes across different treatment and sex combinations, including: genes commonly restored by *B. longum* APC1472 in both females and males, genes commonly restored by FOS+GOS in both females and males, genes commonly restored by *B. longum* APC1472 and FOS+GOS in females, and genes commonly restored by *B. longum* APC1472 and FOS+GOS in males.

We thank the reviewer for this very interesting point.

- xi. Line 647, the difference is not significant for *Lepr*.

We have now updated the text to accurately reflect that the downregulation of *Lepr* expression was non-significant ($q=0.064$), along with the other genes mentioned (*Pomc* and *Ghsr*), and have removed any claims of statistical significance for these findings (lines 880-880). We appreciate the reviewer's attention to detail in ensuring accurate reporting of our statistical results.

- xii. Line 651, Figure 4F doesn't exist (or I couldn't match the text with the figures).

We apologize for the mistake in referencing Figure 4F in line 651. We meant to reference Fig. 4E and have now corrected this reference (lines 885).

- xiii. Line 669: It is only true for females.

We have now updated line 669 to specify that this finding is only true for females (lines 902-903).

- xiv. Line 669: the POMC+ *Ghsr* expressing cells count graph doesn't exist (or I couldn't match the text with the figures).

Prior to submission, we removed the initial graphs including the ARC POMC+ *Ghsr* expressing cells count due to the minor presence of POMC+ cells expressing GHSR receptor in the ARC (less than 10% of POMC express GHSR (Dickson and Luckman,1997). However, it seems we left by error one sentence referring to this initial graph but we have now removed the sentence referencing the POMC+ *Ghsr* expressing cells count graph (lines 902-903).

Dickson, S. L. & Luckman, S. M. Induction of *c-fos* messenger ribonucleic acid in neuropeptide Y and growth hormone (GH)-releasing factor neurons in the rat arcuate nucleus following systemic injection of the GH secretagogue, GH-releasing peptide-6. *Endocrinology* 138, 771-777 (1997).

xv. Line 692: not significant for *Pnoc*.

We apologize for the error regarding *Pnoc* significance. We have now updated the text to highlight that this was a non-significant down-regulation rather than a significant difference (lines 925-926).

xvi. In the section “Enduring molecular impact of an early-life HFHS diet on PNOC GABAergic neurons and *Nod2*-expressing cells in the arcuate nucleus of the hypothalamus of adult male and female mice” please harmonize the writing conventions of the genes expressed and cell populations, this section is very confusing for the reader.

We thank the reviewer for highlighting the inconsistency in gene and cell population nomenclature throughout this section. We have revised the text to establish consistent conventions for describing genes and cell populations. Specifically, we now use:

- Gene names in italics: *Pnoc*, *Nod2*
- Cell populations described as: *Pnoc/Nod2*-expressing cells or PNOC⁺/NOD2⁺ cells
- Co-expressing populations clearly specified as: "*Pnoc*-expressing cells/PNOC⁺ cells that co-express *Nod2*" for instance.

We have also corrected grammatical inconsistencies (e.g., "showed a decreased" to "showed a decrease") to improve readability and scientific precision throughout that section.

Reviewer #4 (Remarks on code availability): The code was not provided so I could not review it

We thank the reviewer for this comment. We have updated the "Code Availability" section of the manuscript to include the GitHub URL where all the code can be accessed. The repository now contains all scripts used for statistical analyses, data processing, visualization and the new integrative multi-omics analyses including the multi-association analyses between blood metabolome and brain transcriptome, and gut microbiome data and blood metabolome.

REVIEWERS' COMMENTS

Reviewer #1 (Remarks to the Author):

I have no additional comments. The authors addressed all my previous comments and concerns very thoroughly. I recommend this manuscript for publication.

Reviewer #1 (Remarks on code availability):

The code is not available. The link provided leads to a "Page not found" error.

Response to Reviewer #1:

We apologize for the broken link to our code repository. We have now corrected this issue and updated both the manuscript and supplementary materials with the proper link that provides full access to all analysis code. We thank the reviewer for bringing this to our attention and for their positive evaluation of our manuscript.

Reviewer #2 (Remarks on code availability):

The link has been provided but the code does not appear to be uploaded yet.

https://github.com/CristinaCuesta-Marti/earlylifeHFHS_feeding_microbiota_2025

Response to Reviewer #2:

We sincerely apologize for the oversight regarding the code repository. As mentioned in our response to Reviewer #1, we have now verified that the link is fully accessible. We have updated both the manuscript and supplementary materials with the correct link providing complete access to all analysis, scripts and data. We thank the reviewer for bringing this to our attention.

Reviewer #4

Remarks to the Author):

The work done by the authors to answer the reviewer's comments both in the manuscript and in the rebuttal is very nice and adds a lot of clarity to the paper. It really helps to understand their work that comprises multiple techniques, analyses and statistical tests. Thank you for taking the time to perform all the changes and answer our comments with clarity and detail and congratulations on this very interesting study.

Remarks on code availability:

Unfortunately, although the Github link to the code and other supplementary material didn't

work on my side and gave me a 404 error. If it is a general problem, please review that there is no typos in the link so the material becomes accessible.

Response to Reviewer #4:

We sincerely thank the reviewer for their positive feedback and for recognizing the improvements made to the manuscript.

Regarding the code availability issue, we apologize for the broken GitHub link. As mentioned in our responses to Reviewers #1 and #2, we have now verified that the link is fully accessible. We have updated both the manuscript and supplementary materials with the correct link. We thank the reviewer for bringing this to our attention.